# Jacobian Sparse Autoencoders: Sparsify Computations, Not Just Activations

Lucy Farnik [1]   Tim Lawson [1]   Conor Houghton [1]   Laurence Aitchison [1]

## Abstract

Sparse autoencoders (SAEs) have been successfully used to discover sparse and human-interpretable representations of the latent activations of LLMs. However, we would ultimately like to understand the computations performed by LLMs and not just their representations. The extent to which SAEs can help us understand computations is unclear because they are not designed to "sparsify" computations in any sense, only latent activations. To solve this, we propose Jacobian SAEs (JSAEs), which yield not only sparsity in the input and output activations of a given model component but also sparsity in the computation (formally, the Jacobian) connecting them. With a naïve implementation, the Jacobians in LLMs would be computationally intractable due to their size. One key technical contribution is thus finding an efficient way of computing Jacobians in this setup. We find that JSAEs extract a relatively large degree of computational sparsity while preserving downstream LLM performance approximately as well as traditional SAEs. We also show that Jacobians are a reasonable proxy for computational sparsity because MLPs are approximately linear when rewritten in the JSAE basis. Lastly, we show that JSAEs achieve a greater degree of computational sparsity on pre-trained LLMs than on the equivalent randomized LLM. This shows that the sparsity of the computational graph appears to be a property that LLMs learn through training, and suggests that JSAEs might be more suitable for understanding learned transformer computations than standard SAEs.

[1]School of Engineering Mathematics and Technology, University of Bristol, Bristol, UK. Correspondence to: Lucy Farnik <lucyfarnik@gmail.com>.

*Proceedings of the 42$^{nd}$ International Conference on Machine Learning*, Vancouver, Canada. PMLR 267, 2025. Copyright 2025 by the author(s).

## 1. Introduction

Sparse autoencoders (SAEs) have emerged as a powerful tool for understanding the internal representations of large language models (Bricken et al., 2023; Cunningham et al., 2023; Gao et al., 2024; Rajamanoharan et al., 2024b; Lieberum et al., 2024; Lawson et al., 2024; Braun et al., 2024; Kissane et al., 2024; Rajamanoharan et al., 2024a). By decomposing neural network activations into sparse, interpretable components, SAEs have helped researchers gain significant insights into how these models process information (Marks et al., 2024; Lieberum et al., 2024; Templeton et al., 2024b; O'Brien et al., 2024; Farrell et al., 2024; Paulo et al., 2024; Balcells et al., 2024; Lan et al., 2024; Brinkmann et al., 2025; Spies et al., 2024).

When trained on the activation vectors from neural network layers, SAEs learn to reconstruct the inputs using a dictionary of sparse 'features', where there are many more features than basis dimensions of the inputs, and each feature tends to capture a specific, interpretable concept. However, the goal of this paper is to improve understanding of *computations* in transformers. While SAEs are designed to disentangle the representations of concepts in the LLM, they are not designed to help us understand the computations performed with those representations.

One approach to understanding computation would be to train two SAEs, one at the input and one at the output of an MLP in a transformer. We can then ask how the MLP maps sparse latent features at the inputs to sparse features in the outputs. For this mapping to be interpretable, it would be desirable that it is sparse, in the sense that each latent in the SAE trained on the output depends on a small number of latents of the SAE trained on the input. These dependencies can be understood as a computation graph or 'circuit' (Olah et al., 2020; Cammarata et al., 2020). SAEs are not designed to encourage this computation graph to be sparse. To address this, we develop Jacobian SAEs (JSAEs), where we include a term in the objective to encourage SAE bases with sparse computational graphs, not just sparse activations. Specifically, we treat the mapping between the latent activations of the input and output SAEs as a function and encourage its Jacobian to be sparse by including an $L^1$ penalty term in the loss function.

With a naïve implementation, it is intractable to compute

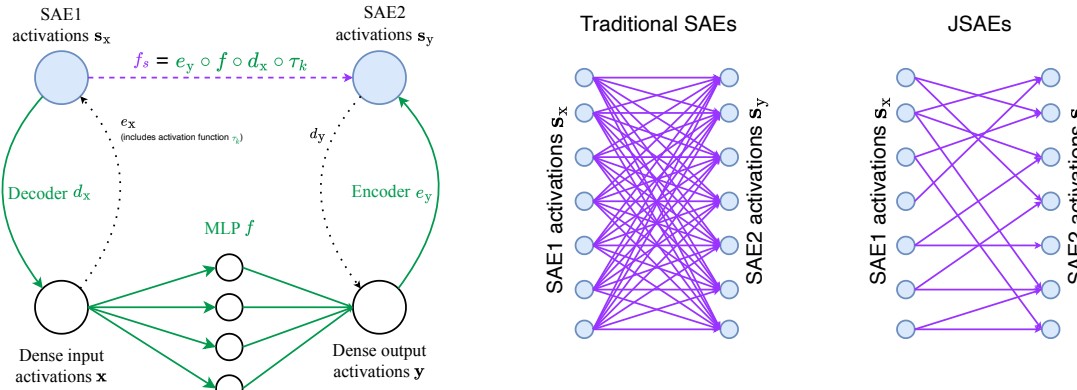

*Figure 1.* A diagram illustrating our setup. We have two SAEs: one trained on the MLP inputs and the other trained on the MLP outputs. We then consider the function $f_s$, which takes the latent activations of the first SAE and returns the latent activations of the second SAE, i.e., $f_s(\mathbf{s}_\mathrm{x}) = \mathbf{s}_\mathrm{y}$. The function $f_s$ is described by the function composition of the TopK activation function of the first (input) SAE $\tau_k$, the decoder of the first SAE $d_\mathrm{x}$, the MLP $f$, and the encoder of the second (output) SAE $e_\mathrm{y}$. We note that the activation function $\tau_k$ is included for computational efficiency only; see Section 4.2 for details. JSAEs optimize for $f_s$ having a sparse Jacobian matrix, which we illustrate by reducing the number of edges in the computational graph that corresponds to $f_s$. Traditional SAEs have sparse SAE latents on either side of the MLP but a dense computational graph between them; JSAEs have both sparse SAE latents *and* a sparse computational graph. Importantly, Jacobian sparsity approximates the computational graph notion, but, as we discuss in Section 5.4 and Appendix B, this approximation is highly accurate due to the fact that $f_s$ is a mostly linear function.

Jacobian matrices because each matrix would have on the order of a trillion elements, even for modestly sized language models and SAEs. Therefore, one of our core contributions is to develop an efficient means to compute Jacobian matrices in this context. The approach we develop makes it possible to train a pair of Jacobian SAEs with only approximately double the computational requirements of training a single standard SAE (Section 4.2). These methods enabled us to make three downstream findings.

First, we find that Jacobian SAEs successfully induce sparsity in the Jacobian matrices between input and output SAE latents relative to standard SAEs without a Jacobian term (Section 5.1). We find that JSAEs achieve the desired increase in the sparsity of the Jacobian with only a slight decrease in reconstruction quality and model performance preservation, which remain roughly on par with standard SAEs. We also find that the input and output latents learned by Jacobian SAEs are approximately as interpretable as standard SAEs, as quantified by auto-interpretability scores. Importantly, we also find that the "computational units" discovered by JSAEs are often highly interpretable – for example, JSAEs find an output latent corresponding to whether the text is in German, which is computed using several input latents corresponding to tokens frequently found in German text (Section 5.2).

Second, inspired by Heap et al. (2025), we investigated the behavior of Jacobian SAEs when applied to random

transformers, i.e., where the parameters have been reinitialized. We find that the degree of Jacobian sparsity that can be achieved when JSAEs are applied to a pre-trained transformer is much greater than the sparsity achieved for a random transformer (Section 5.3). This preliminary finding suggests that Jacobian sparsity may be a useful tool for discovering learned computational structure.

Lastly, we find that Jacobians accurately approximate computational sparsity in this context because the function we are analyzing (i.e., the combination of JSAEs and MLP) is approximately linear (Section 5.4).

Our source code can be found at https://github.com/lucyfarnik/jacobian-saes.

## 2. Related work

### 2.1. Sparse autoencoders

SAEs have been widely applied to 'disentangle' the representations learned by transformer language models into a very large number of concepts, a.k.a. sparse latents, features, or dictionary elements (Sharkey et al., 2022; Cunningham et al., 2023; Bricken et al., 2023; Gao et al., 2024; Rajamanoharan et al., 2024b; Lieberum et al., 2024). Human experiments and quantitative proxies apparently confirm that SAE latents are much more likely to correspond to human-interpretable concepts than raw language-model neurons,

i.e., the basis dimensions of their activation vectors (Cunningham et al., 2023; Bricken et al., 2023; Rajamanoharan et al., 2024a). SAEs have been successfully applied to modifying the behavior of LLMs by using a direction discovered by an SAE to "steer" the model towards a certain concept (Makelov, 2024; O'Brien et al., 2024; Templeton et al., 2024b).

Our work is based on SAEs but has a very different aim: standard SAEs only sparsify activations, while JSAEs also sparsify the computation graph between them (Figure 1).

## 2.2. Transcoders

In this paper, we focus on MLPs. Dunefsky et al. (2024); Templeton et al. (2024a) developed *transcoders*, an alternative SAE-like method to understand MLPs. However, JSAEs and transcoders take radically different approaches and solve radically different problems. This is perhaps easiest to see if we look at what transcoders and JSAEs sparsify. JSAEs are fundamentally an extension of standard SAEs: they train SAEs at the input and output of the MLP and add an extra term to the objective such that these sparse latents are also appropriate for interpreting the MLP (Figure 1). In contrast, transcoders do not sparsify the inputs and outputs; they work with dense inputs and outputs. Instead, transcoders, in essence, sparsify the MLP hidden states. Specifically, a transcoder is an MLP that you train to match (using a mean squared error objective) the input-to-output mapping of the underlying MLP from the transformer. The key difference between the transcoder MLP and the underlying MLP is that the transcoder MLP is much wider, and its hidden layer is trained to be sparse.

Thus, transcoders and JSAEs take fundamentally different approaches. Each transcoder latent tells us 'there is computation in the MLP related to [concept].' By comparison, JSAEs learn a pair of SAEs (which have mostly interpretable latents) and sparse connections between them. At a conceptual level, JSAEs tell us that 'this feature in the MLP's output was computed using only these few input features'. Ultimately, we believe that the JSAE approach, grounded in understanding how the SAE basis at one layer is mapped to the SAE basis at another layer, is potentially powerful and worth thoroughly exploring.

Importantly, it is worth emphasizing that JSAEs and transcoders are asking fundamentally different questions, as can be seen in terms of e.g., differences in what they sparsify. As such, it is not, to our knowledge, possible to design meaningful quantitative comparisons, at least not without extensive future work to develop very general auto-interpretability methods for evaluating methods of understanding MLP circuits.

## 2.3. Automated circuit discovery

In "automated circuit discovery", the goal is to isolate the causally relevant intermediate variables and connections between them necessary for a neural network to perform a given task (Olah et al., 2020). In this context, a circuit is defined as a computational subgraph with an interpretable function. The causal connections between elements are determined via activation patching, i.e., modifying or replacing the activations at a particular site of the model (Meng et al., 2022; Zhang & Nanda, 2023; Wang et al., 2022; Hanna et al., 2023). In some cases, researchers have identified sub-components of transformer language models with simple algorithmic roles that appear to generalize across models (Olsson et al., 2022).

Conmy et al. (2023) proposed a means to automatically prune the connections between the sub-components of a neural network to the most relevant for a given task using activation patching. Given a choice of task (i.e., a dataset and evaluation metric), this approach to automated circuit discovery (ACDC) returns a minimal computational subgraph needed to implement the task, e.g., previously identified 'circuits' like Hanna et al. (2023). Naturally, this is computationally expensive, leading other authors to explore using linear approximations to activation patching (Nanda, 2023; Syed et al., 2024; Kramár et al., 2024). Marks et al. (2024) later improved on this technique by using SAE latents as the nodes in the computational graph.

In a sense, these methods are supervised because they require the user to specify a task. Naturally, it is not feasible to manually iterate over all tasks an LLM can perform, so a fully unsupervised approach is desirable. With JSAEs, we take a step towards resolving this problem, although the architecture introduced in this paper initially only applies to a single MLP layer and not an entire model. Additionally, to the best of our knowledge, no automated circuit discovery algorithm sparsifies the computations inside of MLPs.

There are also other approaches which focus on locating relevant computation in ML models by estimating the contribution of individual model components (Shah et al., 2024; Balasubramanian et al., 2024).

## 3. Background

### 3.1. Sparse autoencoders

In an SAE, we have input vectors, $\mathbf{x} \in \mathcal{X} = \mathbb{R}^{m_\mathrm{x}}$. We want to approximate each vector $\mathbf{x}$ by a sparse linear combination of vectors, $\mathbf{s}_\mathrm{x} \in \mathcal{S}_\mathrm{x} = \mathbb{R}^{n_\mathrm{x}}$. The dimension of the sparse vector, $n_\mathrm{x}$, is typically much larger than the dimension of the input vectors $m_\mathrm{x}$ (i.e. the basis is overcomplete).

In the case of SAEs, we treat the vectors as inputs to an autoencoder with an encoder $e_\mathrm{x} : \mathcal{X} \to \mathcal{S}_\mathrm{x}$ and a decoder

$d_\mathrm{x} : \mathcal{S}_\mathrm{x} \rightarrow \mathcal{X}$ defined by,

$$\mathbf{s}_\mathrm{x} = e_\mathrm{x}(\mathbf{x}) = \phi(\mathbf{W}_\mathrm{x}^\mathrm{enc}\mathbf{x} + \mathbf{b}_\mathrm{x}^\mathrm{enc}) \qquad (1)$$

$$\hat{\mathbf{x}} = d_\mathrm{x}(\mathbf{s}_\mathrm{x}) = \mathbf{W}_\mathrm{x}^\mathrm{dec}\mathbf{s}_\mathrm{x} + \mathbf{b}_\mathrm{x}^\mathrm{dec} \qquad (2)$$

Here, the parameters are the encoder weights $\mathbf{W}_\mathrm{enc} \in \mathbb{R}^{n_\mathrm{x} \times m_\mathrm{x}}$, decoder weights $\mathbf{W}_\mathrm{dec} \in \mathbb{R}^{m_\mathrm{x} \times n_\mathrm{x}}$, encoder bias $\mathbf{b}_\mathrm{x}^\mathrm{enc} \in \mathbb{R}^{n_\mathrm{x}}$, and decoder bias $\mathbf{b}_\mathrm{x}^\mathrm{dec} \in \mathbb{R}^{m_\mathrm{x}}$. The non-linearity $\phi$ can be, for instance, ReLU. These parameters are then optimized to minimize the difference between $\mathbf{x}$ and $\hat{\mathbf{x}}$, typically measured in terms of the mean squared error (MSE), while imposing an $L^1$ penalty on the latent activations $\mathbf{s}_\mathrm{x}$ to incentivize sparsity.

### 3.2. Automatic interpretability of SAE latents

In order to compare the quality of different SAEs, it is desirable to be able to quantify how interpretable its latents are. A popular approach to quantifying interpretability at scale is to collect the examples that maximally activate a given latent, prompt an LLM to generate an explanation of the concept the examples have in common, and then prompt an LLM to predict whether a given prompt activates the SAE latent given the generated explanation. We can then score the accuracy of the predicted activations relative to the ground truth. There are several variants of this approach (e.g., Bills et al., 2023; Choi et al., 2024); in this paper, we use "fuzzing" where the scoring model classifies whether the highlighted tokens in prompts activate an SAE latent given an explanation of that latent (Paulo et al., 2024).

## 4. Methods

The key idea with a Jacobian SAE is to train a pair of SAEs on the inputs and outputs of a neural network layer while additionally optimizing the sparsity of the Jacobian of the function that relates the input and output SAE latent activations (Figure 1). In this paper, we apply Jacobian SAEs to multi-layer perceptrons (MLPs) of the kind commonly found in transformer language models (Radford et al., 2019; Biderman et al., 2023).

### 4.1. Setup

Consider an MLP mapping from $\mathbf{x} \in \mathcal{X}$ to $\mathbf{y} \in \mathcal{Y}$, i.e., $f : \mathcal{X} \rightarrow \mathcal{Y}$ or $\mathbf{y} = f(\mathbf{x})$. We can then train two $k$-sparse SAEs, one on $\mathbf{x}$ and the other on $\mathbf{y}$. The resulting SAEs map from each of $\mathbf{x}$ and $\mathbf{y}$ to corresponding sparse latents $\mathbf{s}_\mathrm{x} \in \mathcal{S}_\mathrm{x}$ and $\mathbf{s}_\mathrm{y} \in \mathcal{S}_\mathrm{y}$, i.e., $\mathbf{s}_\mathrm{x} = e_\mathrm{x}(\mathbf{x})$ and $\mathbf{s}_\mathrm{y} = e_\mathrm{y}(\mathbf{y})$, where $e_\mathrm{x}$ is the encoder of the first SAE and $e_\mathrm{y}$ is the encoder of the second SAE. Each of these SAEs also has a decoder that maps from the sparse latents back to an approximation of the original vector: $\hat{\mathbf{x}} = d_\mathrm{x}(\mathbf{s}_\mathrm{x})$ and $\hat{\mathbf{y}} = d_\mathrm{y}(\mathbf{s}_\mathrm{y})$.

We may now consider the function $f_s : S_X \rightarrow S_Y$, which intuitively represents the function, $f$, but written in terms of the sparse bases learned by the SAE pair for the original vectors $\mathbf{x}$ and $\mathbf{y}$. Specifically, we define $f_s$ by

$$f_s = e_\mathrm{y} \circ f \circ d_\mathrm{x} \circ \tau_k \qquad (3)$$

where $\circ$ denotes function composition. Here, $d_\mathrm{x} : \mathcal{S}_\mathrm{x} \rightarrow \mathcal{X}$ maps the sparse latents given as input to $f_s$ to "dense" inputs. Then, $f : \mathcal{X} \rightarrow \mathcal{Y}$ maps the dense inputs to dense outputs. Finally, $e_\mathrm{y} : \mathcal{Y} \rightarrow \mathcal{S}_\mathrm{y}$ maps the dense outputs to sparse outputs. Note that $f_s$ first applies the TopK activation function $\tau_k$ to the sparse inputs, $\mathbf{s}_\mathrm{x}$. Critically, with $k$-sparse SAEs, we produce the sparse inputs by $\mathbf{s}_\mathrm{x} = e_\mathrm{x}(\mathbf{x})$, implying that $\mathbf{s}_\mathrm{x}$ only has $k$ non-zero elements. In that setting, TopK does not change the inputs, i.e. $\mathbf{s}_\mathrm{x} = \tau_k(\mathbf{s}_\mathrm{x})$, but it does affect the Jacobian and, in particular, allows us to compute it much more efficiently (Section 4.2).

At a high level, we want the function $f_s$ to be 'sparse', in the sense that each of its input dimensions (i.e. SAE latent activations) only affects a small number of its output dimensions, and each of its output dimensions only depends on a small number of its input dimensions. We quantify the sparsity of $f_s$ in terms of its Jacobian matrix. The Jacobian of $f_s$ is, in index notation:

$$J_{f_s,i,j} = \frac{\partial f_{s,i}(s_\mathrm{x})}{\partial s_{\mathrm{x},j}}. \qquad (4)$$

Intuitively, we can consider maximizing the sparsity of the Jacobian as minimizing the number of edges in the computational graph connecting the input and output nodes (Figure 1), i.e. maximizing the number of near-zero elements in the Jacobian matrix. We note that the Jacobian is not a perfect measure of the sparsity of the computational graph, but it is an accurate proxy (see Section 5.4 and Appendix B) while being computationally tractable.

We simultaneously train two separate SAEs on the input and output of a transformer MLP with the objectives of low reconstruction error and sparse relations between the separate SAE latents (via the Jacobian). We do not need to optimize for the sparsity of the latent activations via a penalty term in the loss function because we use $k$-sparse autoencoders, which keep only the $k$ largest latent activations per token position. Hence, our loss function is

$$\mathcal{L} = \mathrm{MSE}(\mathbf{x}, \hat{\mathbf{x}}) + \mathrm{MSE}(\mathbf{y}, \hat{\mathbf{y}}) + \frac{\lambda}{k^2} \sum_{i=1}^{n_\mathrm{y}} \sum_{j=1}^{n_\mathrm{x}} |J_{f_s,i,j}| \quad (5)$$

Here, $k$ is the number of non-zero elements in the TopK activation function, $n_\mathrm{x}, n_\mathrm{y}$ are the dimensionalities of the latent spaces of the input and output SAEs, respectively, and $\lambda$ is the coefficient of the Jacobian loss term. We divide by $k^2$ because, as we will see later, there are at most $k^2$ non-zero elements in the Jacobian. Finally, note that if we set $\lambda = 0$, then our objective effectively trains traditional SAEs for each of $\mathbf{x}$ and $\mathbf{y}$ independently.

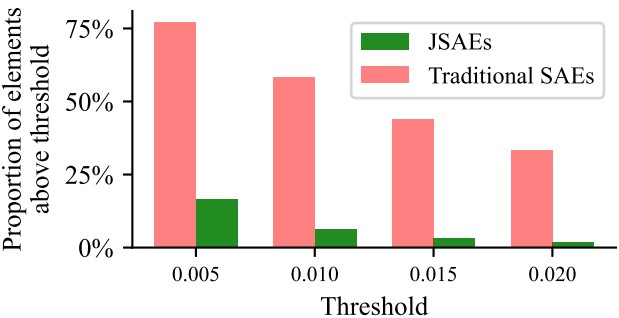

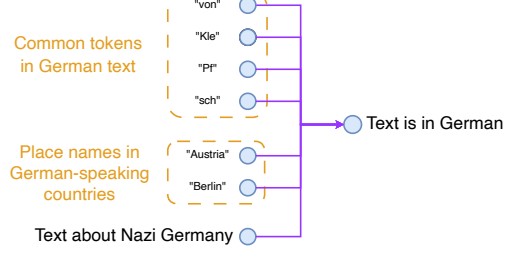

*Figure 2.* JSAEs induce a much greater degree of sparsity in the elements of the Jacobian of $f_s$ than traditional SAEs. The bars show the average proportion of Jacobian elements with absolute values above certain thresholds. At most $k \times k$ elements can be nonzero, so we take 100% on the y-axis to mean $k \times k$. The average was taken across 10 million tokens. This example is from layer 15 of Pythia-410m. For layer 3 of Pythia-70m and layer 7 of Pythia-160m, see Figure 34, for more quantitative information on Jacobian sparsity across model sizes, layers, and hyperparameters see Figures 24, 25, and 26. We present further discussion of the sparsity of the Jacobian in Appendix F.

*Figure 3.* JSAEs allow us to locate the "input features" of each feature computed by the MLP. For instance, in Pythia-410m, the MLP at layer 15 is computing the feature "this text is in German". JSAEs discover the inputs which the MLP uses to decide whether this feature should be on or off. These inputs correspond to tokens frequently found in German text, place names in German-speaking countries, and text about Nazi Germany. See Appendix C for details.

## 4.2. Making the Jacobian calculation tractable

Computing the Jacobian naively (e.g., using an automatic differentiation package) is computationally intractable, as the full Jacobian has size $B \times n_y \times n_x$ where $B$ is the number of tokens in a training batch $n_x$ is the number of SAE latents for the input, and $n_y$ is the number of SAE latents for the output. Unfortunately, typical values are around $1,000$ for $B$ and around $32,000$ for $n_x$ and $n_y$ (taking as an example a model dimension of $1,000$ and an expansion factor of 32). Combined, this gives a Jacobian with around 1 trillion elements. This is obviously far too large to work with in practice, and our key technical contribution is to develop an efficient approach to working with this huge Jacobian.

Our first insight is that for each element of the batch, we have a $n_y \times n_x$ Jacobian, where $n_x$ and $n_y$ are around $32,000$. This is obviously far too large. However, remember that we are interested in the Jacobian of $f_s$, so the input is the sparse SAE latent vector, $\mathbf{s}_x$ and the output is the sparse SAE latent vector, $\mathbf{s}_y$. Importantly, as we are using $k$-sparse SAEs, only $k$ elements of the input and output are "on" for any given token. As such, we really only care about the $k \times k$ elements of the Jacobian of $f_s$, corresponding to the inputs and outputs that are "on". This reduces the size of the Jacobian by around six orders of magnitude, and renders the computation tractable. However, to make this work formally, we need all elements of the Jacobian corresponding to "off" elements of the input and output to be zero. This is where the $\tau_k$ in the definition of $f_s$ becomes important. Specifically, the $\tau_k$ ensures that the gradient of

$f_s$ wrt any of the inputs that are "off" is zero. Without $\tau_k$, the Jacobian could be non-zero for any of the inputs, even if changing those inputs would not make sense, as it would give more than $k$ elements being "on" in the input, and thus could not be produced by the k-sparse SAE.

Our second insight was that computing the Jacobian by automatic differentiation would still be relatively inefficient, e.g., requiring $k$ backward passes. Instead, for standard GPT-2-style MLPs, we noticed that an extremely efficient Jacobian formula can be derived by hand, requiring only three matrix multiplications and along with a few pointwise operations. We present this derivation in Appendix A.

With these optimizations in place, training a pair of JSAEs takes about twice as long as training a single standard SAE. We measured this by training ten of each model on Pythia-70m with an expansion factor of 32 for 100 million tokens on an RTX 3090. The average training durations were 72mins for a pair of JSAEs and 33 mins for a traditional SAE, with standard deviations below 30 seconds for both.

## 5. Results

Our experiments were performed on LLMs from the Pythia suite (Biderman et al., 2023), the figures in the main text contain results from Pythia-410m unless otherwise specified. We trained on 300 million tokens with $k = 32$ and an expansion factor of $64$ for Pythia-410m and 32 for smaller models. We reproduced all our experiments on multiple models and found the same qualitative results (see Appendix E).

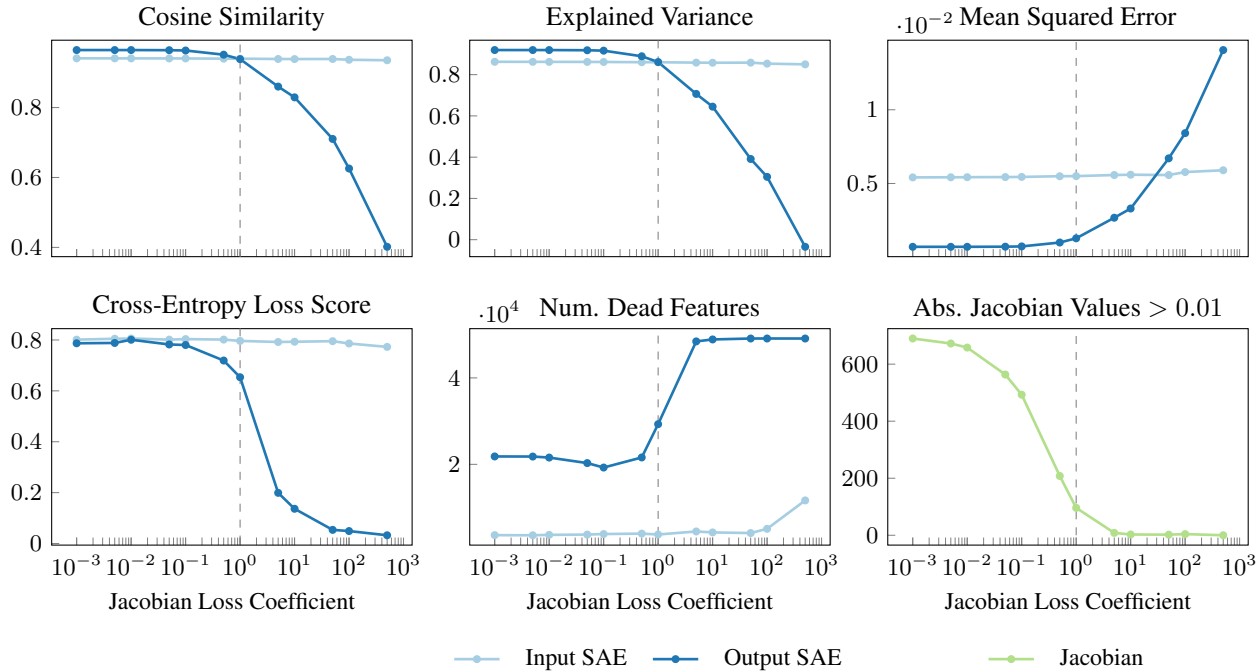

*Figure 4.* Reconstruction quality, model performance preservation, and sparsity metrics against the Jacobian loss coefficient. JSAEs trained on layer 7 of Pythia-160m with expansion factor 64 and $k = 32$; see Figure 26 for layer 3 of Pythia-70m. Recall that the maximum number of non-zero Jacobian values is $k^2 = 1024$. In accordance with Figure 5, all evaluation metrics degrade for values of the coefficient above 1. See Appendix E for details of the evaluation metrics.

### 5.1. Jacobian sparsity, reconstruction quality, and auto-interpretability scores

First, we compared the Jacobian sparsity for standard SAEs and JSAEs. Note that, unlike with SAE latent activations, there is no mechanism for producing exact zeros in the Jacobian elements corresponding to active latents. Hence, we consider the number of near-zero elements rather than the number of exact zeros. To quantify the difference in sparsity between the two, we looked at the proportion of the elements of the Jacobian above a particular threshold when aggregating over 10 million tokens (Figure 2). Here, we found that JSAEs dramatically reduced the number of large elements of the Jacobian relative to traditional SAEs. We also note that the Jacobians are not only sparse on each individual token, but also when averaged across a large number of tokens (see Figure 36 in the appendix).

Importantly, the degree of sparsity depends on our choice of the coefficient $\lambda$ of the Jacobian loss term. Therefore, we trained multiple JSAEs with different values of this parameter. As we might expect, for small values of $\lambda$, i.e., little incentive to sparsify the Jacobian, the input and output SAEs perform similarly to standard SAEs (Figure 4 blue lines), including in terms of the variance explained by the reconstructed activation vectors and the increase in the cross-entropy loss when the input activations are replaced by their reconstructions. Unsurprisingly, as $\lambda$ grows larger and the Jacobian loss term starts to dominate, our evaluation metrics degrade. Interestingly, this degradation happens almost entirely in the output SAE rather than the input SAE — we leave it to future work to investigate this phenomenon further.

Critically, Figure 4 suggests there is a 'sweet spot' of the $\lambda$ hyperparameter where the SAE quality metrics remain reasonable, but the Jacobian is much sparser than for standard SAEs. To further investigate this trade-off, we plotted a measure of Jacobian sparsity (the proportion of elements of the Jacobian above 0.01) against the average cross-entropy (Figures 4, 5, and 29). We found that there is indeed a sweet spot where the average cross-entropy is only slightly worse than a traditional SAE, while the Jacobian is far sparser. For Pythia 410m (Figure 5) this value is around $\lambda = 0.5$, whereas for Pythia-70m, it is around $\lambda = 1$ (Figure 29). We choose this value of the Jacobian coefficient (i.e. $\lambda = 0.5$ for Pythia-410m in the main text, and $\lambda = 1$ for Pythia-160m in the Appendix) in other experiments.

We also measure the interpretability of JSAE latents using the automatic interpretability pipeline developed by Paulo et al. (2024) and compare this to traditional SAEs. We find that JSAEs achieve similar interpretability scores (Figure 6).

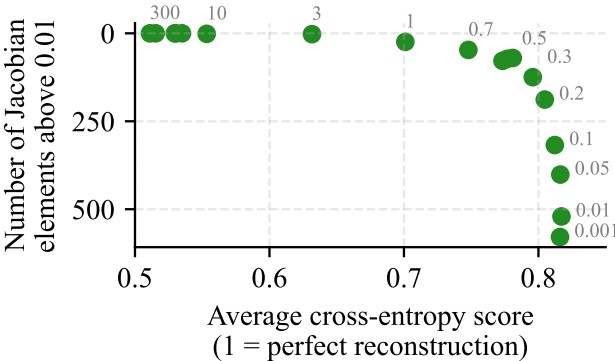

*Figure 5.* The trade-off between reconstruction quality and Jacobian sparsity as we vary the Jacobian loss coefficient. Each dot represents a pair of JSAEs trained with a specific Jacobian coefficient. The value of $\lambda$ is included for some points. We can see that a coefficient of roughly $\lambda = 0.5$ is optimal for Pythia-410m with $k = 32$. Note that the CE loss score is the average of the CE loss scores of the pre-MLP JSAE and the post-mlp JSAE. Measured on layer 15 of Pythia-410m, similar charts with a wider range of models and metrics can be found in Figures 27, 28, and 29.

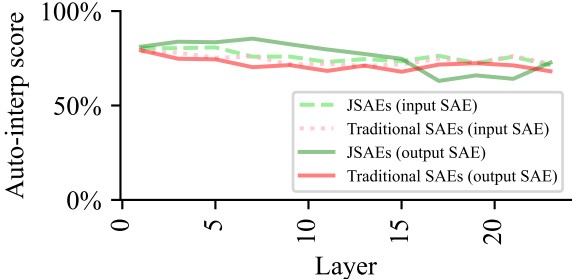

*Figure 6.* Automatic interpretability scores of JSAEs are very similar to traditional SAEs. Measured on all odd-numbered layers of Pythia-410m using the "fuzzing" scorer from Paulo et al. (2024). For all layers of Pythia-70m see Figure 37.

## 5.2. Max-activating examples of JSAEs

Next, we interpreted the "max-activating" examples of JSAEs in order to verify that JSAEs can locate semantically meaningful computational units. Namely, we took the latents of the output SAE $i$ which have large Jacobian values when averaging across a wide distribution of text. Then for each output SAE latent $i$, we found the 10 input SAE latents $j$ which have the largest average Jacobian elements $J_{f_s, i, j}$. We find that these combinations are often highly interpretable. For example, as shown in Figure 3, the very first output latent of layer 15 of Pythia-410m as sorted by average Jacobian value corresponds to "this text is in German". We find that it is computed as a function of input latents corresponding to:

- Tokens which frequently appear in German text, such

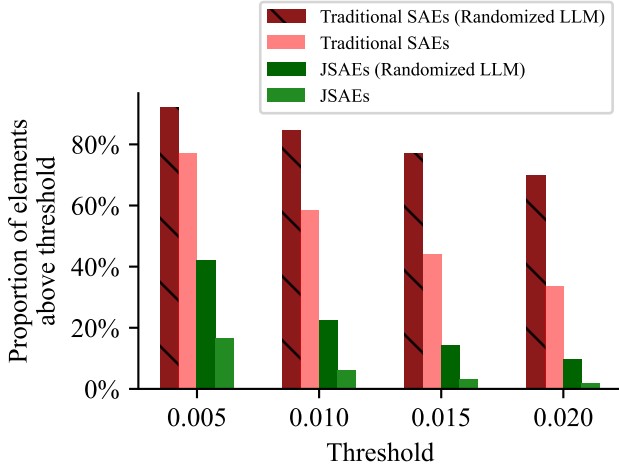

*Figure 7.* Jacobians are substantially more sparse in pre-trained LLMs than in randomly initialzied transformers. This holds both when you actively optimize for Jacobian sparsity with JSAEs, and when you don't optimize for it and use traditional SAEs. The figure shows the proportion of Jacobian elements with absolute values above certain thresholds. At most $k^2$ elements can be nonzero, we therefore take $k^2$ to be 100% on the y-axis. Jacobians are significantly more sparse in pre-trained transformers than in randomly re-initialized transformers. This shows that Jacobian sparsity is, at least to some extent, connected to the structures that LLMs learn during training. This stands in contrast to recent work by Heap et al. (2025) showing that traditional SAEs achieve roughly equal auto-interpretability scores on randomly initialized transformers as they do on pre-trained LLMs. Measured on layer 15 of Pythia-410m, for layer 3 of Pythia-70m see Figure 38. Averaged across 10 million tokens.

as "Pf", "sch", "Kle", and "von"

- Names of places where people speak German, such as "Berlin" or "Austria"

- Words and phrases related to the Third Reich, such as "Nazi", "concentration camp", "Hitler", and "Holocaust"

For a few of handpicked examples, see Appendix C. A large number of examples which are not handpicked is available at tinyurl.com/jsaes-qualitative.

## 5.3. Performance on re-initialized transformers

To confirm that JSAEs are extracting information about the complex learned computation, we considered a form of control analysis inspired by Heap et al. (2025). Specifically, we would expect that trained transformers have carefully learned specific, structured computations while randomly initialized transformers do not. Thus, a possible desideratum for tools in mechanistic interpretability is that they

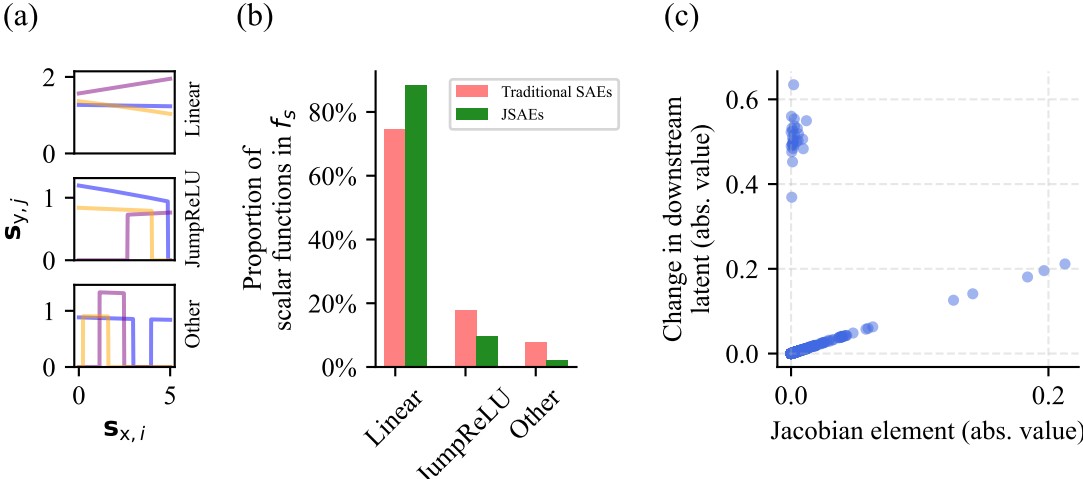

*Figure 8.* The function $f_s$, which combines the decoder of the first SAE, the MLP, and the encoder of the second SAE, is mostly linear. Specifically, the vast majority of scalar functions going from $s_{x,j}$ to $s_{y,i}$ are linear. (a) Examples of linear, JumpReLU, and other functions relating individual input SAE latents and output SAE latents. See Figure 9 for more examples. (b) For the empirically observed $s_x$ and randomly selected $i, j$ (of those corresponding to active SAE latents), the vast majority of scalar functions from $s_{x,j}$ to $s_{y,i}$ are linear. For details see Appendix B. The proportion of linear function also noticeably increases with JSAEs compared to traditional SAEs, meaning that JSAEs induce additional linearity in $f_s$. (c) Because the vast majority of functions are linear, the Jacobian usually precisely predicts the change observed in the output SAE latent when we make a large change to the input SAE latent's value (namely subtracting 1, note that the empirical median value of $s_{x,j}$ is 2.5). Each dot corresponds to an $(s_{x,j}, s_{y,i})$ pair. For 97.7% of pairs (across a sample size of 10 million) their Jacobian value nearly exactly predicts the change we see in the output SAE latent when making large changes to the input SAE latent's activation, i.e. $|\Delta s_{y,i}| \approx |J_{f_s,ij}|$. The scatter plot shows a randomly selected subset of 1,000 $(s_{x,j}, s_{y,i})$ pairs. For further details see Appendix B. Measured on layer 15 of Pythia-410m, for layer 3 of Pythia-70m see Figure 39, for the linearity results on other models and hyperparameters see Figures 15, 16, and 17.

ought to work substantially better when analyzing the complex computations in trained LLMs than when applied to LLMs with randomly re-initialized weights. This is precisely what we find. Specifically, we find that the Jacobians for trained networks are always substantially sparser than the corresponding random trained network, and this holds for both traditional SAEs and JSAEs (Figure 7). Further, the relative improvement in sparsity from the traditional SAE to the JSAE is much larger for trained than random LLMs, again indicating that JSAEs are extracting structure that only exists in the trained network. Note that we also see that for traditional SAEs, there is a somewhat more sparse Jacobian for the trained than randomly initialized transformer. This makes sense: we would hope that the traditional SAE basis is somewhat more aligned with the computation (as expressed by a sparse Jacobian) than we would expect by chance. However, it turns out that without a "helping hand" from the Jacobian sparsity term, the alignment in a traditional SAE is relatively small. Thus, Jacobian sparsity is a property related to the complex computations LLMs learn during training, which should make it substantially useful for discovering the learned structures of LLMs.

### 5.4. $f_s$ is mostly linear

Importantly, the Jacobian is a local measure. Thus, strictly speaking, a near-zero element of the Jacobian matrix implies only that a small change to the input SAE latent does not affect the corresponding output SAE latent. It may, however, still be the case that a large change to the input SAE latent would change the output SAE latent. We investigated this question and found that $f_s$ is usually approximately linear in a wide range and is often close to linear. Specifically, of the scalar functions relating individual input SAE latents $s_{x,j}$ to individual output SAE latents $s_{y,i}$, the vast majority are linear (Figure 8b). This is important because, for any linear function, its local slope is completely predictive of its global shape, and therefore, a near-zero Jacobian element implies a near-zero causal relationship. For the scalar functions which are not linear, we frequently observed they have a JumpReLU structure[1] (Erichson et al., 2019). Notably, a JumpReLU is linear in a subset of its input space, so even for these scalar functions the first derivative is still an accurate measure within some range of $s_{x,j}$ values. It is also worth

---

[1] By JumpReLU, we mean any function of the form $f(x) = a\text{JumpReLU}(bx + c)$. Recall that $\text{JumpReLU}(x) = x$ if $x > d$ and 0 otherwise. $a, b, c, d \in \mathbb{R}$ are constants.

noting that with JSAEs, the proportion of linear functions is noticeably higher than with traditional SAEs, so at least to a certain extent, JSAEs induce additional linearity in the MLP. To confirm these results, we plotted the Jacobian against the change of output SAE latent $s_{y,i}$ as we change the input SAE latent $s_{x,j}$ by subtracting 1 (Figure 8c)[2]. We found that 97.7% of the time, $|\Delta s_{y,i}| \approx |J_{f_s,ij}|$. For details see Appendix B. While these results are strongly suggestive, we would caution that it is difficult to interpret them definitively as we are not evaluating the reconstruction error for a linear model fitted to the input-output relationship for the MLP latents.

## 6. Conclusion

We introduced Jacobian sparse autoencoders (JSAEs), a new approach for discovering sparse computation in LLMs in a fully unsupervised way. We found that JSAEs induce sparsity in the Jacobian matrix of the function that represents an MLP layer in the sparse basis found by JSAEs, with minimal degradation in the reconstruction quality and downstream performance of the underlying model and no degradation in the interpretability of latents. We demonstrated that the computation found by JSAEs is often highly interpretable, allowing us to see not only the concepts computed by MLPs, but also the "input concepts" which are used to compute each "output concept". We also found that Jacobian sparsity is substantially greater in pre-trained LLMs than in randomly initialized ones suggesting that Jacobian sparsity is indeed a proxy for learned computational structure. Lastly, we found that Jacobians are a highly accurate measure of computational sparsity due to the fact that the MLP in the JSAE basis consists mostly of linear functions relating input to output JSAE latents.

## Acknowledgements

The authors wish to thank Callum McDougall and Euan Ong for helpful discussions. We also thank the contributors to the open-source mechanistic interpretability tooling ecosystem, in particular the authors of SAELens (Bloom et al., 2024), which formed the backbone of our codebase. The authors wish to acknowledge and thank the financial support of the UK Research and Innovation (UKRI) [Grant ref EP/S022937/1] and the University of Bristol. This work was carried out using the computational facilities of the Advanced Computing Research Centre, University of Bristol - http://www.bristol.ac.uk/acrc/. We would like to thank Dr. Stewart for funding for GPU resources.

---

[2]For reference, the median value of $s_{x,j}$ without any interventions is 2.5.

## Impact Statement

The work presented in this paper advances the field of mechanistic interpretability. Our hope is that interpretability will prove beneficial in making LLMs safer and more robust in ways ranging from better detection of model misuse to editing LLMs to remove dangerous capabilities.

## Author contribution statement

Conceptualization was done by LF and LA. Derivation of an efficient way to compute the Jacobian was done by LF and LA. Implementation of the training codebase was done by LF. The experiments in Jacobian sparsity, auto-interpretability, reconstruction quality, and approximate linearity of $f_s$ were done by LF. Qualitative examples of the computations found by JSAEs were done by TL. LA and CH provided supervision and guidance throughout the project. The text was written by LF, LA, TL, and CH. Figures were created by LF and TL with advice from LA and CH.

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

## A. Efficiently computing the Jacobian

A simple form for the Jacobian of the function $f_s = e_y \circ f \circ d_x \circ \tau_k$, which describes the action of an MLP layer $f$ in the sparse input and output bases, follows from applying the chain rule. Note that here, the subscripts $f_s$, $e_y$, etc. denote the function in question rather than vector or matrix indices. For the GPT-2-style MLPs that we study, the components of $f_s$ are:

1. **TopK**. This function takes sparse latents $\mathbf{s}_x$ and outputs sparse latents $\bar{\mathbf{s}}_x$. Importantly, $\mathbf{s}_x = \bar{\mathbf{s}}_x$. This step makes the backward pass of the Jacobian computation more efficient but does not affect the forward pass.

$$\bar{\mathbf{s}}_x = \tau_k(\mathbf{s}_x) \tag{6}$$

2. **Input SAE Decoder**. This function takes sparse latents $\bar{\mathbf{s}}_x$ and outputs dense MLP inputs $\hat{\mathbf{x}}$:

$$\hat{\mathbf{x}} = d_x(\bar{\mathbf{s}}_x) = \mathbf{W}_x^{\text{dec}} \bar{\mathbf{s}}_x + \mathbf{b}_x^{\text{dec}} \tag{7}$$

3. **MLP**. This function takes dense inputs $\hat{\mathbf{x}}$ and outputs dense outputs $\mathbf{y}$:

$$\mathbf{z} = \mathbf{W}_1 \hat{\mathbf{x}} + \mathbf{b}_1 \,, \quad \mathbf{y} = \mathbf{W}_2 \phi_{\text{MLP}}(\mathbf{z}) + \mathbf{b}_2 \tag{8}$$

   where $\phi_{\text{MLP}}$ is the activation function of the MLP (e.g., GeLU in the case of Pythia models).

4. **Output SAE Encoder**. This function takes dense outputs $\mathbf{y}$ and outputs sparse latents $\mathbf{s}_y$:

$$\mathbf{s}_y = e_y(\mathbf{y}) = \tau_k \left( \mathbf{W}_y^{\text{enc}} \mathbf{y} + \mathbf{b}_y^{\text{enc}} \right) \tag{9}$$

The Jacobian $\mathbf{J}_{f_s} \in \mathbb{R}^{n_y \times n_x}$ for a single input activation vector has the following elements, in index notation:

$$J_{f_s, ij} = \frac{\partial s_{y,i}}{\partial s_{x,j}} = \sum_{k\ell mn} \frac{\partial s_{y,i}}{\partial y_k} \frac{\partial y_k}{\partial z_\ell} \frac{\partial z_\ell}{\partial \hat{x}_m} \frac{\partial \hat{x}_m}{\partial \bar{s}_{x,n}} \frac{\partial \bar{s}_{x,n}}{\partial s_{x,j}} \tag{10}$$

We compute each term like so:

1. **Output SAE Encoder derivative**:

$$\frac{\partial s_{y,i}}{\partial y_k} = \tau_k' \left( \sum_j W_{ij}^{\text{enc}} y_j + b_{\text{enc},i} \right) W_{y,ik}^{\text{enc}} = \begin{cases} W_{y,ik}^{\text{enc}} & \text{if } i \in \mathcal{K}_2 \\ 0 & \text{otherwise} \end{cases} \tag{11}$$

   where $\mathcal{K}_2$ is the set of indices selected by the TopK activation function $\tau_k$ of the second (output) SAE. Importantly, the subscript $k$ *does not* indicate the $k$-th element of $\tau_k$, whereas it *does* indicate the $k$-th column of $W_{y,ik}^{\text{enc}}$.

2. **MLP derivatives**:

$$\frac{\partial y_k}{\partial z_\ell} = W_{2,k\ell} \, \phi_{\text{MLP}}'(z_\ell) \,, \quad \frac{\partial z_\ell}{\partial \hat{x}_m} = W_{1,\ell m} \tag{12}$$

3. **Input SAE Decoder derivative**:

$$\frac{\partial \hat{x}_m}{\partial \bar{s}_{x,n}} = W_{x,mn}^{\text{dec}} \tag{13}$$

4. **TopK derivative**:

$$\frac{\partial \bar{s}_{x,n}}{\partial s_{x,j}} = \begin{cases} 1 & \text{if } j \in \mathcal{K}_1 \\ 0 & \text{otherwise} \end{cases} \tag{14}$$

   where $\mathcal{K}_1$ is the set of indices (corresponding to SAE latents) that were selected by the TopK activation function $\tau_k$ of the first (input) SAE, which we explicitly included in the definition of $f_s$ above.

When we combine all the terms:

$$J_{f_s,ij} = \begin{cases} \sum_{k\ell m} W_{\text{y},ik}^{\text{enc}} W_{2,k\ell} \, \phi'_{\text{MLP}}(z_\ell) \, W_{1,\ell m} \, W_{\text{x},mj}^{\text{dec}} & \text{if } i \in \mathcal{K}_2 \wedge j \in \mathcal{K}_1 \\ 0 & \text{otherwise} \end{cases} \tag{15}$$

Let $\mathbf{W}_{\text{y}}^{\text{enc(active)}} \in \mathbb{R}^{k \times m_{\text{y}}}$ and $\mathbf{W}_{\text{x}}^{\text{dec(active)}} \in \mathbb{R}^{m_{\text{x}} \times k}$ contain the active rows and columns, i.e., the rows and columns corresponding to the $\mathcal{K}_2$ or $\mathcal{K}_1$ indices respectively. The Jacobian then simplifies to:

$$\mathbf{J}_{f_s}^{\text{(active)}} = \underbrace{\mathbf{W}_{\text{y}}^{\text{enc(active)}}\mathbf{W}_2}_{\mathbb{R}^{k \times d_{\text{MLP}}}} \cdot \underbrace{\phi'_{\text{MLP}}(\mathbf{z})}_{\mathbb{R}^{d_{\text{MLP}} \times d_{\text{MLP}}}} \cdot \underbrace{\mathbf{W}_1 \mathbf{W}_{\text{x}}^{\text{dec(active)}}}_{\mathbb{R}^{d_{\text{MLP}} \times k}} \tag{16}$$

where $d_{\text{MLP}}$ is the hidden size of the MLP. Note that $\mathbf{J}_{f_s}^{\text{(active)}}$ is of size $k \times k$, while the full Jacobian matrix $\mathbf{J}_{f_s}$ is of size $n_{\text{y}} \times n_{\text{x}}$. However, $\mathbf{J}_{f_s}^{\text{(active)}}$ contains all the nonzero elements of $\mathbf{J}_{f_s}$, so it is all we need to compute the loss function to train Jacobian SAEs (Section 4.1).

### A.1. JSAEs with GLUs

The equations above can be easily adapted to work with gated linear units (GLUs), which are significantly more common in modern LLMs than GPT-2-style MLPs.

To do this, we modify the MLP equations like so:

$$\mathbf{g} = \mathbf{W}_{\text{g}}\hat{\mathbf{x}} + \mathbf{b}_{\text{g}} \tag{17}$$
$$\mathbf{s} = \phi_{\text{MLP}}(\mathbf{g}) \tag{18}$$
$$\mathbf{h} = \mathbf{W}_1\hat{\mathbf{x}} + \mathbf{b}_1 \tag{19}$$
$$\mathbf{z} = \mathbf{h} \odot \mathbf{s} \tag{20}$$
$$\mathbf{y} = \mathbf{W}_2\mathbf{z} + \mathbf{b}_2 \tag{21}$$

where $\odot$ is elementwise multiplication.

We then modify the derivatives accordingly:

$$\frac{\partial y_k}{\partial z_\ell} = W_{2,k\ell} \tag{22}$$
$$\frac{\partial z_\ell}{\partial \hat{x}_m} = h_\ell \frac{\partial s_\ell}{\partial \hat{x}_m} + s_\ell \frac{\partial h_\ell}{\partial \hat{x}_m} \tag{23}$$
$$\frac{\partial h_\ell}{\partial \hat{x}_m} = W_{1,\ell m} \tag{24}$$
$$\frac{\partial s_\ell}{\partial g_\ell} = \phi'_{\text{MLP}}(g_\ell) \tag{25}$$
$$\frac{\partial g_\ell}{\partial \hat{x}_m} = W_{\text{g},\ell m} \tag{26}$$
$$\tag{27}$$

Combining the terms again:

$$J_{f_s,ij} = \begin{cases} \sum_{k\ell m} W_{\text{y},ik}^{\text{enc}} W_{2,k\ell} \left( h_\ell \, \phi'_{\text{MLP}}(g_\ell) \, W_{\text{g},\ell m} + s_\ell \, W_{1,\ell m} \right) W_{\text{x},mj}^{\text{dec}} & \text{if } i \in \mathcal{K}_2 \wedge j \in \mathcal{K}_1 \\ 0 & \text{otherwise} \end{cases} \tag{28}$$

The Jacobian is then:

$$\mathbf{J}_{f_s}^{\text{(active)}} = \underbrace{\mathbf{W}_{\text{y}}^{\text{enc(active)}}\mathbf{W}_2}_{\mathbb{R}^{k \times d_{\text{MLP}}}} \cdot \underbrace{\left(\text{diag}\left(\mathbf{h} \odot \phi'_{\text{MLP}}(\mathbf{g})\right)\mathbf{W}_{\text{g}} + \text{diag}(\mathbf{s})\mathbf{W}_1\right)}_{\mathbb{R}^{d_{\text{MLP}} \times m_{\text{x}}}} \cdot \underbrace{\mathbf{W}_{\text{x}}^{\text{dec(active)}}}_{\mathbb{R}^{m_{\text{x}} \times k}} \tag{29}$$

## B. $f_s$ is approximately linear

Consider the scalar function $f_{s,(i,j)}|_{\mathbf{s_x}} : \mathbb{R} \to \mathbb{R}$ which takes as input the $j$-th latent activation of the first SAE (i.e. $s_{\mathbf{x},j}$) and returns as output the $i$-th latent activation of the second SAE (i.e., $s_{\mathbf{y},i}$), while keeping the other elements of the input vector fixed at the same values as $\mathbf{s_x}$. In other words, this function captures the relationship between the $j$-th input SAE latent and the $i$-th output SAE latent in the context of $\mathbf{s_x}$. Geometrically, we start off at the point $\mathbf{s_x}$, and we move from it through the input spaces in parallel to the $j$-th basis vector, and then we observe how the output of $f_s$ projects onto the $i$-th basis vector. Formally,

$$f_{s,(i,j)}|_{\mathbf{s_x}}(x) = f_s\left(\psi(\mathbf{s_x}, i, x)\right)_j \tag{30}$$

$$\psi(\mathbf{s_x}, i, x)_k = \begin{cases} x & \text{if } i = k \\ s_{\mathbf{x},j} & \text{otherwise} \end{cases} \tag{31}$$

These are the functions shown in Figure 8a, of which the vast majority are linear (Figure 8b).

As we showed in Figure 8c, the absolute value of a Jacobian element nearly perfectly predicts the change we see in the output SAE latent activation value when we make a large intervention on the input SAE latent activation. However, in the same figure, there is a small cluster of approximately 2.5% of samples, where the Jacobian element is near zero, but the change observed in the downstream feature is quite large. We proceed by exploring the cause behind this phenomenon.

Note that each point in Figure 8 corresponds to a single scalar function $f_{s,(i,j)}|_{\mathbf{s_x}}$ (a pair of latent indices). An expanded version of Figure 8 is presented in Figure 10. Importantly, we show the 'line', the top-left cluster, and outliers visible in Figure 8 in different colors, which we re-use in the following charts (Figures 11 and 12). It also includes 10K samples, compared to 1K in Figure 8c: as above, most samples remain on the line, but the greater number of samples makes the behavior of the top-left cluster and outliers clearer.

Figure 11 illustrates some examples of functions $f_{s,(i,j)}|_{\mathbf{s_x}}$ taken from each category shown in Figure 8, i.e., the line, cluster, and outliers. The vast majority of functions belong to the line category and are typically either linear or akin to JumpReLU activation functions (which include step functions as a special case). By contrast, the minority of functions belonging to the cluster or outliers are typically also JumpReLU-like, except where the unmodified input latent activation is close to the point where the function 'jumps', so when we subtract an activation value of 1 from the input (as in Figures 8c and 10), this moves to the flat region where the output latent activation value is zero.

As we can see, the vast majority of these functions are either linear or JumpReLUs. Indeed, we verify this across the sample size of 10,000 functions and find that 88% are linear, 10% are JumpReLU (excl. linear, which is arguably a special case of JumpReLU), and only 2% are neither[3]. This result is encouraging – for a linear function, the first-order derivative is constant, so its value (i.e., the corresponding element of the Jacobian) completely expresses the relationship between the input and output values (up to a constant intercept). For the 88% of these scalar functions that are linear, the Jacobian thus accurately captures the notion of computational sparsity that interests us, rather than serving only as a proxy. And for the 10% of JumpReLUs, the Jacobians still perfectly measure the computational change we observe when changing the input latent within some subset of the input space.

While we expect the remaining 2% of scalar functions (Jacobian elements) to contribute only a small fraction of the computational structure of the underlying model, we preliminarily investigated their behavior. Figure 12 shows 12 randomly selected non-linear, non-JumpReLU $f_{s,(i,j)}|_{\mathbf{s_x}}$ functions. Even though these functions are nonlinear, they are still reasonably close to being linear, i.e., their first derivative is still predictive of the change we see throughout the input space. Indeed, most of them are on the diagonal line in Figure 10.

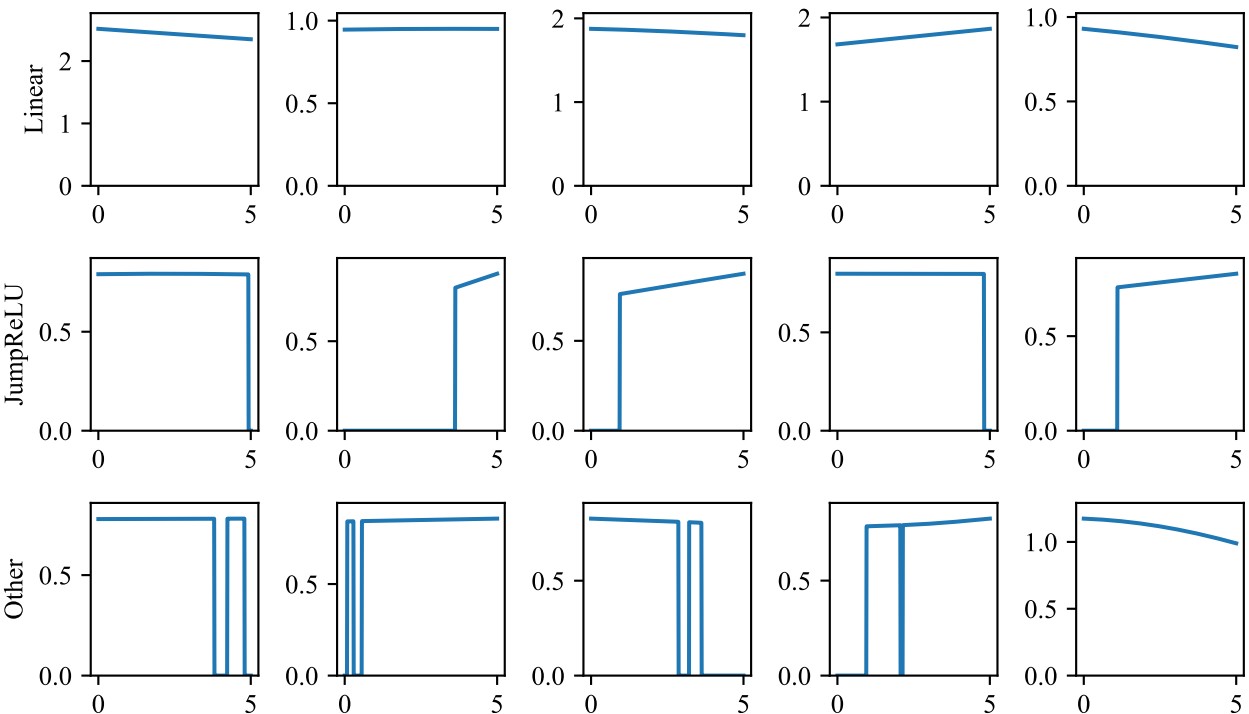

Figure 9. Additional examples of scalar functions between $s_{\mathrm{x},j}$ to $s_{\mathrm{y},i}$. The top row shows linear functions, the middle row shows JumpReLU functions, and the bottom row shows other functions. Recall that linear functions constitute a majority of the functions we observe empirically and that using JSAEs instead of traditional SAEs further increases the proportion of linear functions.

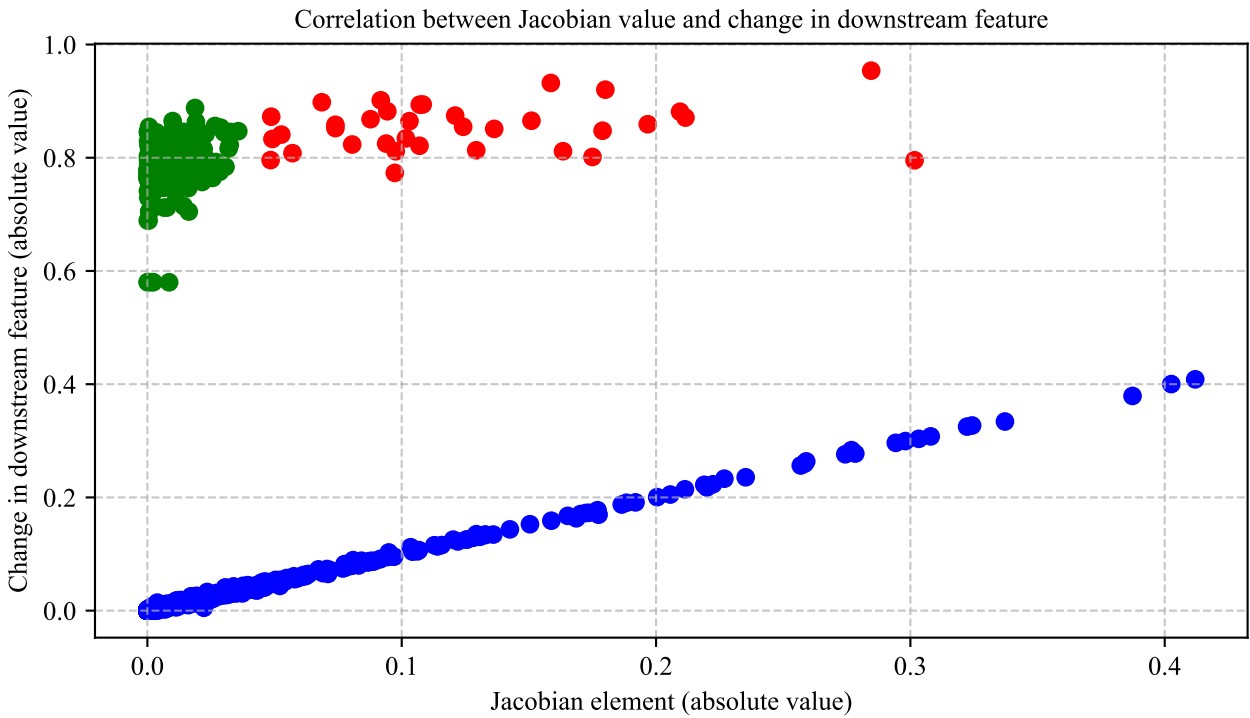

*Figure 10.* An expanded version of Figure 8c, measured on layer 3 of Pythia-70m. A scatter plot showing that values of Jacobian elements tend to be approximately equal to the change we see in the downstream feature when we modify the value of the upstream feature, namely when we subtract 1 from it. Each dot corresponds to an (input SAE latent, output SAE latent) pair. Unlike Figure 8c, this figure colors in the dots depending on which cluster they belong to – blue for "on the line", green for "in the cluster", red for "outlier". Additionally, this figure contains 10,000 samples (rather than 1,000 as in Figure 8c), which allows us to see more of the outliers and edge cases, though at the cost of visually obfuscating the fact that 97.5% of the samples are on the diagonal line, 2.1% are in the cluster, and 0.4% are outliers.

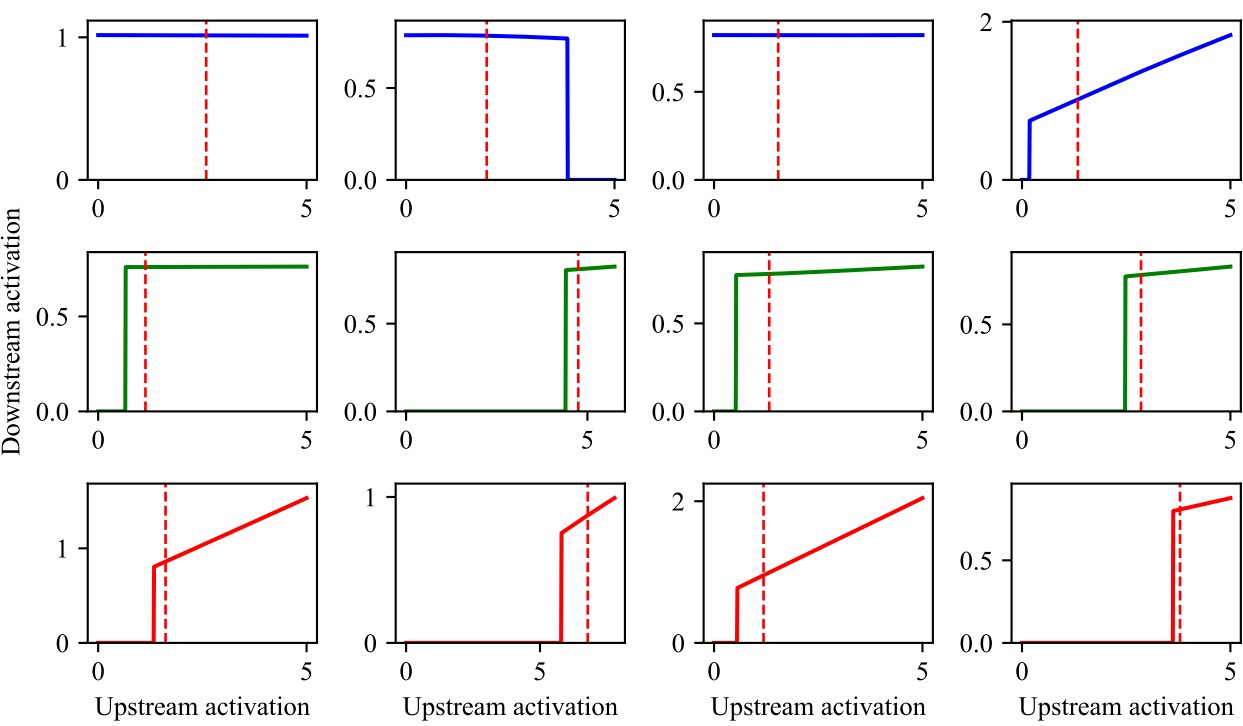

*Figure 11.* A handful of $f_{s,(i,j)}|_{\mathbf{s_x}}$ functions corresponding to the points in Figure 10. The color matches the group (and therefore the color) they were assigned in Figure 10. The red dashed vertical line denotes $\mathbf{s}_{\mathbf{x},i}^{(l)}$, i.e. the activation value of the SAE latent before we intervened on it. Note that the functions are not selected randomly but rather hand-selected to demonstrate the range of functions. We will quantitatively explore what proportion of $f_{s,(i,j)}|_{\mathbf{s_x}}$ functions have which structure in other figures.

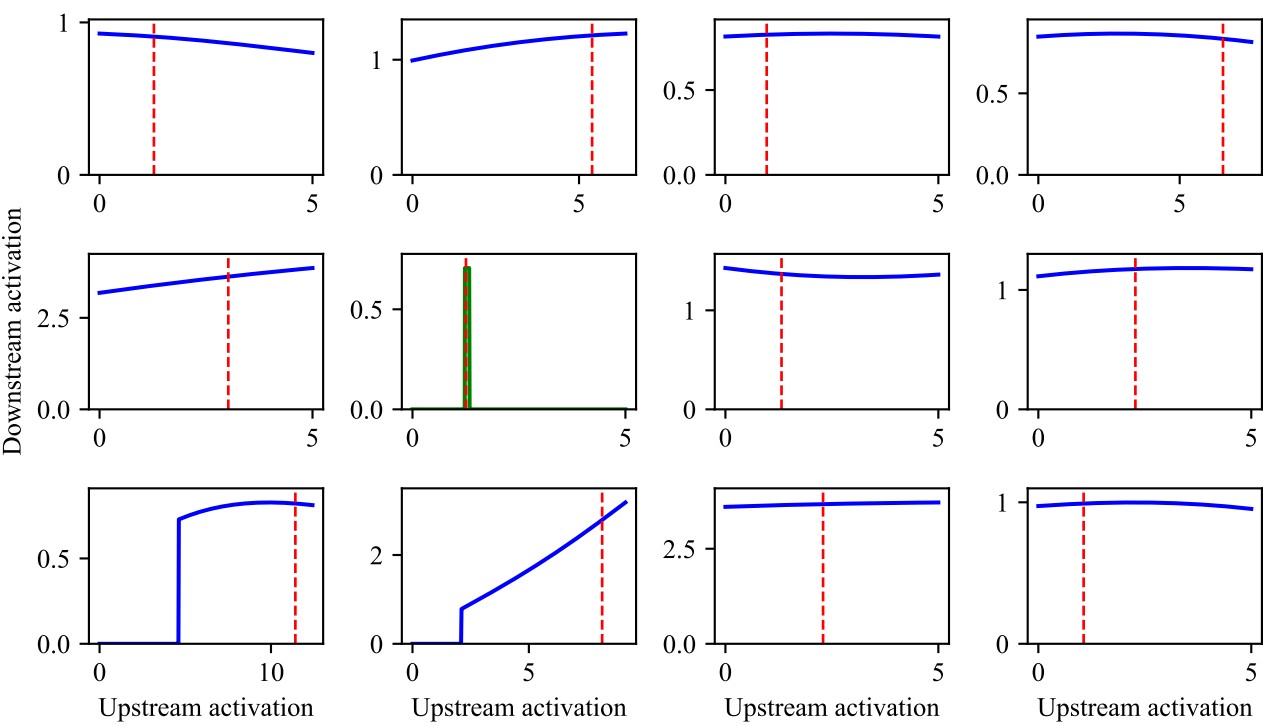

*Figure 12.* A random selection of the non-linear, non-JumpReLU $f_{s,(i,j)}|_{\mathbf{s_x}}$ functions. Note that non-linear, non-JumpReLU functions only constitute about 2% of $f_{s,(i,j)}|_{\mathbf{s_x}}$ functions. Even though these functions are clearly somewhat non-linear, their slope does still change quite slowly for the most part, which means that a first-order derivative at any point in the function is still reasonably predictive of the function's behavior in at least some portion of the input space (though there are some rare exceptions). The color again matches the group (and therefore the color) they were assigned in Figure 10; the red dashed vertical line denotes $\mathbf{s}_{\mathbf{x},i}^{(l)}$, i.e. the activation value of the SAE latent before we intervened on it.

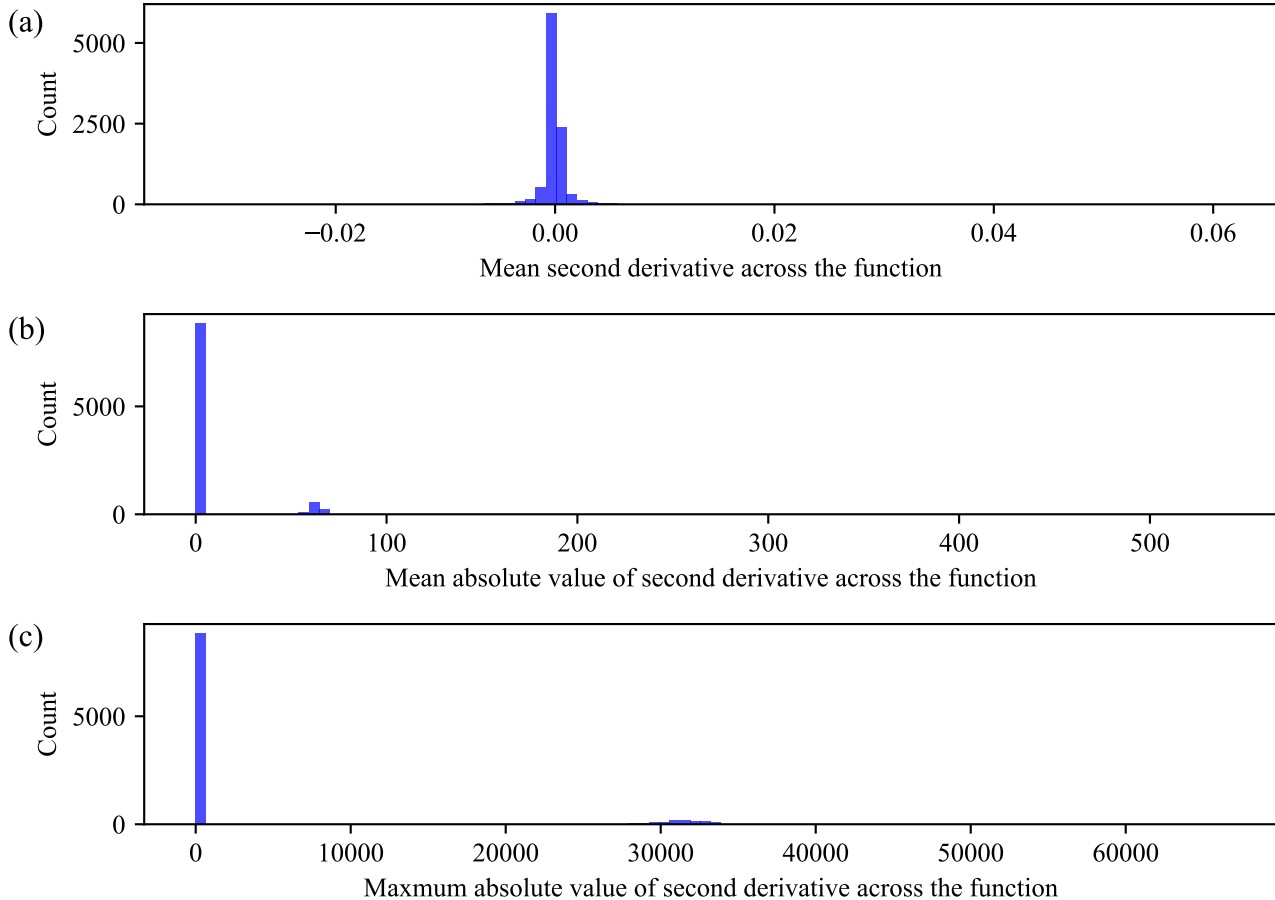

*Figure 13.* Distribution of second-order derivatives of functions $f_{s,(i,j)}|_{\mathbf{s_x}}$. Includes all functions, regardless of whether they are linear, JumpReLU, or neither. For a version that only includes non-linear, non-JumpReLU functions, see Figure 14. (a) The mean of the second-order derivative over the region of the input space. (b) The mean of the absolute value of the second-order derivative over the region of the input space. (c) The maximum value the second-order derivative takes in the region of the input space. Note that we are approximating the second derivative by looking at changes over a very small region (specifically 0.005), i.e., we do not take the limit as the size of this small region goes to zero; this is important because derivatives which would otherwise be undefined or infinite become finite with this approximation and therefore can be shown on the histograms. Also, we note that the means and maxima are taken over the region of the input space in which SAE features exist; see the footnote on page 14.

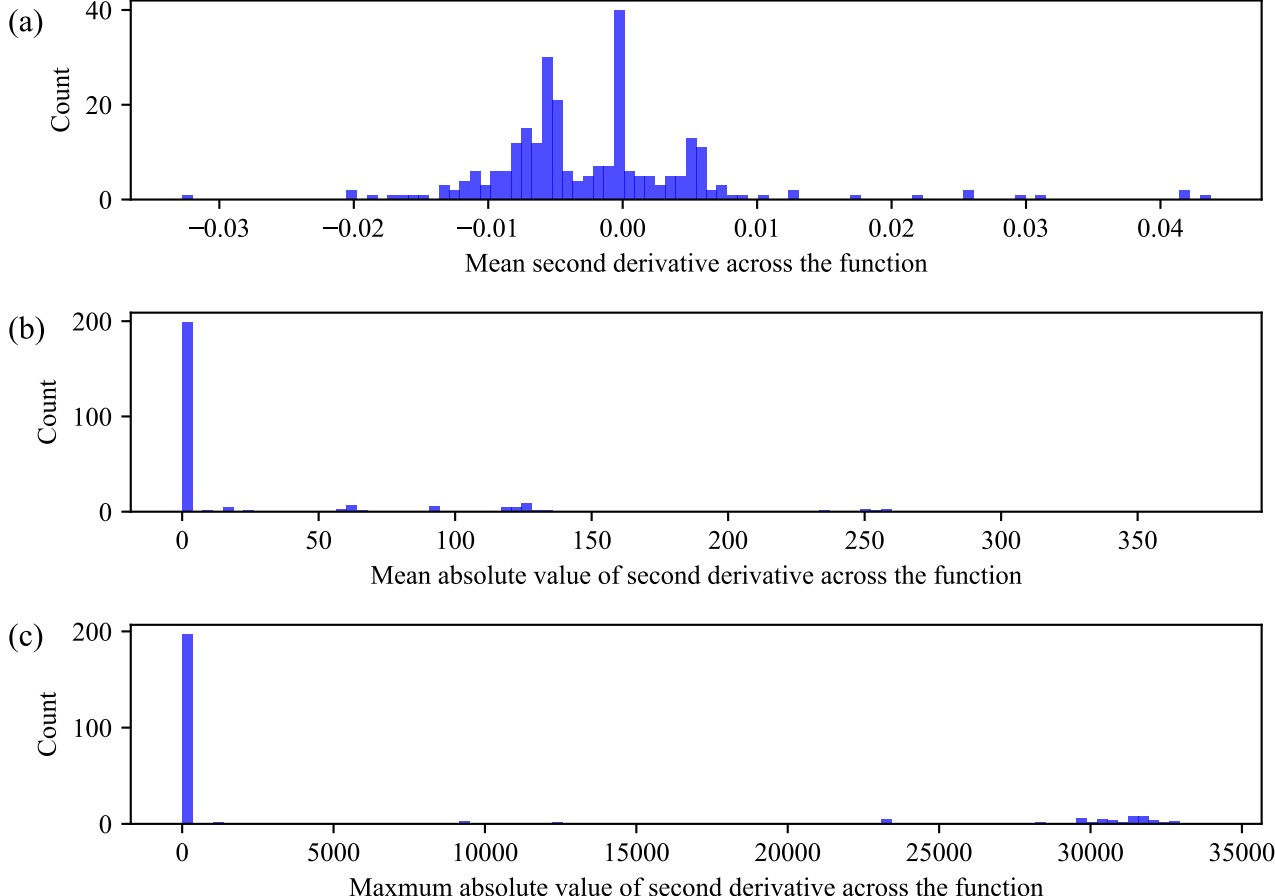

*Figure 14.* Distribution of second-order derivatives of functions $f_{s,(i,j)}|_{\mathbf{s_x}}$. Unlike Figure 13, this figure only includes the subset of the functions that are neither linear nor JumpReLU=like. (a) The mean of the second-order derivative over the region of the input space. (b) The mean of the absolute value of the second-order derivative over the region of the input space. (c) The maximum value the second-order derivative takes in the region of the input space. Note that we are approximating the second derivative by looking at changes over a very small region (specifically 0.005), i.e. we do not take the limit as the size of this small region goes to zero; this is important because derivatives which would otherwise be undefined or infinite become finite with this approximation and therefore can be shown on the histograms. Also, we note that the means and maxima are taken over the region of the input space in which SAE features exist; see the footnote on page 14.

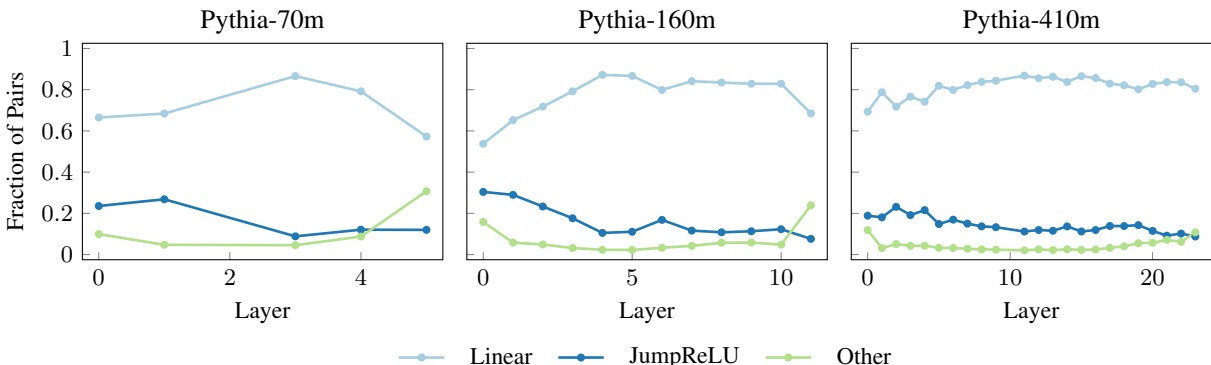

*Figure 15.* The fractions of Jacobian elements that exhibit a linear relationship between the input and output SAE latent activations, a JumpReLU-like relationship, and an uncategorized relationship, as described in Section 5.4. Here, we consider Jacobian SAEs trained on the feed-forward network at different layers of Pythia-70m, 160m, and 410m with fixed expansion factors $R = 64$ and $k = 32$. We computed the fractions over 1 million samples.

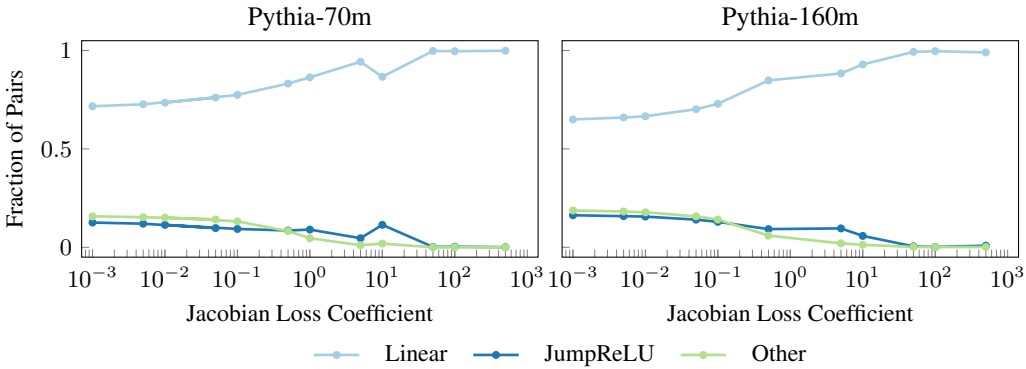

*Figure 16.* The fractions of Jacobian elements that exhibit a linear relationship between the input and output SAE latent activations, a JumpReLU-like relationship, and an uncategorized relationship, as described in Section 5.4. Here, we consider Jacobian SAEs trained on the feed-forward network at layer 3 of Pythia-70m (left) and layer 7 of Pythia-160m (right), with fixed expansion factors $R = 64$ and $k = 32$ and varying Jacobian loss coefficient (Section 4). We computed the fractions over 1 million samples.

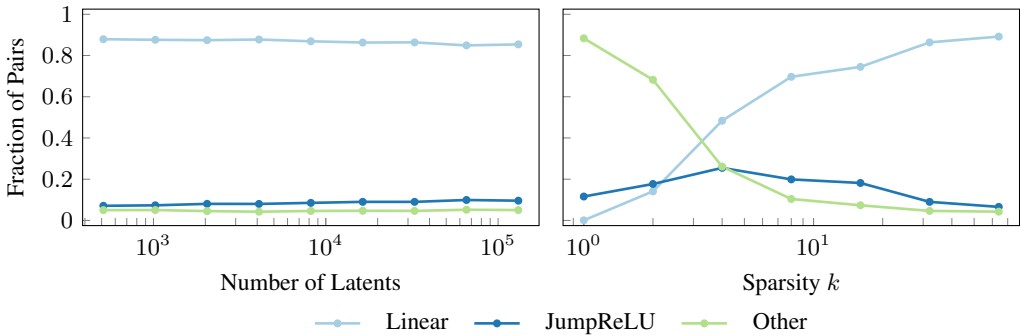

*Figure 17.* The fractions of Jacobian elements that exhibit a linear relationship between the input and output SAE latent activations, a JumpReLU-like relationship, and an uncategorized relationship, as described in Section 5.4. Here, we consider Jacobian SAEs trained on the feed-forward network at layer 3 of Pythia-70m with varying expansion factors (and hence numbers of latents; left) but fixed sparsities $k = 32$, and varying sparsities but fixed expansion factors $R = 64$ (Section 4). We computed the fractions over 1 million samples.

We can measure this more precisely by looking at the second-order derivative of $f_{s,(i,j)}|_{s_x}$. A zero second-order derivative across the whole domain would imply a linear function and, therefore, perfect predictive power of the Jacobian, while the larger the absolute value of the second-order derivative, the less predictive the Jacobian will be. This distribution is shown in Figure 13. The same distribution, which only includes the non-linear, non-JumpReLU functions, is shown in Figure 14. On average, the second derivative is extremely small for all features and effectively zero for the vast majority.

## C. Qualitative examples of the computations discovered by JSAEs

A common approach to interpreting LLM components like neurons or SAE latents is to collect token sequences and the corresponding activations over a text dataset (e.g., Yun et al., 2021; Bills et al., 2023). For example, the greatest latent activations may be retained, or activations from different quantiles of the distribution over the dataset (Bricken et al., 2023; Choi et al., 2024; Paulo et al., 2024).

We determined the set of 'top' output SAE latent indices by collecting the mean absolute values of non-zero Jacobian elements over a text dataset and sorting the output latents in descending order. Then, for each output latent, we found the input SAE latents that were most strongly connected to the output latent, again by sorting the input latents in descending order of the mean absolute value of non-zero Jacobian elements over the dataset. Finally, for both the output and input latents, we collected the individual latent activations over text samples with a context length of 16 tokens, retaining samples where at least one token produced a non-zero activation for the SAE latent. We chose a short context length to conveniently display the examples in a table format, and display here the top eight examples for each latent index, sorting the examples in descending order of the maximum latent activation over its tokens.

Each of the following figures comprises a table for a single output SAE latent (in pink), and a series of tables for the input latents with the greatest influences on the output latent, as determined by the mean absolute value of non-zero Jacobian elements. Conceptually, one may consider each figure as describing a single 'function', where the output and input latents represent the function output and inputs, respectively. Each table within the figure of examples displays a list of at most 12 examples, each comprising 16 tokens; we exclude the end-of-sentence token for brevity. The values of non-zero Jacobian elements and the activations of the corresponding input and output SAE latent indices are indicated by the opacity of the background color for each token. We take the opacity to be the element or activation divided by the maximum value over the dataset, i.e., all the examples with a non-zero Jacobian element for a given pair of input and output SAE latent indices. For clarity, we report the maximum element or activation alongside the colored tokens.

---

[3]Note that we are testing whether functions are linear or JumpReLUs only in the region of input space within which SAE activations exist. In particular, this means that we are excluding negative numbers. More specifically, the domain within which we test the function's structure is $[0, \max(5, s_{x,i}^{(l)} + 1)]$. In 92% of cases, $s_{x,i}^{(l)} + 1 < 5$; the median $s_{x,i}^{(l)}$ is 2.5.

| Example tokens | Max. activation |
|---|---|
| W enn Sie auf We iter k lic ken ist I hr Bon us ver | 5.596 |
| ro A14ber fl A14ss ig gew orden ist . Lau f zeit 36 Mon ate | 5.542 |
| es muss von einer aus lA nd ischen Bank sein . Log ischer we ise | 5.522 |
| . Sie kAnnen sich j eder zeit ab mel den . i . Click Here | 5.387 |
| bei der Be re it st ell ung unse rer D ien ste . \n | 5.315 |
| s icher lich sch on ge hA rt haben , kann es sehr ze | 5.275 |
| G eld aus . \n J ump No multiple accounts or free bonuses in a | 5.254 |
| we il be borrowing food from China , next .. Oh what a web we | 5.247 |
| k auf en will , kann ich tro tz dem das Gold V isa be | 5.225 |
| bef A14r wort ete auch die vol l stA nd ige K ontrol le A14ber | 5.223 |
| bonus codes 2019 , so dass ein We g in ein W ett b A14 | 5.208 |
| re Imm ob il ie in Sp an ien auf ih ren N amen beh | 5.207 |

(a) The top 12 examples that produce the maximum latent activations for the output SAE latent with index 34455.

| Example tokens | Max. activation |
|---|---|
| dam it ein ver stand en , dass wir Cookies ver w enden . Die | $1.292 \times 10^1$ |
| we isen , dass Sie nicht mehr als 11 Mon ate a uss er hal | $1.229 \times 10^1$ |
| st super auf me ine W A14n sche e ing eg ang en . Ins | $1.173 \times 10^1$ |
| Golden V isa ), die gle ich zeit ig er la ub t , dass | $1.167 \times 10^1$ |
| f A14r das Gold V isa in 2013 er lass en . Zu dies em | $1.152 \times 10^1$ |
| W enn Sie auf We iter k lic ken ist I hr Bon us ver | $1.139 \times 10^1$ |
| age : W enn ich die Imm ob il ie durch eine Ges ells chaft | $1.134 \times 10^1$ |
| ung abs ch icken d auer b ren ner zur A14ck le hn en : | $1.132 \times 10^1$ |
| lich die ID K arte ver l Ang ert werden , das sehr s chn | $1.126 \times 10^1$ |
| te " \n " A uf alle W A14n sche e ing eg ang en | $1.125 \times 10^1$ |
| keine oder an le icht er f A14 ll bare B eding ungen g ek | $1.124 \times 10^1$ |
| hat Sp an ien ein Ges etz er lass en , dass einem N icht | $1.116 \times 10^1$ |

(b) The top 12 examples that produce the maximum latent activations for the input SAE latent with index 39503. Paired with the output SAE latent with index 34455, the Jacobian element is non-zero for 15130 tokens, and has a mean of $3.966 \times 10^{-1}$ (rank 0 for the output SAE latent) and a standard deviation of $7.743 \times 10^{-2}$ over its non-zero values.

| Example tokens | Max. activation |
|---|---|
| ZA14 rich St utt gart Le ipzig / H alle | 6.834 |
| OTC cad ets , students , and staff of Ans bach Middle High School | 6.788 |
| Out reach announced that two groups of students from Ans bach Middle High School won | 6.693 |
| ride from Zur ich to Rh ine Falls ( with a small section of our | 6.679 |
| aud . U 55 Brand en bur ger Tor Bus 200 00 U hr die | 6.644 |
| at the Ans bach Middle High School track . \n G ram my award winning | 6.566 |
| , Ans bach , Germany , on May 18 , 2015 . Fry e was | 6.546 |
| parents to attend . \n W ies b aden students find math a little fish | 6.480 |
| into the Gi essen b acht al valley . On the trail along the edge | 6.471 |
| was the first case for the b MC Kob len z office . Capt . | 6.448 |
| ol ie . Follow the signs Brand en bur ger Tor Brand enburg Gate or | 6.423 |
| for the Kob len z office . \n b MC ' s Captain Dennis Brand | 6.391 |

(c) The top 12 examples that produce the maximum latent activations for the input SAE latent with index 3387. Paired with the output SAE latent with index 34455, the Jacobian element is non-zero for 10355 tokens, and has a mean of $3.437 \times 10^{-1}$ (rank 1 for the output SAE latent) and a standard deviation of $2.873 \times 10^{-2}$ over its non-zero values.

| Example tokens | Max. activation |
|---|---|
| , Dr Alex German and his team found the main reason for that is the | $1.149 \times 10^1$ |
| for Nuclear Physics , R up recht Kar ls University , the German Cancer Research | $1.142 \times 10^1$ |
| , Chris Kend rick , Sally Anne F itter and Penny German . \n More | $1.137 \times 10^1$ |
| 1 . 2 in Czech oslov akia , 1 . 1 in the German Democratic | $1.119 \times 10^1$ |
| mind , intellect "), from Pro to − German ic ∗ mund iz , ∗ | $1.089 \times 10^1$ |
| and Hyde F li ppo ' s Aut ob ahn articles from The German Way and | $1.084 \times 10^1$ |
| " L ili Mar lene ", the song that German and American soldiers both loved | $1.075 \times 10^1$ |
| ls − Univers itАt , the German Cancer Research Center ( DK FZ ), | $1.070 \times 10^1$ |
| . In addition , Dr German ' s team also confirmed that cats ate about | $1.068 \times 10^1$ |
| Art . 3 of the German Constitution that addresses equality before the law . \n | $1.064 \times 10^1$ |
| under constant international external laboratory control of the German acc reditation system In stand . | $1.052 \times 10^1$ |
| But the history of carn ival can be traced back to German ic tribes who | $1.051 \times 10^1$ |

(d) The top 12 examples that produce the maximum latent activations for the input SAE latent with index 41811. Paired with the output SAE latent with index 34455, the Jacobian element is non-zero for 764 tokens, and has a mean of $1.619 \times 10^{-1}$ (rank 2 for the output SAE latent) and a standard deviation of $8.654 \times 10^{-3}$ over its non-zero values.

| Example tokens | Max. activation |
|---|---|
| National Archives report " Hit ler ' s Shadow : Nazi War Crim inals , | $1.203 \times 10^1$ |
| Th ousands of Holocaust victims transported to Nazi concentration camps by a French railway company | $1.195 \times 10^1$ |
| Air Corps after WW II ( where he survived a year as a Nazi prisoner | $1.170 \times 10^1$ |
| a name inspired by the chief Ukrainian Nazi leader , Ste pan Band era ) | $1.168 \times 10^1$ |
| Powers refused cooperation with the company until connections with Nazi Germany were severed . \n | $1.155 \times 10^1$ |
| orrow , the World ! ," from 1944 , about a teen − age Nazi | $1.145 \times 10^1$ |
| , 000 French Jews to Nazi concentration camps , though experts disagree on its degree | $1.145 \times 10^1$ |
| of the Jewish people during the Nazi era it was the work of white Europeans | $1.118 \times 10^1$ |
| computing pioneer , and his development of a system to crack Nazi codes in order | $1.099 \times 10^1$ |
| it had no effective control over operations during the Nazi occupation from 1940 to 1944 | $1.090 \times 10^1$ |
| it with heavy water + Nazi chemicals , prayed to god for assistance , struck | $1.079 \times 10^1$ |
| symbol of Russian resistance to Nazi invasion . Russia ' s " Window on the West | $1.067 \times 10^1$ |

(e) The top 12 examples that produce the maximum latent activations for the input SAE latent with index 32619. Paired with the output SAE latent with index 34455, the Jacobian element is non-zero for 316 tokens, and has a mean of $1.518 \times 10^{-1}$ (rank 3 for the output SAE latent) and a standard deviation of $1.193 \times 10^{-2}$ over its non-zero values.

| Example tokens | Max. activation |
|---|---|
| owned by the von W aken itz family in 14 34 . The same family | $1.605 \times 10^1$ |
| World Nuclear Industry Status Report 2013 , July . \n 8 David von Hipp el | $1.526 \times 10^1$ |
| as Mart a , one of the singing von Tra pp children . She i12 | $1.509 \times 10^1$ |
| knockdown kiss is administered when Von locks lips with GG way more enthusi astically than | $1.437 \times 10^1$ |
| , I assume . Gre ta Gar bo is here with Von , and not | $1.427 \times 10^1$ |
| HATE THAT I DON ' T HATE YOU " \n Fel ix von | $1.423 \times 10^1$ |
| was already published 150 years ago by the german physician Herm ann von Hel | $1.388 \times 10^1$ |
| new life by Edward von LA ng us and augmented reality technologies , exploring | $1.377 \times 10^1$ |
| ( 19 28 , Joseph von Stern berg ; silent ) . A T ribute to | $1.366 \times 10^1$ |
| ily Von has again been sight ed in that infamous suit . Was Metro econom | $1.352 \times 10^1$ |
| artists and performers , known as the Castle von Tra pp , will im min | $1.341 \times 10^1$ |
| ung . D egen feld − F ests chr ift , Vienna : von Lag | $1.320 \times 10^1$ |

(f) The top 12 examples that produce the maximum latent activations for the input SAE latent with index 63157. Paired with the output SAE latent with index 34455, the Jacobian element is non-zero for 84 tokens, and has a mean of $1.479 \times 10^{-1}$ (rank 4 for the output SAE latent) and a standard deviation of $8.717 \times 10^{-3}$ over its non-zero values.

| Example tokens | Max. activation |
|---|---|
| Dum ont a G John Ax el rod , MS NBC al um now at Berlin | $1.040 \times 10^1$ |
| for determining adhesive and tensile strengths ( w inner of the Berlin – Br anden | $1.026 \times 10^1$ |
| That Dem on Within was selected for the 64 th Berlin International Film Festival Pan | $1.025 \times 10^1$ |
| M irt 1 , and M . Sch mid , eds . Berlin : Springer | $1.017 \times 10^1$ |
| processes in tropical forests ( pp . 153 – 172 ). Springer , Berlin , | $1.011 \times 10^1$ |
| Twenty – five years after the fall of the Berlin wall in 1989 , Cont | $1.008 \times 10^1$ |
| the Berlin Wall : legacy SA– 2 , SA– 3 , SA– | 9.963 |
| emerged after the collapse of the Berlin Wall and the end of the cold war | 9.932 |
| fall of the Berlin Wall was a great spur to Germany , though it took | 9.898 |
| exhibitions in museums and galleries in Cape Town , Johannes burg , Berlin and C | 9.885 |
| of the following options for the Berlin to Sch ild ow route : Michel in | 9.885 |
| t have to get a new one ( with the Berlin citizen center , it | 9.847 |

(g) The top 12 examples that produce the maximum latent activations for the input SAE latent with index 63657. Paired with the output SAE latent with index 34455, the Jacobian element is non-zero for 195 tokens, and has a mean of $1.387 \times 10^{-1}$ (rank 5 for the output SAE latent) and a standard deviation of $6.091 \times 10^{-3}$ over its non-zero values.

| Example tokens | Max. activation |
|---|---|
| no zzles , are known from the Mercedes G LA . In addition , Mercedes | 3.361 |
| an open end – hole . \n Moreover , when cathe ters of this type | 2.830 |
| N asm ith . machine rollers for conveyor belt The invention relates , inter alia | 2.829 |
| cathe ters which have a closed distal end . The principles of the invention therefore | 2.757 |
| are also suitable . \n If it is possible for you to get your blood | 2.487 |
| analog matching devices can be used advantage ously for and monitoring and diagnostics of process | 2.399 |
| be the case that such support can be secured , for example , in cases | 2.249 |
| ic acid . Pro per ventilation and her met ically sealed production apparatus are therefore | 2.240 |
| of residence , in particular copyright , data protection and competition law . The provider | 2.234 |
| products of this type work great for increasing your store ' s revenue , as they | 2.133 |
| action by the customer or from errors in the information provided by the latter . | 2.113 |
| optionally a rear axle differential lock . \n In contrast to the prospective main competitor | 2.099 |

(h) The top 12 examples that produce the maximum latent activations for the input SAE latent with index 7969. Paired with the output SAE latent with index 34455, the Jacobian element is non-zero for 331 tokens, and has a mean of $1.322 \times 10^{-1}$ (rank 6 for the output SAE latent) and a standard deviation of $1.176 \times 10^{-2}$ over its non-zero values.

| Example tokens | Max. activation |
|---|---|
| and a private one . With a public key , it A~ s possible to | 5.947 |
| Cons olid ation 25 92 . In the last years of its active life , | 5.068 |
| forward the realization of artistic works linked to this whole context , taking into account | 4.883 |
| test different runtime environments . As a consequence , cloud− based tool ing currently | 4.585 |
| release I recommend to have a look at J SF− Spring ( http :// | 4.558 |
| customer satisfaction . \nWhat is more , with the app , it 's possible | 4.501 |
| of the hotel . On the hotel land it is possible to camp in tents | 4.438 |
| introduce additional modifications do not forget about re− publ ishing the whole site so | 4.426 |
| change is requested , this tool enables to know which other requirements , design and | 4.418 |
| structures / steel constructions , according to the highest requirements . \nDepending on | 4.370 |
| 2 O 3 , or , more precisely , a mixture of NO and NO | 4.309 |
| if you have any questions ( you can also have a look at our website | 4.272 |

(i) The top 12 examples that produce the maximum latent activations for the input SAE latent with index 18964. Paired with the output SAE latent with index 34455, the Jacobian element is non-zero for 3502 tokens, and has a mean of $1.209 \times 10^{-1}$ (rank 7 for the output SAE latent) and a standard deviation of $2.073 \times 10^{-2}$ over its non-zero values.

| Example tokens | Max. activation |
|---|---|
| will enjoy music inspired in or by America , France , Austria , Scotland , | $1.170 \times 10^1$ |
| in the world since its adoption by the Austrian army in 1977 . Now available | $1.116 \times 10^1$ |
| , Australia , China , Austria , England , Israel and Ireland . \nThrough | $1.112 \times 10^1$ |
| teaching the basics of K ung Fu in Russia , Austria , Spain , China | $1.053 \times 10^1$ |
| occupied areas in 1848 ; by the 1880 s , the Austrian − govern ed | $1.042 \times 10^1$ |
| Russia , and American Ambassador to Austria , William Each o , we had the | $1.033 \times 10^1$ |
| , which traces its roots to the work of the Cold War Austrian − American | $1.016 \times 10^1$ |
| released in Germany , Austria and Switzerland . \n Irish man in America was recorded | $1.014 \times 10^1$ |
| W ahl led to an invitation by the Austrian Ministry of Culture , for myself | 9.958 |
| of State from Austria , Poland , and Hungary , Min isters from Israel and | 9.951 |
| practice first abroad , in countries such as Spain , Germany or Austria , where | 9.771 |
| . \nThe Che chen youth , who came to Austria as a refugee , | 9.769 |

(j) The top 12 examples that produce the maximum latent activations for the input SAE latent with index 28112. Paired with the output SAE latent with index 34455, the Jacobian element is non-zero for 200 tokens, and has a mean of $1.156 \times 10^{-1}$ (rank 8 for the output SAE latent) and a standard deviation of $8.429 \times 10^{-3}$ over its non-zero values.

| Example tokens | Max. activation |
|---|---|
| Reynolds & Jake Gy ll en ha al Ar rives ! \n Earlier tonight during | 9.162 |
| Gy ll en ha al , Ryan Reynolds , and Rebecca Ferguson star in the | 8.938 |
| Jim Mc El re ath , Mario And ret ti , Gary Bet ten haus | 8.771 |
| M . ; Sp icken he uer , A . ; W agen f A14 | 8.518 |
| followed by Nick H ohl be in who finished fourth . \n If Friday night | 8.185 |
| aud . U 55 Brand en bur ger Tor Bus 200 00 U hr die | 8.145 |
| the ritual He iden j acht en are not really a closed chapter of history | 8.071 |
| the husband to the late Mary Jane Hol tz cl aw . Sc oop was | 8.004 |
| aG Chris Har low ... Eric F ink be iner ... Mike Deutsch , FAA | 7.912 |
| ding Music Pro ducer and Artist LA EL , Jeff Sch nee we is , | 7.688 |
| team is only hitting . 242 as a team . Bob Stein b ock joined | 7.678 |
| Red d , Tina Den ise L oll is , Timothy Ray Hol tz cl | 7.668 |

(k) The top 12 examples that produce the maximum latent activations for the input SAE latent with index 4287. Paired with the output SAE latent with index 34455, the Jacobian element is non-zero for 1820 tokens, and has a mean of $1.131 \times 10^{-1}$ (rank 9 for the output SAE latent) and a standard deviation of $2.500 \times 10^{-2}$ over its non-zero values.

| Example tokens | Max. activation |
|---|---|
| Mah avir Mer i Dr is hti Me in . One afternoon he was resting | $1.030 \times 10^1$ |
| haben . \n Un se rer Me in ung nach wird es in den n A | 9.290 |
| waukee Magazine prof iled Le in bach and the U EC last year . \n | 8.524 |
| about Milwaukee ' s Urban Ec ology Center ? \n Ken Le in bach , | 7.583 |
| DO KOPY AR BH ARE AL FA ZO me in bol new | 7.505 |
| and song writing team of P ia Le in onen and Jon i T ial | 7.337 |
| on . April 27 , 2015 . By . Dom Ein horn . Cos m | 6.740 |
| term growth and sustainability . Luc as And S hel by . Ein Smith . | 6.319 |
| – top casino sites worldwide Das V ere in ig te K An ig reich | 6.226 |
| z ond heid / vo eding – met – we in ig – cal orie | 6.008 |
| inated A . C . Ke in m ies as their candidate . \n ment | 5.644 |
| my heart to live . \n D ilon me in tum ap ni bet abi | 5.064 |

(l) The top 12 examples that produce the maximum latent activations for the input SAE latent with index 62769. Paired with the output SAE latent with index 34455, the Jacobian element is non-zero for 48 tokens, and has a mean of $1.121 \times 10^{-1}$ (rank 10 for the output SAE latent) and a standard deviation of $7.225 \times 10^{-3}$ over its non-zero values.

| Example tokens | Max. activation |
|---|---|
| op 316 ( 18 67 ) . Vienna PO / Carl os Kle iber . \n | $1.541 \times 10^1$ |
| , J ape , K rystal Kle ar , K orm ac ' s Big | $1.460 \times 10^1$ |
| , Jordan Kle eman , goes over some of the upcoming features coming to the | $1.445 \times 10^1$ |
| Golden Wings Music . Following his early beginnings on E el ke Kle ijn ' | $1.431 \times 10^1$ |
| ap enem resistant Pseud omonas aeruginosa ( CRPA ), Kle bs iella pneumoniae ( | $1.373 \times 10^1$ |
| drink . I bring my Kle an K ante en with me everywhere and use | $1.336 \times 10^1$ |
| ina G arr ig ues ); 2009– Barcelona ( w / K le pac | $1.293 \times 10^1$ |
| jin ( w / K le pac ); 2010– Budapest ( w / Med | $1.263 \times 10^1$ |
| . \n Dog poison No . 2 : In sect icides . \n F le | 5.116 |
| there and no one was boarding yet . We parked the car , sch le | 4.971 |
| mark the perm al ink . \n F le as or even worse , we | 4.956 |
| left . \n G le be Rd South to Route 1 . Turn right onto | 4.754 |

(m) The top 12 examples that produce the maximum latent activations for the input SAE latent with index 14871. Paired with the output SAE latent with index 34455, the Jacobian element is non-zero for 30 tokens, and has a mean of $1.095 \times 10^{-1}$ (rank 11 for the output SAE latent) and a standard deviation of $9.772 \times 10^{-3}$ over its non-zero values.

| Example tokens | Max. activation |
|---|---|
| Lucia Rodriguez ( Com cast ) as Chair– Elect ; Emma Pf ister ( | $1.913 \times 10^1$ |
| another clever design from Pf ister , but it ' s really hard to set | $1.897 \times 10^1$ |
| cast Administration from Boston University . \n Emma Pf ister ( Tre asure r | $1.854 \times 10^1$ |
| t ips / my college application essay one of my jobs at Pf izer since | $1.817 \times 10^1$ |
| , Pf izer , and Comp umed ics Limited , among others . \n Profile | $1.781 \times 10^1$ |
| . Pf ann kuch and M . O . J . Thomas , eds | $1.711 \times 10^1$ |
| case in Pf le ide rer and IV G Immob il ien . The | $1.685 \times 10^1$ |
| am ore ; Ronald Bra ut ig am , p f ; Leo van Does | 7.684 |
| Pen elope Th wait es , p f . \n M oz art , W | 6.898 |
| ). Yu ja Wang , p f . \n R ach man in ov , | 6.714 |
| ino , p f . \n Resp igh i , O . Belk is | 6.594 |
| Pre vin , p f . \n B oc cher ini , L . String | 6.460 |

(n) The top 12 examples that produce the maximum latent activations for the input SAE latent with index 32693. Paired with the output SAE latent with index 34455, the Jacobian element is non-zero for 24 tokens, and has a mean of $1.092 \times 10^{-1}$ (rank 12 for the output SAE latent) and a standard deviation of $9.177 \times 10^{-3}$ over its non-zero values.

| Example tokens | Max. activation |
|---|---|
| A . A . planned , coordinated and sch m oo zed for four months | $2.058 \times 10^1$ |
| there and no one was boarding yet . We parked the car , sch le | $2.045 \times 10^1$ |
| intervention . For most of us poor sch mucks it ' s just something | $2.004 \times 10^1$ |
| – to– speech . Now print out a copy of this page , sch | $1.945 \times 10^1$ |
| st inky melting stage where you can just sch m ear it on crust y | $1.934 \times 10^1$ |
| Paul VI for he res y , sch ism and scandal ten years ago , | $1.926 \times 10^1$ |
| slate , sch ist , quartz ite , and limestone on the west ; met | $1.909 \times 10^1$ |
| he res y , sch ism and scandal , it is in fact more grave | $1.893 \times 10^1$ |
| is needed to sch irm ish on a 3 area front . USSR should in | $1.881 \times 10^1$ |
| as what the French call changes to the " sch ol astic rhythms ." \n | $1.877 \times 10^1$ |
| ' s sch tick was new and the moment was right ," and that we | $1.876 \times 10^1$ |
| ac ris , ro bin sch ul z , will . i . am , | $1.849 \times 10^1$ |

(o) The top 12 examples that produce the maximum latent activations for the input SAE latent with index 30568. Paired with the output SAE latent with index 34455, the Jacobian element is non-zero for 449 tokens, and has a mean of $1.049 \times 10^{-1}$ (rank 13 for the output SAE latent) and a standard deviation of $1.109 \times 10^{-2}$ over its non-zero values.

| Example tokens | Max. activation |
|---|---|
| 31 . \n Ra ich JW , Sch les inger WH , 1992 . | $1.313 \times 10^1$ |
| Health and Care , the Sch les wig– Hol stein Ministry for Social Affairs | $1.276 \times 10^1$ |
| Productions celebrated Justin Sch leg el ' s birthday by presenting him with a very special | $1.274 \times 10^1$ |
| from Politics and Society to Culture and Entertainment . Ber lin : Sch les inger | $1.262 \times 10^1$ |
| irling and mixing and clear mixing lines visible in it ( Sch lie ren as | $1.247 \times 10^1$ |
| K yle Sch war ber may be out for the season , but the home | $1.231 \times 10^1$ |
| a Sab ot age . Arnold Sch war zen eg ger is back and looking | $1.229 \times 10^1$ |
| 21 ) but Sch les inger ' s pla te number suggests a earlier publication date | $1.224 \times 10^1$ |
| the ' one field ' target . \n Just about one field by Sch war | $1.222 \times 10^1$ |
| ville , Texas , Sch re iner University is a small four year private college | $1.218 \times 10^1$ |
| ) spoke at an event of B DO 10 . Sch if ff ah | $1.211 \times 10^1$ |
| , Mike Sch re iber and countless more . She volunteers at D ade Correction | $1.203 \times 10^1$ |

(p) The top 12 examples that produce the maximum latent activations for the input SAE latent with index 47756. Paired with the output SAE latent with index 34455, the Jacobian element is non-zero for 451 tokens, and has a mean of $1.040 \times 10^{-1}$ (rank 14 for the output SAE latent) and a standard deviation of $1.167 \times 10^{-2}$ over its non-zero values.

| Example tokens | Max. activation |
|---|---|
| li L ai a~K34 al~ aIJi e~ i aIJI a12l aIJI c12 al | $1.484 \times 10^1$ |
| e~N , aa ei ah l eoaL aI cI e~ | $1.478 \times 10^1$ |
| L ij ao a34I a1 l j aI e34 34 . e~ | $1.422 \times 10^1$ |
| a N cl a i al a14 eGl a~~ 1~ to be , ei cK~ a | $1.419 \times 10^1$ |
| aH a I eL cl e~ aa1 a~ e~~ IIJ : I suspect | $1.385 \times 10^1$ |
| : content cK~ a12l aIJi e~ i , e~ co cL a12k , | $1.385 \times 10^1$ |
| 1~ e~ e~ N a12 a1 e~ i , iG aa G eI | $1.379 \times 10^1$ |
| Ii aoi k ). \n aL I a aL l Ig eg e La | $1.378 \times 10^1$ |
| co aI a1 L , ei e~ co I 12 e a1 i | $1.374 \times 10^1$ |
| aI a La I e ao ILH i 14I 2J li eg 12 | $1.374 \times 10^1$ |
| the contents page ( cL a12k e ). a a , e~ | $1.368 \times 10^1$ |
| aIJi e~ i aii a12l al e~ N a1 1~ aHN , aH : | $1.362 \times 10^1$ |

(q) The top 12 examples that produce the maximum latent activations for the input SAE latent with index 59459. Paired with the output SAE latent with index 34455, the Jacobian element is non-zero for 1602 tokens, and has a mean of $1.036 \times 10^{-1}$ (rank 15 for the output SAE latent) and a standard deviation of $1.771 \times 10^{-2}$ over its non-zero values.

*Figure 18.* The top 12 examples that produce the maximum latent activations for the output SAE latent with index 34455, and the input SAE latents with which the mean values of the corresponding Jacobian elements are greatest. The Jacobian SAE pair was trained on layer 15 of Pythia-410m with an expansion factor of $R = 64$ and sparsity $k = 32$. The examples were collected over the first 10K records of the English subset of the C4 text dataset with a context length of 16 tokens.

| Example tokens | Max. activation |
|---|---|
| ke vAl retro f Ar g ade f Arden kA ns lig e . Barn | 4.636 |
| v lig ut s ikt Aver Lip nos j An . Mys ig och barn | 4.549 |
| och f Arf Ar st Ar k are , upp till 20 tim mar ut an | 4.370 |
| kar , men du mAste vara in log g ad som an st Al ld | 4.337 |
| Det kan fin nas fl er pro jek t dAr Lars Bank vall med ver | 4.165 |
| C ancel . E tt ex emp el pA fore x hand el Ar att | 4.149 |
| Sp es ial sp ill ut val g av m ors om me og | 4.121 |
| atal ogen under Download s i men yn dAr du en k elt bl A | 4.080 |
| A till A ! Cong rat z Eva Ny strAm and Ad riel Young ! | 4.037 |
| kA pa Euro och s Al ja US Dollar . \n Results per trade binary | 3.989 |
| v An lig at mos f Ar och sk A na rum , dock kans | 3.985 |
| i sm ak pA det som ble ig jen . \n E arn ed the | 3.977 |

(a) The top 12 examples that produce the maximum latent activations for the output SAE latent with index 64386.

| Example tokens | Max. activation |
|---|---|
| Sp es ial sp ill ut val g av m ors om me og | $1.218 \times 10^1$ |
| ass el og and en dans ke kon ge af den Gl 252 ; cks | $1.175 \times 10^1$ |
| i sm ak pA det som ble ig jen . \n E arn ed the | $1.137 \times 10^1$ |
| var kon ge af Dan mark fra 1906 til 1912 . Han var det 230 | $1.136 \times 10^1$ |
| v lig ut s ikt Aver Lip nos j An . Mys ig och barn | $1.018 \times 10^1$ |
| ke vAl retro f Ar g ade f Arden kA ns lig e . Barn | $1.007 \times 10^1$ |
| Det kan fin nas fl er pro jek t dAr Lars Bank vall med ver | 9.832 |
| v An lig at mos f Ar och sk A na rum , dock kans | 9.703 |
| . Det g l der de fl este v arer , her under l sk | 9.656 |
| LA~ ks et ind en 12 42 m 10 69 m 119 1 m | 9.561 |
| pA Chal mers f Ar att kun na se dem . Sometimes I like to add | 9.513 |
| jeg g ings , stock ings well anything that covers my legs , is | 9.509 |

(b) The top 12 examples that produce the maximum latent activations for the input SAE latent with index 23581. Paired with the output SAE latent with index 64386, the Jacobian element is non-zero for 2755 tokens, and has a mean of $2.578 \times 10^{-1}$ (rank 0 for the output SAE latent) and a standard deviation of $2.428 \times 10^{-2}$ over its non-zero values.

| Example tokens | Max. activation |
| --- | --- |
| . \n400 Pet erm ans Bj erg 2940 m 1200 m 2940 | 7.399 |
| from Sig urd . I think we were lied to . Don ' t you have | 7.343 |
| d " was invented by veter in arian Sig urd Ke il gaard in the | 6.987 |
| us Met z Ped ers en the film stars S ver rir Gud n | 6.918 |
| began working as an artist and teacher . \n In g ib j Ar g | 6.917 |
| ALE V intage Metal Ch airs From B ali By Bj A r k heim With | 6.837 |
| house , making Read Also : In grid N ils en Bio , D ating | 6.760 |
| film shot in Trin idad in 2013 , work by A y v ind F | 6.647 |
| og R oms dal , in Norway . \n484 Lav ang st ind en | 6.632 |
| K j ell ! \n I intended the first piece to basically be an introduction | 6.463 |
| world around me . Cong ratulations to Tor stein H org mo , the latest | 6.459 |
| and modern technology . \n As Tar je N issen − M eyer writes : | 6.413 |

(c) The top 12 examples that produce the maximum latent activations for the input SAE latent with index 48028. Paired with the output SAE latent with index 64386, the Jacobian element is non-zero for 3567 tokens, and has a mean of $2.417 \times 10^{-1}$ (rank 1 for the output SAE latent) and a standard deviation of $1.944 \times 10^{-2}$ over its non-zero values.

| Example tokens | Max. activation |
| --- | --- |
| ., Netherlands , Italy , Canada , Denmark , Norway , and Turkey ) and | 9.613 |
| EU , Germany , Norway , US and UK ) . \n PROM AN has been | 9.009 |
| Alaska and the Ale ut ian Islands , Green land , Britain , Norway , | 8.417 |
| ), Russia , Norway and Iceland which promotes dialogue , practical cooperation and development . | 8.119 |
| Sah ara . So have human rights bodies from Norway and elsewhere . \n The | 8.066 |
| the matern ity leave in Norway ( 46 weeks ) , Denmark ( 52 weeks ) | 8.006 |
| W ochen ende oder an Fe iert agen . THE Norwegian Fisher y Council | 7.970 |
| lands and islands and through the Norwegian f j ords and will continue to operate | 7.870 |
| single market by following Norway ' s model by joining the European Economic Area . | 7.853 |
| can either subs ist without Norway or simply dead zone it by stacking Uk r | 7.850 |
| is a member of the Norwegian Visual Artists Association and the Young Artists ' Society | 7.838 |
| Children Norway has been awarded the service contract for the Support to the Education Sector | 7.760 |

(d) The top 12 examples that produce the maximum latent activations for the input SAE latent with index 11698. Paired with the output SAE latent with index 64386, the Jacobian element is non-zero for 295 tokens, and has a mean of $1.344 \times 10^{-1}$ (rank 2 for the output SAE latent) and a standard deviation of $6.924 \times 10^{-3}$ over its non-zero values.

| Example tokens | Max. activation |
|---|---|
| kar , men du mAste vara in log g ad som an st Al ld | $1.023 \times 10^1$ |
| kA pa Euro och s Al ja US Dollar . \n Results per trade binary | 9.895 |
| l add ning och k lar att k opp las till slut st eg eller | 9.721 |
| ke vAl retro f Ar g ade f Ar den kA ns lig e . Barn | 9.363 |
| pA Chal mers fAr att kun na se dem . Sometimes I like to add | 9.343 |
| v An lig at mos f Ar och skA na rum , dock kans | 8.634 |
| och fAr fAr st Ar k are , upp till 20 tim mar ut an | 8.587 |
| C ancel . E tt ex emp el pA fore x hand el Ar att | 8.477 |
| dd rar mell an oli ka produk ter . Dev ices and systems on | 7.977 |
| A till A ! Cong rat z Eva Ny strAm and Ad riel Young ! | 7.812 |
| Det kan fin nas fl er pro jek t dAr Lars Bank vall med ver | 7.368 |
| pool och le k pl ats ! Tre v lig personal . \n – ub | 7.195 |

(e) The top 12 examples that produce the maximum latent activations for the input SAE latent with index 22804. Paired with the output SAE latent with index 64386, the Jacobian element is non-zero for 895 tokens, and has a mean of $1.342 \times 10^{-1}$ (rank 3 for the output SAE latent) and a standard deviation of $9.149 \times 10^{-3}$ over its non-zero values.

| Example tokens | Max. activation |
|---|---|
| so i fixed it myself but thats the first place you will get rust . | 5.811 |
| Models for Rust ing T ail gate Str uts . \n Tak ata recalls at | 5.395 |
| because you gave in easily Rust ic guys country a woman who is very to | 5.063 |
| bring to full resolution . Graphics are from a Rust ic W re ath collection | 4.752 |
| Rust ic Pic nic Table Popular Out door Furn iture Hand made By App al | 3.102 |
| we present . R ust ic Pic nic Table Amazing 31 All uring Ideas Tables | 2.680 |
| seeing . I have it in rust y pots and here it is in an | 2.359 |
| yields a Future , gen . cor out ine schedules the generator to be resumed | 2.104 |
| operations in order to ensure that the resulting LL VM IR can be ing ested | 1.849 |
| with @ gen . cor out ine . \n In Python 2 , the sub | 1.805 |
| from ‘ () <> ‘. The ‘ / ‘ has no meaning besides being a | 1.739 |
| you will have to upload the . pub file and copy its contents to authorized | 1.702 |

(f) The top 12 examples that produce the maximum latent activations for the input SAE latent with index 12754. Paired with the output SAE latent with index 64386, the Jacobian element is non-zero for 7 tokens, and has a mean of $1.168 \times 10^{-1}$ (rank 4 for the output SAE latent) and a standard deviation of $5.760 \times 10^{-3}$ over its non-zero values.

| Example tokens | Max. activation |
|---|---|
| Turkey the League B victory . \n Swed en , making its Nations League debut | $1.362 \times 10^1$ |
| 68 m 10 38 m 12 18 m HP on the Norwegian– Swed ish | $1.330 \times 10^1$ |
| for Lat v ian independence , R iga has been ruled by Germans , Swed | $1.286 \times 10^1$ |
| bar . \n summer when the Swed e began to shine . \n Cup | $1.180 \times 10^1$ |
| en , Al Un ser , Swed e Savage , Bobby Un ser , Gordon | $1.124 \times 10^1$ |
| 25 ppm in Denmark , France , Spain , Sweden and UK . \n " | $1.107 \times 10^1$ |
| \n This study was supported by the Swedish Research Council ( Grant K 2015– | $1.090 \times 10^1$ |
| Canada , Finland , Italy , Sweden and the UK . In the next few | $1.088 \times 10^1$ |
| ra ps in English , French , Spanish , Swedish and Mand arin . You | $1.076 \times 10^1$ |
| , the Middle East and Southeast Asia . \n In reality , the Swedish and | $1.070 \times 10^1$ |
| this threat is found from different countries such as Sweden , Malaysia , India , | $1.040 \times 10^1$ |
| iron ore mine in the world . It ' s owned by the Swedish government | $1.020 \times 10^1$ |

(g) The top 12 examples that produce the maximum latent activations for the input SAE latent with index 30912. Paired with the output SAE latent with index 64386, the Jacobian element is non-zero for 175 tokens, and has a mean of $1.133 \times 10^{-1}$ (rank 5 for the output SAE latent) and a standard deviation of $6.893 \times 10^{-3}$ over its non-zero values.

| Example tokens | Max. activation |
|---|---|
| ga , 2012 ClA ir se ach n A3 PAN ob , | 8.579 |
| la ri Paul Berg ag us Walter Gilbert . \n Dh ' ain mic | 8.236 |
| A j ir ag us Spar An abh fu il lu ach aH | 8.205 |
| a l A3 d A il ar an su AN om h . TA | 7.989 |
| . An san n AN m A3 r don i om ath A3 ir 5 | 7.893 |
| ire adh Ch un Ce oil ag us tr AN R na G . Is | 7.412 |
| m A nd (" mind , reason "). Related to Old English my nt an | 7.394 |
| 2013 na u irl is AN g iol ca igh mi ot ail ( | 7.251 |
| . \n Import o de k rom program o estas fac ila . \n A | 7.154 |
| \n A I iu event o lig i A L as al la orig ina | 7.080 |
| A il te ! St air na h Ai ire ann will take you on | 7.046 |
| b ild oj de event oj k aj b ild oj de k ateg orio | 7.012 |

(h) The top 12 examples that produce the maximum latent activations for the input SAE latent with index 57769. Paired with the output SAE latent with index 64386, the Jacobian element is non-zero for 321 tokens, and has a mean of $8.286 \times 10^{-2}$ (rank 6 for the output SAE latent) and a standard deviation of $1.343 \times 10^{-2}$ over its non-zero values.

| Example tokens | Max. activation |
| --- | --- |
| iGe eIKe i , i i iL~ | 2.700 |
| i H1 iLIJ i l l i i  i l e I | 2.051 |
| do that ? \n1 : 10 : 38 KEVIN : If you | 1.881 |
| e~e3 ei eIK i l ii iLl i i l | 1.822 |
| i iL~ ik 14 e IJ el , e i iL 14 ikl | 1.763 |
| L i ik~ e ei . e I i Il e L | 1.637 |
| IIJ aI aoI !! \n aI a al~ c2 e~ a I cli aI | 1.592 |
| e 314 e IJ e ge I i g 14 i l e | 1.564 |
| Bin is one of pictures that are related with the picture before in the collection | 1.488 |
| iLGe i iL 14 ikl e3 eiL 14 ikl i | 1.467 |
| sealed bearing . Now I feel kinda like a ∗∗∗∗∗ $$ for asking in | 1.458 |
| e L ik I ik ii l ik iLl i h I | 1.416 |

(i) The top 12 examples that produce the maximum latent activations for the input SAE latent with index 42113. Paired with the output SAE latent with index 64386, the Jacobian element is non-zero for 3 tokens, and has a mean of $7.971 \times 10^{-2}$ (rank 7 for the output SAE latent) and a standard deviation of $6.880 \times 10^{-3}$ over its non-zero values.

| Example tokens | Max. activation |
| --- | --- |
| and  arthritis  . \nY og ar aj G ugg ulu : Ing redients  include | $1.332 \times 10^1$ |
| Europe and Rest of World . Ing redients of this market are Sol vents . | $1.315 \times 10^1$ |
| \n 04 . Ing ros so , L io hn & Salv atore − Fl | $1.298 \times 10^1$ |
| an con el audio  original  en Ing les . El cost ode adm isiA3n | $1.298 \times 10^1$ |
| 2 , 000 calorie  diet . ∗ Daily Value not  established  . \n Other Ing | $1.245 \times 10^1$ |
| Sm ooth ie , Six Ing red ient Sp ir ul ina Sn ack ies | $1.201 \times 10^1$ |
| . 100 Ing redients In A Pictures For Kitchen Number 79 Is Im possible !. | $1.173 \times 10^1$ |
| Ind io Pap ago Te pez coh u ite Cream 2 Oz Ing redients : | $1.153 \times 10^1$ |
| . Ma Ing alls had no cabinets or  refrigerator  and she raised five children and | $1.137 \times 10^1$ |
| under stood Ing red ient , With Rec ipes , Simply Rec ipes ’ explanation | $1.132 \times 10^1$ |
| rop ur Ing redients is a global supplier of ingredients and services developed to create | $1.065 \times 10^1$ |
| Bah asa Ing gr is Sing kat is  just about the image we ascertained on | $1.045 \times 10^1$ |

(j) The top 12 examples that produce the maximum latent activations for the input SAE latent with index 8827. Paired with the output SAE latent with index 64386, the Jacobian element is non-zero for 21 tokens, and has a mean of $7.330 \times 10^{-2}$ (rank 8 for the output SAE latent) and a standard deviation of $5.439 \times 10^{-3}$ over its non-zero values.

| Example tokens | Max. activation |
|---|---|
| to scholars in Oriental Studies , Med ieval Literature , and History , The Gh | 9.650 |
| ia world . Be part of a medieval revolution against the Word King and his | 8.678 |
| trying to make a living in a medieval town full of warriors , sor ce | 8.207 |
| a couple who play medieval and Renaissance instruments , sing , dance and celebrate at | 8.198 |
| Sem ikh ah was the rise of the medieval university , which began to issue | 8.038 |
| located in Med ieval Sp on Street , Cov entry . Mark Andrew offers an | 8.015 |
| armies of the Cal iph ates , through the action of bloody medieval battles , | 7.999 |
| Ng aw ang Nam gy al , father and un ifier of medieval Bh ut | 7.911 |
| , I needed to know what are they ate in medieval northern England ? Lots | 7.847 |
| th Annual Award Win ning L ough rea Med ieval Festival takes place from the | 7.761 |
| or like the individual parts in a ' cycle ' of Med ieval mystery plays | 7.681 |
| id yll ic picture for the season . \nThe medieval charm of Vil n | 7.672 |

(k) The top 12 examples that produce the maximum latent activations for the input SAE latent with index 21697. Paired with the output SAE latent with index 64386, the Jacobian element is non-zero for 87 tokens, and has a mean of $6.570 \times 10^{-2}$ (rank 10 for the output SAE latent) and a standard deviation of $6.127 \times 10^{-3}$ over its non-zero values.

| Example tokens | Max. activation |
|---|---|
| football of his career , Russell Wilson finished Monday night ' s win over the Minnesota | $1.082 \times 10^1$ |
| . ST . PA UL , Minn . ( AP )− The Minnesota Department | $1.068 \times 10^1$ |
| the Minnesota Birth Center ; and D ian na J ol les , faculty at | $1.060 \times 10^1$ |
| the Minnesota Historical Society . \n Begin ning with the 1929 volume there is a | $1.060 \times 10^1$ |
| . \n The Winnipeg Row ing Club , the Minnesota Boat Club and the Saint | $1.055 \times 10^1$ |
| Miami Marl ins Milwaukee Brew ers Minnesota Tw ins . Kentucky Wild cats Win C | $1.050 \times 10^1$ |
| H . Allen . The lock out of the musicians at the Minnesota Orchestra has | $1.042 \times 10^1$ |
| Post started spec ulating this past week that the Minnesota Orche str al Association might | $1.032 \times 10^1$ |
| The Minnesota Wild have recalled defense man Ryan Murphy from A HL Iowa per a | $1.032 \times 10^1$ |
| An index to the Minnesota period ical collection in the South St . Paul Library | $1.010 \times 10^1$ |
| Marl ins Milwaukee Brew ers Minnesota . Add the United States of Baseball to your | $1.003 \times 10^1$ |
| Paul L ur line Row ing Club as the Minnesota and Winnipeg Row ing Association | $1.002 \times 10^1$ |

(l) The top 12 examples that produce the maximum latent activations for the input SAE latent with index 13110. Paired with the output SAE latent with index 64386, the Jacobian element is non-zero for 61 tokens, and has a mean of $6.486 \times 10^{-2}$ (rank 11 for the output SAE latent) and a standard deviation of $6.027 \times 10^{-3}$ over its non-zero values.

| Example tokens | Max. activation |
|---|---|
| , 2019 Posted in Official News Tags : Fl utter programming Leave a comment on | 6.019 |
| ? \n If you ' re not familiar with the Fl utter programming language and | 5.044 |
| Your Business with Amazon F BA ? \n Why You Should Master Fl utter Programming | 4.472 |
| mater the sought− after Google Fl utter language . Let ' s begin now | 3.056 |
| Why You Should Master Fl utter Programming ? rub ber plants indoor rubber plant growing | 2.706 |
| ew ar Mah ots av will be celebrated from 8 th April to 10 th | 1.795 |
| compatible with the industry . \n Widget s for your website . Order them . | 1.511 |
| this place . Eleanor notices this upon first arriving , thinking to herself that " | 1.489 |
| of illness and health . \n Find out why Swift Skin and W ound is | 1.488 |
| , some patients feel a st inging sensation from the injection & think it isn | 1.473 |
| You can also use the Contact Information widget from the Form W idget s . | 1.358 |
| Col oring Pages . Image Source : hou zz . com . Free Online | 1.283 |

(m) The top 12 examples that produce the maximum latent activations for the input SAE latent with index 26452. Paired with the output SAE latent with index 64386, the Jacobian element is non-zero for 4 tokens, and has a mean of $6.316 \times 10^{-2}$ (rank 12 for the output SAE latent) and a standard deviation of $6.164 \times 10^{-3}$ over its non-zero values.

| Example tokens | Max. activation |
| --- | --- |
| been made in extending Medicaid coverage to American Indians and Alaska N atives , the | $1.224 \times 10^1$ |
| percent of American Indian and Alaska Native children were enrolled in Medicaid or CHIP | $1.224 \times 10^1$ |
| uninsured rate for American Indian and Alaska Native children and families remain un accept ably | $1.153 \times 10^1$ |
| issues in the Park Service . \n Former Alaska governor and advisory board chairman Tony | $1.147 \times 10^1$ |
| and mindset for the Alaska Con ce aled Hand gun Perm it . T aught | $1.106 \times 10^1$ |
| popular appet izers , s oups and sandwiches . \n Loc ally caught Alaska seafood | $1.082 \times 10^1$ |
| is Canada , Alaska and Hawaii . Your mattress will be refund ed , however | $1.081 \times 10^1$ |
| for shipping quotes to other destinations such as Alaska and Hawaii . \n This product | $1.079 \times 10^1$ |
| Mo ose and Wolf hunt . \n We hunt the Alaska Yuk on mo ose | $1.066 \times 10^1$ |
| row se listings of Member users here at Alaska Fl irt that are associated with | $1.064 \times 10^1$ |
| Cherry , BL M ' s Alaska regional manager , said the agency was pleased | $1.057 \times 10^1$ |
| . The paint appeared to still be wet . The package includes Alaska Yuk on | $1.045 \times 10^1$ |

(n) The top 12 examples that produce the maximum latent activations for the input SAE latent with index 32153. Paired with the output SAE latent with index 64386, the Jacobian element is non-zero for 100 tokens, and has a mean of $6.202 \times 10^{-2}$ (rank 13 for the output SAE latent) and a standard deviation of $5.347 \times 10^{-3}$ over its non-zero values.

| Example tokens | Max. activation |
| --- | --- |
| Han over , Wake , and Fors y th have more than half of their | $1.788 \times 10^1$ |
| he told SMS F Ad vis er . \n Mr Fors y th explained this | $1.643 \times 10^1$ |
| arr ion " ' . Thanks to Anthony Fors of Absolute Clean for volunte ering | $1.616 \times 10^1$ |
| O read more into AT O IDs than they should ," Mr Fors y th | $1.531 \times 10^1$ |
| son Super Consult ing director Stuart Fors y th said shortly after the release of | $1.517 \times 10^1$ |
| web h omes fors ale . com help you find the Jacksonville , FL homes | $1.051 \times 10^1$ |
| uit your id ols ; fors ake your fond do ings ; and the promised | 7.176 |
| you , nor fors ake you " ( He b . 13 : 5 ); | 6.366 |
| for comfort in my difficult situation . Do not fors ake me , Good Jesus | 6.248 |
| ume already led many to fors ake the temple , and hold her ordin ances | 5.815 |
| which needs to be confessed and fors aken . \n 2 . We profit from | 5.704 |
| be submitted by e– mail ( sc anned to fors ik ring @ lease | 5.697 |

(o) The top 12 examples that produce the maximum latent activations for the input SAE latent with index 56394. Paired with the output SAE latent with index 64386, the Jacobian element is non-zero for 8 tokens, and has a mean of $5.939 \times 10^{-2}$ (rank 14 for the output SAE latent) and a standard deviation of $3.459 \times 10^{-3}$ over its non-zero values.

| Example tokens | Max. activation |
|---|---|
| Norwegian f j ords around St ord , Norway . \n Read more about our | $1.093 \times 10^1$ |
| 250 m crossing under a f j ord between Krist ians und and A ver | $1.089 \times 10^1$ |
| coast in Turkey , cru ising the F j ords of Norway , etc , | $1.033 \times 10^1$ |
| zones and sc ramble up to the top of the f j ord . There | $1.001 \times 10^1$ |
| 60 m 16 10 m S ogn og F j ord ane , Norway . | 9.988 |
| lands and islands and through the Norwegian f j ords and will continue to operate | 9.885 |
| You will also experience a f j ord cruise on the mighty S og ne | 9.601 |
| f j 80 stud is just slightly larger , so all I had to do | 9.451 |
| steering box anyways , so I decided to convert to f j 80 tie | 9.137 |
| is 57 . 5 kg , S agne f j orden , Norway , in | 9.027 |
| \n 426 Al ks f j ord j A~ kel en 120 4 m 113 | 8.791 |
| k j v Noah : November 29 , Ethical dile mmas occur when a situation | 8.114 |

(p) The top 12 examples that produce the maximum latent activations for the input SAE latent with index 36481. Paired with the output SAE latent with index 64386, the Jacobian element is non-zero for 36 tokens, and has a mean of $5.820 \times 10^{-2}$ (rank 15 for the output SAE latent) and a standard deviation of $4.891 \times 10^{-3}$ over its non-zero values.

*Figure 19.* The top 12 examples that produce the maximum latent activations for the output SAE latent with index 64386, and the input SAE latents with which the mean values of the corresponding Jacobian elements are greatest. The Jacobian SAE pair was trained on layer 15 of Pythia-410m with an expansion factor of $R = 64$ and sparsity $k = 32$. The examples were collected over the first 10K records of the English subset of the C4 text dataset with a context length of 16 tokens.

| Example tokens | Max. activation |
|---|---|
| surge into the cathode , producing a current that **does** the all − important work | 6.720 |
| so that he avoids distribution of designated product that apparently **does** not meet legal requirements | 6.606 |
| , but also expansive alien environments that **do their** part to make the audience feel | 6.584 |
| of 2016 ( 88 percent ) were opportun istic attacks that **did** not target a | 6.467 |
| to make therapeutic claims , were found to **do** so . \n Unfortunately the proposed | 6.406 |
| . Children are the most severely affected by poverty because they **do** not have the | 6.331 |
| hospital admissions ; emergency room visits that **do** not result in admission are excluded . | 6.282 |
| a court order . Pro pos als relating to children often **do** not need to | 6.264 |
| . S uitable exercises for pregnant women are those that **do** not strain the lower | 6.255 |
| ? \n Unfortunately , the traditional Chinese approach to training in the modern world **does** | 6.240 |
| also request fat sh aming to be made illegal because it **does** not have any | 6.232 |
| select the 1 dB degradation to noise as the interference standard , since it **does** | 6.221 |

(a) The top 12 examples that produce the maximum latent activations for the output SAE latent with index 60542. Output latent 60542 responds to a very specific use of the word "do": its use as a pro-verb. In a pro-verb a simple verb stands in for another more complex one and here "do" is a shorthand for an action that can only be understood from the context, for example, in "were found to do so" the "to do so" stands in for "to make therapeutic claims". Some of the inputs include very different uses of "do", one for example deals with the "Done" in "Donegal", an Irish county. However, another input includes a subtly different use of "do": cases where "do" is used as an auxiliary, modifying another verb, as in "[t]his does not meet the requirements". Clearly this circuit is creating a very fine distinction between different ways the word "do" can be used, a distinction we make in language comprehension, but one we would have trouble identifying or describing.

| Example tokens | Max. activation |
|---|---|
| Your information will not be stored on HSBC ' s systems if you **do** | $2.098 \times 10^1$ |
| . Children are the most severely affected by poverty because they **do** not have the | $2.077 \times 10^1$ |
| surge into the cathode , producing a current that **does** the all − important work | $2.056 \times 10^1$ |
| . 001 ) and fewer over tri aged children who **did** not require inpatient management | $2.043 \times 10^1$ |
| , but also expansive alien environments that **do their** part to make the audience feel | $2.032 \times 10^1$ |
| us at your own cost within days of delivery . If you **do** not | $2.027 \times 10^1$ |
| them post menopausal . They were compared with more than 600 women who **did** not | $2.014 \times 10^1$ |
| despite the risk . When persecution came , they **did** not scatter . They remained | $2.001 \times 10^1$ |
| levels protect public health and welfare if they **do** not exceed 45 dB . \n | $1.997 \times 10^1$ |
| so that he avoids distribution of designated product that apparently **does** not meet legal requirements | $1.996 \times 10^1$ |
| hospital admissions ; emergency room visits that **do** not result in admission are excluded . | $1.984 \times 10^1$ |
| . S uitable exercises for pregnant women are those that **do** not strain the lower | $1.976 \times 10^1$ |

(b) The top 12 examples that produce the maximum latent activations for the input SAE latent with index 21465. Paired with the output SAE latent with index 60542, the Jacobian element is non-zero for 13107 tokens, and has a mean of $2.529 \times 10^{-1}$ (rank 0 for the output SAE latent) and a standard deviation of $5.392 \times 10^{-2}$ over its non-zero values.

| Example tokens | Max. activation |
|---|---|
| . We expect the market to do the opposite of what the indicators are saying | $1.438 \times 10^1$ |
| with the 2018 elections looming . \nThey believe that Trump has done a good | $1.313 \times 10^1$ |
| only thing that she wanted in the world , but she did a bad thing | $1.292 \times 10^1$ |
| led by its President Jim Cor cor an , have done a wonderful job of | $1.292 \times 10^1$ |
| MBTA does so as well . And many state laws do the same . | $1.264 \times 10^1$ |
| a " t ote ; " I just buy the boxes .) These do a | $1.261 \times 10^1$ |
| can ' t win games , you ' re in trouble . \n Saint s did a | $1.258 \times 10^1$ |
| be ex orbit ant . You need do your own research to find out the | $1.257 \times 10^1$ |
| up for grabs , it ' s always Jews doing the grabbing ! \n Here | $1.251 \times 10^1$ |
| your muscles do the work , going at a sn ail ' s pace doesn ' t | $1.246 \times 10^1$ |
| to worry about whether the Black haw ks are doing the right thing with defense | $1.244 \times 10^1$ |
| You ' re doing the same thing on Amazon through sponsored ads and proving yourself | $1.239 \times 10^1$ |

(c) The top 12 examples that produce the maximum latent activations for the input SAE latent with index 61756. Paired with the output SAE latent with index 60542, the Jacobian element is non-zero for 2683 tokens, and has a mean of $1.076 \times 10^{-1}$ (rank 1 for the output SAE latent) and a standard deviation of $2.253 \times 10^{-2}$ over its non-zero values.

| Example tokens | Max. activation |
|---|---|
| A du grand lux e , il fait ce qu ' on lui dit , | $1.044 \times 10^1$ |
| . Para mAs inform aciA3n , h aga cl ic en el la zo ab | 9.095 |
| cl ase de ing l As y por hacer la present aciA3n . A E | 8.587 |
| qu A h aces ? \n if you want to go back to a previous | 6.694 |
| si vous fa ite des codes ! \n l NGNh12 1 NI1 | 5.713 |
| co , donde se real iza la seg unda esc en a y final mente , | 5.181 |
| into the software . The K art ra interface is faire ly well designed for | 4.361 |
| which they draw drinking water . \nLeg isl ators must find the faire st | 4.314 |
| en do en . \n ' Living with Fran ', van af z ond ag | 4.312 |
| . De k ache ls m aken ve el law a ai . We ver | 4.253 |
| organis ieren of tm als ih re eigen en Games . Es tut uns le | 4.225 |
| e en p oner los hue v os en el interior del hu A sp | 4.202 |

(d) The top 12 examples that produce the maximum latent activations for the input SAE latent with index 51331. Paired with the output SAE latent with index 60542, the Jacobian element is non-zero for 90 tokens, and has a mean of $8.908 \times 10^{-2}$ (rank 2 for the output SAE latent) and a standard deviation of $9.883 \times 10^{-3}$ over its non-zero values.

| Example tokens | Max. activation |
|---|---|
| the middle of war again . \nThe Seventh `Done` gal took part in the | $1.341 \times 10^1$ |
| all hurting tonight . I ' ll share details later this week . \n`Done` ! | $1.216 \times 10^1$ |
| fit or there ' s something wrong please contact us . \n`Done` with Multi | $1.211 \times 10^1$ |
| this point . \n`Done` . Will be interested to learn more about this myself | $1.192 \times 10^1$ |
| the goal of Universal Male– Female Liter acy a `done` deal . \nNo | $1.191 \times 10^1$ |
| feeling of being let down , overwhelmed and `done` in . Dis appoint ment is | $1.191 \times 10^1$ |
| there ' s something wrong please contact us ! \n`Done` with Egg– shaped | $1.188 \times 10^1$ |
| first gin distilled in Co . `Done` gal . D Ao lam An is the | $1.179 \times 10^1$ |
| Cl osure and re clamation of a mining property is a complex process . `Done` | $1.168 \times 10^1$ |
| they could take . \nThe `Done` gal Guards , like no other arm of | $1.155 \times 10^1$ |
| were able to finish rebuilding efforts , and the `Done` gal Guards found themselves in | $1.150 \times 10^1$ |
| out his heart to the `done` ky . One day , K esh ava was | $1.146 \times 10^1$ |

(e) The top 12 examples that produce the maximum latent activations for the input SAE latent with index 11694. Paired with the output SAE latent with index 60542, the Jacobian element is non-zero for 540 tokens, and has a mean of $7.039 \times 10^{-2}$ (rank 3 for the output SAE latent) and a standard deviation of $1.967 \times 10^{-2}$ over its non-zero values.

| Example tokens | Max. activation |
|---|---|
| relax mechanisms . \nOn `the` other hand , between the classic models , it | $1.057 \times 10^1$ |
| protein . For zinc prot op orph yr in , on `the` other hand , | $1.020 \times 10^1$ |
| compared to Cauc as ians . On `the` other hand , the latter have a | 9.974 |
| all change associated with un ification . \nOn `the` first point , see J | 9.717 |
| ) the effect of class size varies across students . \nOn `the` first form | 9.667 |
| disease . \nOn `the` second day of hospitalization , B its y ' s | 9.630 |
| much scope for development . On `the` other hand , there is much scope for | 9.605 |
| for whom it is of minimal use . On `the` other hand , when hazards | 9.602 |
| regulates its collective temperature at different ambient air temperatures ; on `the` left side it | 9.537 |
| is . On `the` inside there are also lent icular films that diffuse the view | 9.520 |
| associated with these ex libr ises . On `the` one hand , one part of | 9.519 |
| states . On `the` other hand , we need to define our destinations in a | 9.441 |

(f) The top 12 examples that produce the maximum latent activations for the input SAE latent with index 48418. Paired with the output SAE latent with index 60542, the Jacobian element is non-zero for 91 tokens, and has a mean of $5.105 \times 10^{-2}$ (rank 4 for the output SAE latent) and a standard deviation of $9.043 \times 10^{-3}$ over its non-zero values.

| Example tokens | Max. activation |
| --- | --- |
| of the date of filing of the petition . \n A change of name or | 5.590 |
| the jury . \n An interruption of the opposing counsel with a speech rather than | 5.370 |
| of these statutory changes . \n A new form , called a " Dis closure | 5.359 |
| . The Court will examine only marginally whether the principle is fulfilled . A detailed | 5.215 |
| filing of the petition . \n A merger is recorded by means of a petition | 5.214 |
| had committed a crime while out on release : obstruction of justice . \n A | 4.802 |
| does not accept liability for failure to deliver within the stated time . \n A | 4.802 |
| igated sentencing . An experienced Mar ic opa County D U I Attorney will know what | 4.660 |
| 169 of the Revised Statutes . \n An te cedent : Ref erring to a | 4.463 |
| trial ; judge 's availability limited . \n A second discrimination trial against the University | 4.404 |
| a grounds for dealing with the issue . A pharmacy is also nearby which sells | 4.393 |
| to comply with its requirements . The court noted that "[ a ] d vance | 4.323 |

(g) The top 12 examples that produce the maximum latent activations for the input SAE latent with index 32517. Paired with the output SAE latent with index 60542, the Jacobian element is non-zero for 1 tokens, and has a mean of $5.015 \times 10^{-2}$ (rank 5 for the output SAE latent) and a standard deviation of 0.000 over its non-zero values.

| Example tokens | Max. activation |
| --- | --- |
| photos . Import ant : This product does not include the techniques used in the | $1.195 \times 10^1$ |
| pump you are looking for . \n This product does not contain 1 , | $1.182 \times 10^1$ |
| Furthermore , exp or ail . org and web sites associated does not sell user | $1.146 \times 10^1$ |
| . \n The name T ee berg does not appear on any of the top | $1.130 \times 10^1$ |
| suggestion for ASC Utt imate . \n Do you do not need to worry about | $1.123 \times 10^1$ |
| ube . com Limited does not endorse any user submission , and expressly disclaim s | $1.119 \times 10^1$ |
| ube . com Limited does not permit copyright infring ing activities or infringement of intellectual | $1.119 \times 10^1$ |
| ' s family , it doesn ' t for Mark he told Ost rov | $1.095 \times 10^1$ |
| ig ences . com website or webpage , Dil ig ences does not represent or | $1.087 \times 10^1$ |
| market , it does not help anyone no mater how great the thought is . | $1.085 \times 10^1$ |
| marketing , Blog ging , internet marketing \t . \n The world doesn ' t | $1.075 \times 10^1$ |
| a daily basis . \n Hey , I don ' t know if I 'm posting | $1.075 \times 10^1$ |

(h) The top 12 examples that produce the maximum latent activations for the input SAE latent with index 23968. Paired with the output SAE latent with index 60542, the Jacobian element is non-zero for 2561 tokens, and has a mean of $4.760 \times 10^{-2}$ (rank 6 for the output SAE latent) and a standard deviation of $1.204 \times 10^{-2}$ over its non-zero values.

| Example tokens | Max. activation |
| --- | --- |
| you can lower your standard of living . And a lower standard of living does | 9.428 |
| ? Of course not . A post humous diagnosis does not change who he | 9.376 |
| holds equity in a private business or has a large portfolio this does not | 9.172 |
| that this is a negative review , the above observations do constitute qu ib bles | 9.046 |
| Tax ote re and permanent hair loss , those warnings do not appear on any | 8.894 |
| have no responsibility or control over them . The existence of these links does not | 8.834 |
| and ambitious . \nM ission success does not require these complex maneu vers , | 8.667 |
| . Please note these restrictions do not apply to exhib itors . \nTo help | 8.648 |
| Yellow stone if there are only that many sub species . That does not include | 8.556 |
| But the Tribunal explained , " Training alone does not meet the requirements of due | 8.529 |
| \nSome schools prefer to be data controllers . This does make it easier for | 8.483 |
| extreme heat . It ' s naturally darker colour doesn ' t do itself any | 8.473 |

(i) The top 12 examples that produce the maximum latent activations for the input SAE latent with index 19973. Paired with the output SAE latent with index 60542, the Jacobian element is non-zero for 2691 tokens, and has a mean of $4.681 \times 10^{-2}$ (rank 7 for the output SAE latent) and a standard deviation of $9.146 \times 10^{-3}$ over its non-zero values.

| Example tokens | Max. activation |
| --- | --- |
| play and revenue . Well , that no longer appears to be the case . | 8.329 |
| than he believes or cares to say . Poll s now demonstrate this to be | 8.314 |
| systems are often thought to be much more securely protected than is actually the case | 7.874 |
| relations with residents . Unfortunately , with some boards this is not always the case | 7.272 |
| some cases the spam detection stopped working . This is no longer the case . | 7.104 |
| ' t ) and if there are no clear winners between them ( which is usually | 7.036 |
| found the same to be true for them !). Ch ur ro Fried Ice Cream | 6.854 |
| . \nHowever , even if that is not correct , the Court erroneously uses | 6.826 |
| final . However , that ' s not always the case . There are plenty | 6.787 |
| if that were not bad enough for Thor pe ' s membership , in the same | 6.773 |
| the lease period has expired and no oil has been produced . If this is | 6.688 |
| or missing from the root of the domain you added . If this is the | 6.680 |

(j) The top 12 examples that produce the maximum latent activations for the input SAE latent with index 13058. Paired with the output SAE latent with index 60542, the Jacobian element is non-zero for 431 tokens, and has a mean of $4.528 \times 10^{-2}$ (rank 8 for the output SAE latent) and a standard deviation of $1.961 \times 10^{-2}$ over its non-zero values.

| Example tokens | Max. activation |
|---|---|
| Compan ies must give workers the information they need to `perform` their jobs safely , | $1.448 \times 10^1$ |
| too short to drink bad coffee that makes you `perform` poorly and feel awful . | $1.448 \times 10^1$ |
| of your reserve fund . By `performing` a reserve study , you can determine how | $1.426 \times 10^1$ |
| g mail account , `perform` my edit , and then un install in a matter | $1.410 \times 10^1$ |
| worn fill at the tail pipe as permitted to `perform` every engine and every rate | $1.409 \times 10^1$ |
| s car just got away from him and he `performed` a lazy spin into the | $1.400 \times 10^1$ |
| from , modify , publish , edit , translate , distribute , `perform` , display | $1.400 \times 10^1$ |
| The firm did not `perform` impurity testing during stability on two products since 2016 . | $1.397 \times 10^1$ |
| stop thinking about mess ing up . You `performed` so tent atively that you hoped | $1.389 \times 10^1$ |
| like how black hat hackers are incent iv ized by money to `perform` malicious activities | $1.388 \times 10^1$ |
| and payments encourage PBM s to actually `perform` to their contractual guarantees instead of | $1.367 \times 10^1$ |
| individual educational plans ; assists in inventory control ; and `performs` other duties related to | $1.364 \times 10^1$ |

(k) The top 12 examples that produce the maximum latent activations for the input SAE latent with index 56700. Paired with the output SAE latent with index 60542, the Jacobian element is non-zero for 574 tokens, and has a mean of $4.375 \times 10^{-2}$ (rank 9 for the output SAE latent) and a standard deviation of $1.275 \times 10^{-2}$ over its non-zero values.

| Example tokens | Max. activation |
|---|---|
| share the posts as if your life depended on it . Which `it` doesn `’ t` | 8.326 |
| encounters an impossible reality that shouldn ’ t exist , `but` `does` . \nA | 8.273 |
| ask ! Who knew las agna was for dessert ? `We` `sure` didn ’ `t` | 7.129 |
| to make sure that it works correctly and `it` `does` . \nWhat ’ | 6.960 |
| , status and aside . Not every WordPress theme supports post formats and those `that` | 6.757 |
| you ever thanked me ? \n `no` `you` `actually` `did` , but I can count | 6.578 |
| if this would kick in , but `it` didn ’ `t` . \n After installing | 6.342 |
| don ’ t need convincing . But `do` `you` know who `does` ? \n And | 6.318 |
| big one . It seems to work . Or `does` `it` ? Then three fish | 6.302 |
| . It seems to work . Or `does` `it` ? Then three fish get the | 6.277 |
| Spin ! You dont switch but your opponent `does` so you can stack damage on | 6.275 |
| Some how I envisioned our family as a family of four . I `still` `do` | 6.228 |

(l) The top 12 examples that produce the maximum latent activations for the input SAE latent with index 46097. Paired with the output SAE latent with index 60542, the Jacobian element is non-zero for 3589 tokens, and has a mean of $3.995 \times 10^{-2}$ (rank 10 for the output SAE latent) and a standard deviation of $2.127 \times 10^{-2}$ over its non-zero values.

| Example tokens | Max. activation |
|---|---|
| its spark . Employees are not inspired , stakeholders are becoming color less and you | 5.325 |
| are talking about are humans , and not robots , and so a greater amount | 4.681 |
| nice " not too fast , not too slow " rhythm and settled in to | 4.430 |
| end up being about communication not between professions , but with clients and families | 4.387 |
| pre− s lic ed is easiest not too thin , and nothing wet | 4.313 |
| , I did not know how the French health care worked and I did not | 4.230 |
| move frequently , not be on school district census rolls , and are less likely | 4.183 |
| Total PM . A way of life . Not a job . \n It was | 4.141 |
| the fire which does not burn , the water which does not wet the hands | 4.113 |
| I do not move , I do not breath . I close my eyes and | 4.075 |
| It was not for a meeting with an international leader . \n And it was | 4.069 |
| such thing as safe plastic aG not for you and your family , and not | 4.018 |

(m) The top 12 examples that produce the maximum latent activations for the input SAE latent with index 4510. Paired with the output SAE latent with index 60542, the Jacobian element is non-zero for 133 tokens, and has a mean of $3.978 \times 10^{-2}$ (rank 11 for the output SAE latent) and a standard deviation of $1.006 \times 10^{-2}$ over its non-zero values.

| Example tokens | Max. activation |
|---|---|
| witnessing before moving on to more difficult or resistant areas . Rather the opposite was | $1.143 \times 10^1$ |
| decision making authority . Actually , the opposite is implied . Let me put it | $1.059 \times 10^1$ |
| should come before focusing on one ' s own , and I feel the reverse | $1.048 \times 10^1$ |
| t faith . It ' s actually the opposite of hope . Our other options | $1.017 \times 10^1$ |
| tourist − rich locale , but rather , the opposite is true . El is | $1.004 \times 10^1$ |
| good guy ; if anything the opposite was said when saying Zimm erman ' s | 9.978 |
| out of fear mode , you return to love mode . The opposite of love | 9.799 |
| in any way−− just the opposite . The fine balance between supp len ess | 9.498 |
| . process as perform argues the opposite : that learn on method must take heavily | 9.484 |
| says the right words to us , but he acts just the opposite . He | 9.464 |
| ' s a bad thing , in this case , it means the opposite . | 9.423 |
| − EGR− val ve Without voltage applied , it ' s just the opposite | 9.166 |

(n) The top 12 examples that produce the maximum latent activations for the input SAE latent with index 28695. Paired with the output SAE latent with index 60542, the Jacobian element is non-zero for 53 tokens, and has a mean of $3.952 \times 10^{-2}$ (rank 12 for the output SAE latent) and a standard deviation of $7.511 \times 10^{-3}$ over its non-zero values.

| Example tokens | Max. activation |
|---|---|
| \n I ' ve got to admit , I ' ve heard some " do oz ie | 8.284 |
| silent film . Google ' s ever changing " google do od le " logo | 7.809 |
| our annual celebrations can be . Do pp el k lic ke den Number Sl | 7.234 |
| 3 . \n Here ' s another do oz ie : Both Av ig dor | 7.154 |
| doing a few do od les of people in suits with their arms crossed . | 6.835 |
| the institutions tend first information in using the has hers , print able do od | 6.828 |
| ee F aret . This was long before the Do ob ie Brothers ever thought | 6.275 |
| keeping were a part of our life and I wanted to know about Do ob | 6.262 |
| an object I do remember is Do ob ee F aret . It was a | 6.177 |
| adopt me . Name on birth c irt ificate was baby do y j ohn | 6.137 |
| found that do od lers performed 29% better than non− d ood lers | 5.989 |
| a do v ish Fed and institutional instability in the US ," said Kyle Rod | 5.951 |

(o) The top 12 examples that produce the maximum latent activations for the input SAE latent with index 26469. Paired with the output SAE latent with index 60542, the Jacobian element is non-zero for 29 tokens, and has a mean of $3.831 \times 10^{-2}$ (rank 13 for the output SAE latent) and a standard deviation of $1.040 \times 10^{-2}$ over its non-zero values.

| Example tokens | Max. activation |
|---|---|
| Sp an ien k auf en , kAnnen Sie eine span ische Auf enthal ts | 8.197 |
| d abe i auch auf Sport w et ten , die man auch be qu | 8.086 |
| acht eine Ab z oc ke , da her w A14r de ich r A14ck | 8.018 |
| k auf en will , kann ich tro tz dem das Gold V isa be | 7.818 |
| ft wer de ich A fter im Dr A14ck Gl A14ck Online Casino an Turn | 7.550 |
| and und mehr ! \n Fr age : K ann ich ein span is ches | 7.512 |
| sind . \n Fr age : K ann ich eine Fin anz ierung er hal | 7.413 |
| , dass es ges etz lich an erk ann te An ge h Ar ige | 7.390 |
| K ann ich nur we ite remp fe h len ." \n " This | 7.366 |
| we isen , dass Sie dire kt oder ind ire kt In hab er der | 7.351 |
| we isen , dass Sie nicht mehr als 11 Mon ate a uss er hal | 7.351 |
| age : W enn ich die Imm ob il ie durch eine Ges ells chaft | 7.225 |

(p) The top 12 examples that produce the maximum latent activations for the input SAE latent with index 14813. Paired with the output SAE latent with index 60542, the Jacobian element is non-zero for 2 tokens, and has a mean of $3.811 \times 10^{-2}$ (rank 14 for the output SAE latent) and a standard deviation of $8.688 \times 10^{-3}$ over its non-zero values.

| Example tokens | Max. activation |
|---|---|
| . \n4 Does the Pale o Diet E lim inate Healthy Foods Like Brown | 3.101 |
| for all for profit show tickets as required by Louisiana law . \nDoes not | 3.047 |
| 0334 . \nDoes my infant / child need a ticket for the show | 2.843 |
| 000 . \nDoes CODE undertake its own due diligence ? \nYes , should | 2.575 |
| duties . \nDoes this mean we are moving to a system where the Pope | 1.695 |
| am ore ; Ronald Bra ut ig am , p f ; Leo van Does | 1.543 |
| . \nDoes n ' t clog pores nor does cause any acne break out | 1.511 |
| . \nDoes he have a lot of toys ? Yeah , can we bring | 1.511 |
| . Does Ben Bishop fix their go alt ending ? Where do Alexander Rad ul | 1.479 |
| . Does this mean that the leaders who engage in the methods are bad people | 1.479 |
| . Does yours look as pretty as mine ? \n I ' m going to | 1.479 |
| day . \nDoes your marketing team regularly grapp le with problems ? If so | 1.306 |

(q) The top 12 examples that produce the maximum latent activations for the input SAE latent with index 41425. Paired with the output SAE latent with index 60542, the Jacobian element is non-zero for 17 tokens, and has a mean of $3.734 \times 10^{-2}$ (rank 15 for the output SAE latent) and a standard deviation of $8.725 \times 10^{-3}$ over its non-zero values.

*Figure 20.* The top 12 examples that produce the maximum latent activations for the output SAE latent with index 60542, and the input SAE latents with which the mean values of the corresponding Jacobian elements are greatest. The Jacobian SAE pair was trained on layer 15 of Pythia-410m with an expansion factor of $R = 64$ and sparsity $k = 32$. The examples were collected over the first 10K records of the English subset of the C4 text dataset with a context length of 16 tokens.

# D. Training

Our training implementation is based on the open-source SAELens library (Bloom et al., 2024). We train each pair of SAEs on 300 million tokens from the Pile (Gao et al., 2020), excluding the copyrighted Books3 dataset, for a single epoch. Except where noted, we use a batch size of 4096 sequences, each with a context size of 2048. At a given time, we maintain 32 such batches of activation vectors in a buffer that is shuffled before training, which reduces variance in the training signal.

We use the Adam optimizer (Kingma & Ba, 2017) with the default beta parameters and a constant learning-rate schedule with 1% warm-up steps, 20% decay steps, and a maximum value of $5 \times 10^{-4}$ Additionally, we use 5% warm-up steps for the coefficient of the Jacobian term in the training loss. We initialize the decoder weight matrix to the transpose of the encoder, and we scale the decoder weight vectors to unit norm at initialization and after each training step (Gao et al., 2024).

Except where noted, we choose an expansion factor $R = 32$, keep the $k = 32$ largest latents in the TopK activation function of each of the input and output SAEs, and choose a coefficient of $\lambda = 1$ for the Jacobian term in the training loss.

## D.1. Training signal stability

We initially considered the following setup:

$$\mathbf{s_x} = e_x(\mathbf{x}), \quad \hat{\mathbf{x}} = d_x(\mathbf{s_x}), \quad \mathbf{y} = f(\hat{\mathbf{x}}), \quad \mathbf{s_y} = e_y(\mathbf{y}), \quad \hat{\mathbf{y}} = d_y(\mathbf{s_y}) \tag{32}$$

The problem with this arrangement is that the second SAE depends on an output from the first SAE. Since both SAEs are trained simultaneously, we found that this compromised training signal stability – whenever the first SAE changed, the training distribution of the second SAE changed with it. Additionally, at the start of training, when the first SAE was not yet capable of outputting anything meaningful, the second SAE had no meaningful training data at all, which not only made it impossible for the second SAE to learn but also made the first SAE less stable via the Jacobian sparsity loss term.

To address this problem, we instead used this setup:

$$\mathbf{s_x} = e_x(\mathbf{x}), \quad \hat{\mathbf{x}} = d_x(\mathbf{s_x}), \quad \mathbf{y} = f(\mathbf{x}), \quad \mathbf{s_y} = e_y(\mathbf{x}), \quad \hat{\mathbf{y}} = d_y(\mathbf{s_y}) \tag{33}$$

Importantly, we pass the actual pre-MLP activations $\mathbf{x}$ rather than the reconstructed activations $\hat{\mathbf{x}}$ into the MLP $f$. In addition to improving training stability, we believe this setup to be more faithful to the underlying model because both SAEs are trained on the unmodified activations that pass through the MLP.

# E. Evaluation

We evaluated each of the input and output SAEs during training on ten batches of eight sequences, where each sequence has a context size of 2048, i.e., approximately 160K tokens. We computed the sparsity of the Jacobian, measured by the mean number of absolute values above $0.01$ for a single token, separately after training. In this case, we collected statistics over 10 million tokens from the validation subset of the C4 text dataset.

For reconstruction quality, we report the mean cosine similarity between input activation vectors and their autoencoder reconstructions, the explained variance (MSE reconstruction error divided by the variance of the input activation vectors), and the MSE reconstruction error.

For model performance preservation, we report the cross-entropy loss score, which is the increase in the cross-entropy loss when the input activations are replaced by their autoencoder reconstruction divided by the increase in the loss when the input activations are ablated (set to zero).

For sparsity, we report the number of 'dead' latents that have not been activated (i.e., appeared in the $k$ largest latents of the TopK activation function) within the preceding 10 million tokens during training and the number of latents that have activated fewer than once per 1 million tokens during training on average.

Given an expansion factor of $64$, $k = 32$, and a Jacobian loss coefficient of 1, i.e., fixed hyperparameters, we find that the reconstruction error and cross-entropy loss score are consistently better for the input SAE than the output SAE. Additionally, we find that the performance is generally poorer for the intermediate layers than early and later layers.

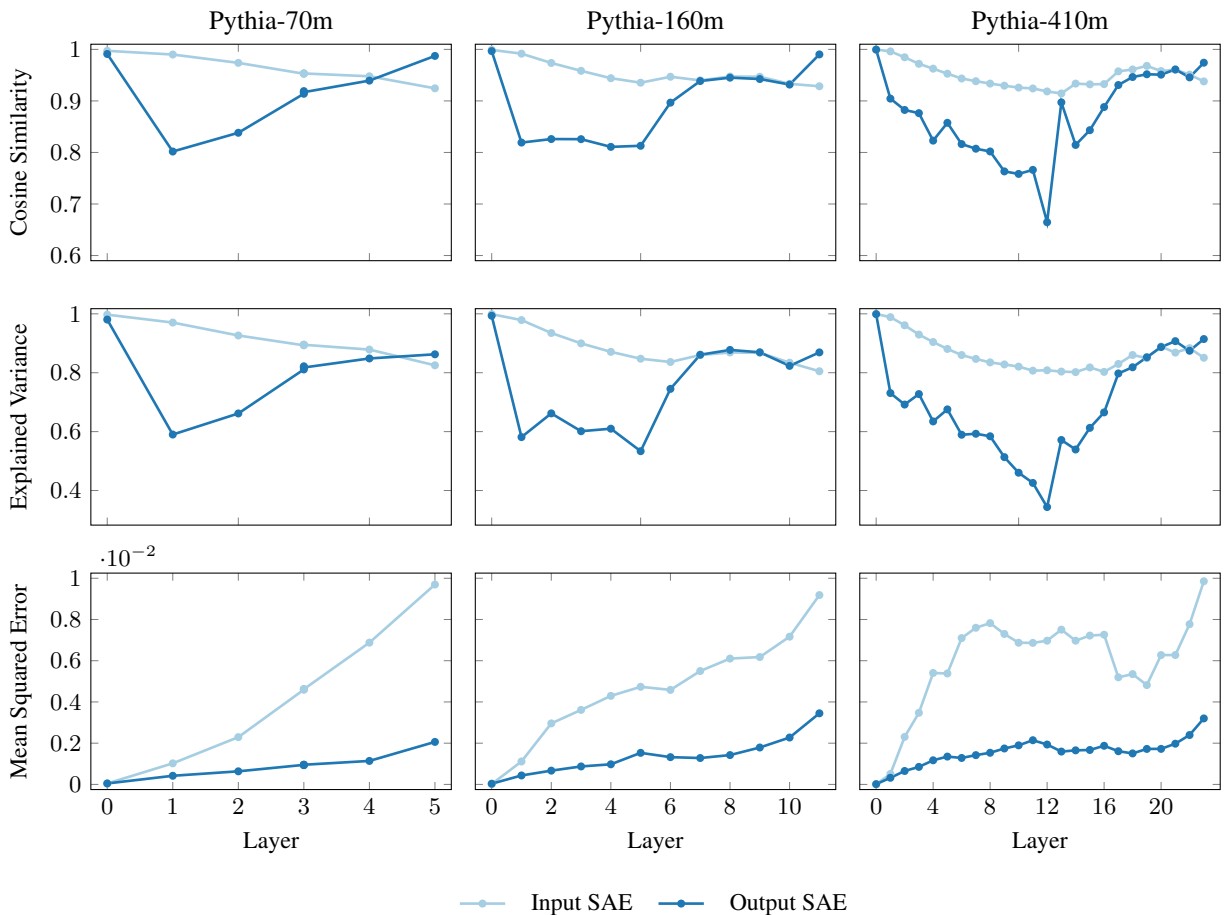

*Figure 21.* Reconstruction quality metrics for Jacobian SAEs trained on the feed-forward networks at every layer (residual block) of Pythia transformers. The cosine similarity is taken between the input and reconstructed activation vectors, and the explained variance is the MSE reconstruction error divided by the variance of the input activations. For each SAE, the expansion factor is $R = 64$ and $k = 32$; the Jacobian loss coefficient is 1.

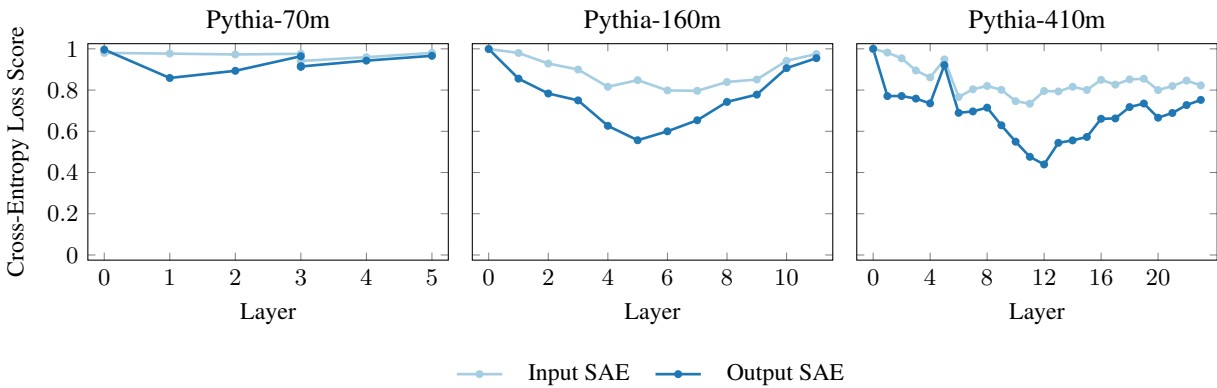

*Figure 22.* Model performance preservation metrics for Jacobian SAEs trained on the feed-forward networks at every layer (residual block) of Pythia transformers. The cross-entropy loss score is the increase in the cross-entropy loss when the input activations are replaced by their autoencoder reconstruction divided by the increase when the input activations are ablated (set to zero). For each SAE, the expansion factor is $R = 64$ and $k = 32$; the Jacobian loss coefficient is 1.

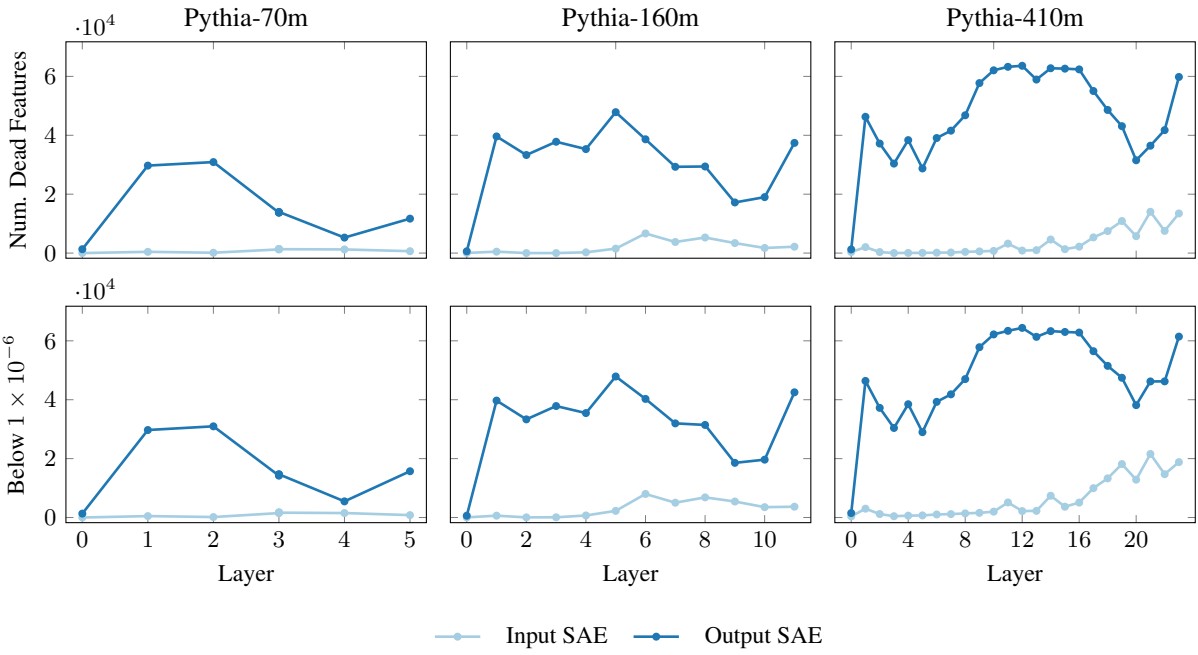

*Figure 23.* Sparsity metrics per layer for Jacobian SAEs trained on the feed-forward networks at every layer (residual block) of Pythia transformers. Recall that the $L^0$ norm per token for each of the input and output SAEs is fixed at $k$ by the TopK activation function. For each SAE, the expansion factor is $R = 64$ and $k = 32$; the Jacobian loss coefficient is 1.

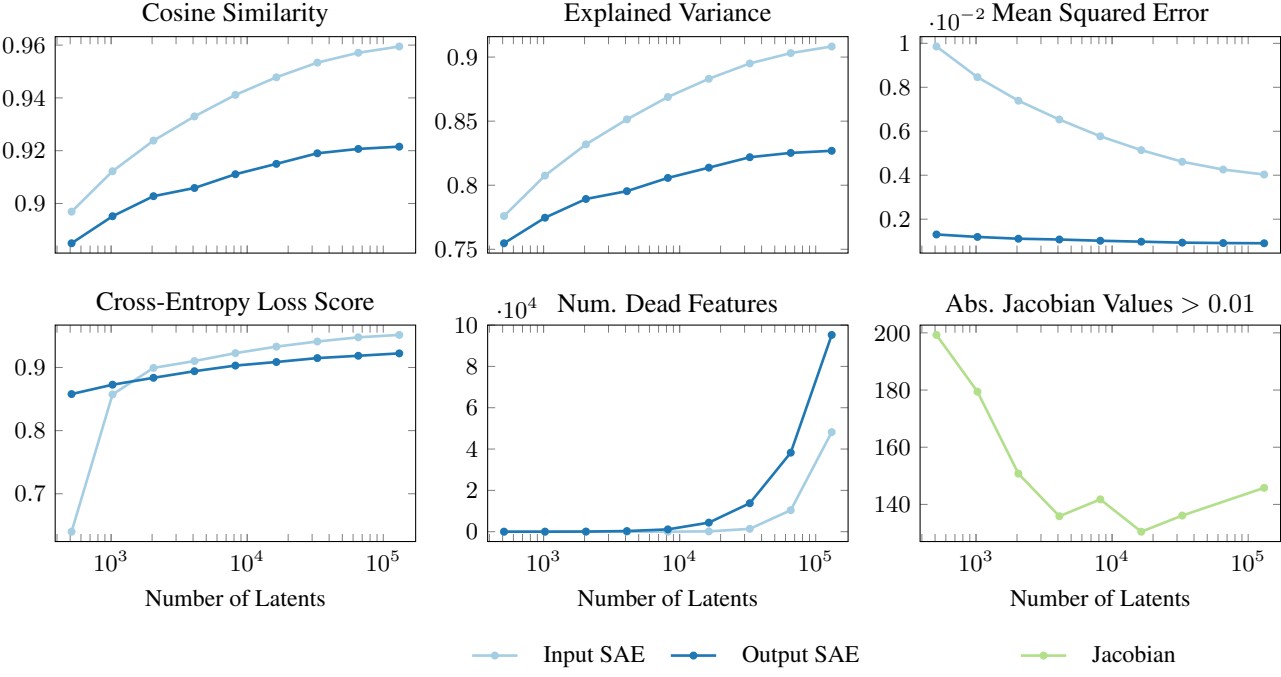

*Figure 24.* Reconstruction quality, model performance preservation, and sparsity metrics against the number of latents. Here, we consider Jacobian SAEs trained on the feed-forward network at layer 3 of Pythia-70m (model dimension 512) with $k = 32$. Recall that the maximum number of non-zero Jacobian values is $k^2 = 1024$. The reconstruction quality and cross-entropy loss score improve as the number of latents increases, and the number of dead features grows more quickly for the output SAE than the input SAE. See Appendix E for details of the evaluation metrics.

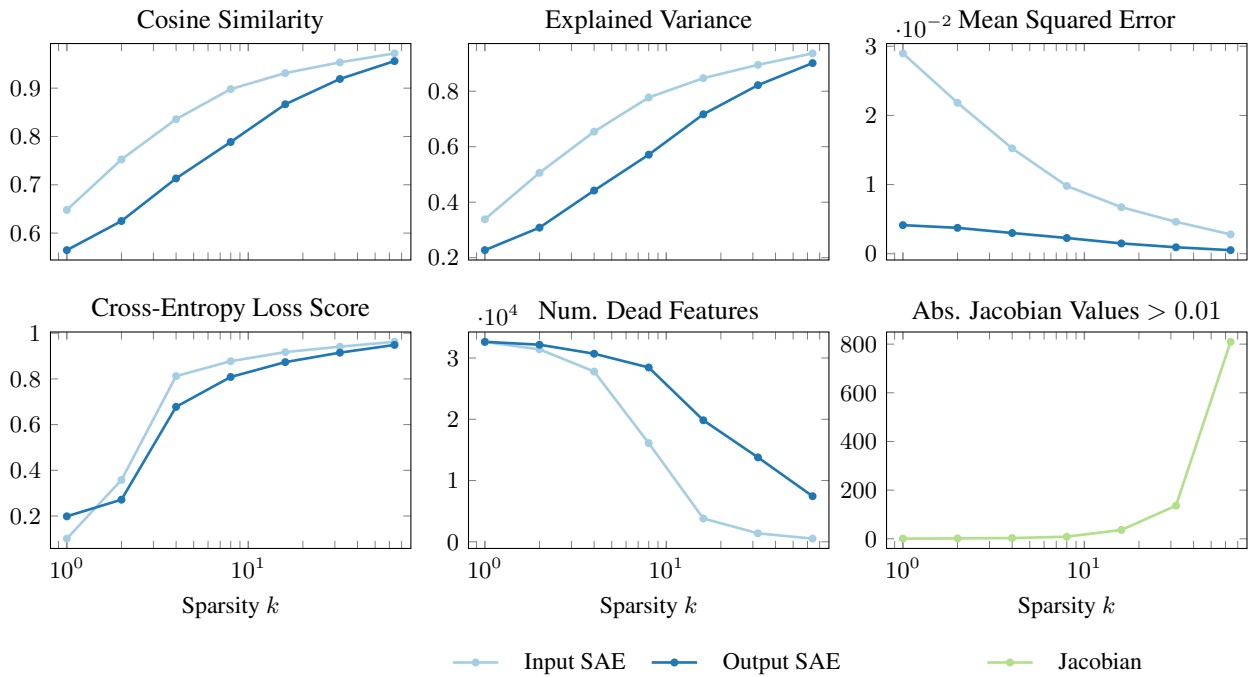

Figure 25. Reconstruction quality, model performance preservation, and sparsity metrics against the $k$ largest latents to keep in the TopK activation function. Here, we consider Jacobian SAEs trained on the feed-forward network at layer 3 of Pythia-70m with expansion factor $R = 64$. Recall that the maximum number of non-zero Jacobian values is $k^2$. The reconstruction quality and cross-entropy loss score improve as $k$ increases, and the number of dead features decreases. See Appendix E for details of the evaluation metrics.

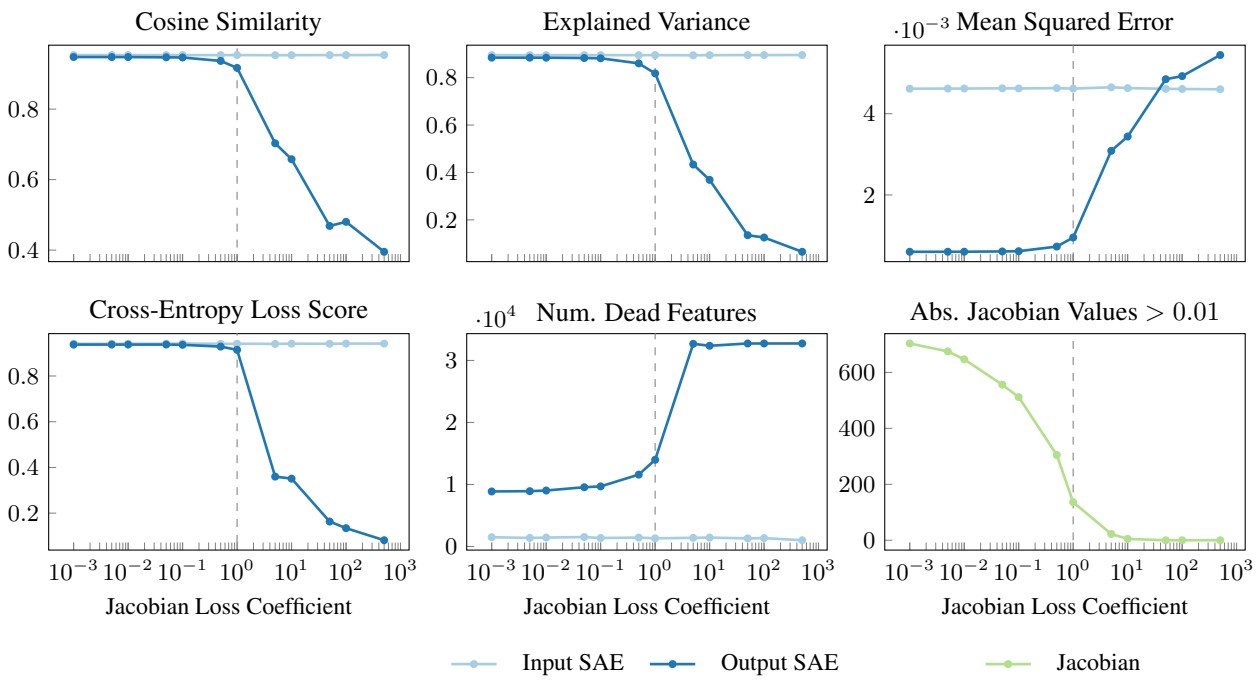

Figure 26. Reconstruction quality, model performance preservation, and sparsity metrics against the Jacobian loss coefficient. Here, we consider Jacobian SAEs trained on the feed-forward network at layer 3 of Pythia-70m with expansion factor $R = 64$ and $k = 32$. Recall that the maximum number of non-zero Jacobian values is $k^2 = 1024$. In accordance with Figure 5, all evaluation metrics degrade for values of the coefficient above 1. See Appendix E for details of the evaluation metrics.

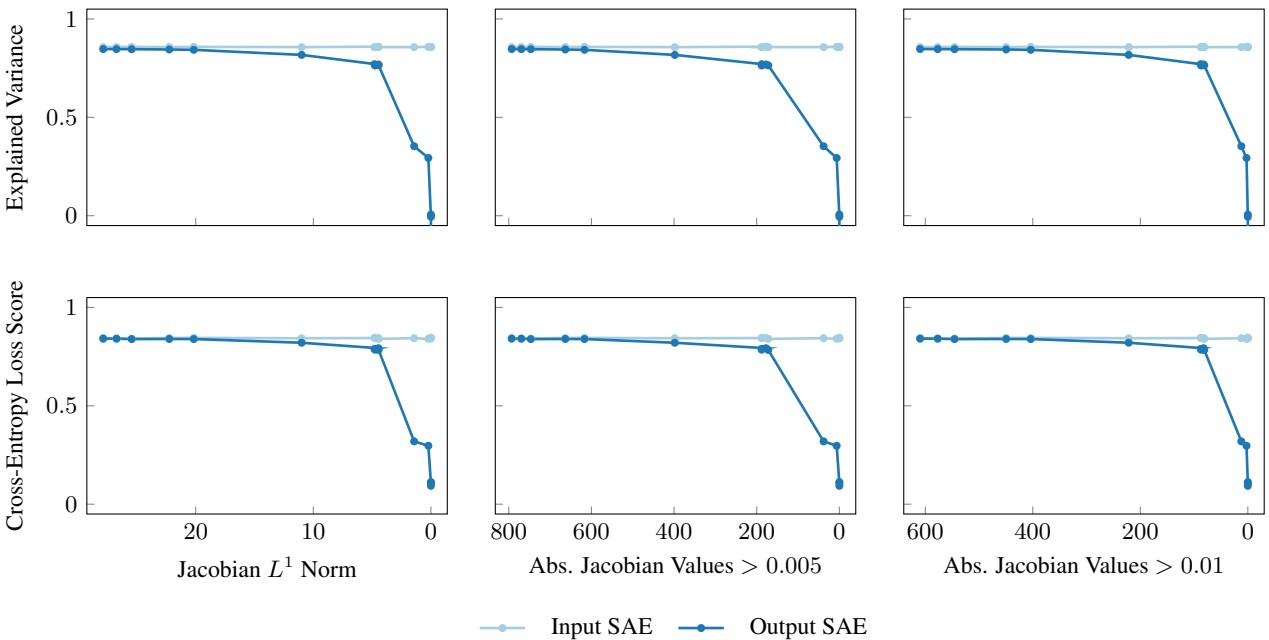

*Figure 27.* Pareto frontiers of the explained variance and cross-entropy loss score against different sparsity measures when varying the Jacobian loss coefficient. Here, we consider Jacobian SAEs trained on the feed-forward network at layer 3 of Pythia-70m with expansion factor $R = 64$ and $k = 32$. Recall that the maximum number of (dead) latents is 32768 (64 times the model dimension 512), and the maximum number of non-zero Jacobian values is $k^2 = 1024$. See Appendix E for details of the evaluation metrics.

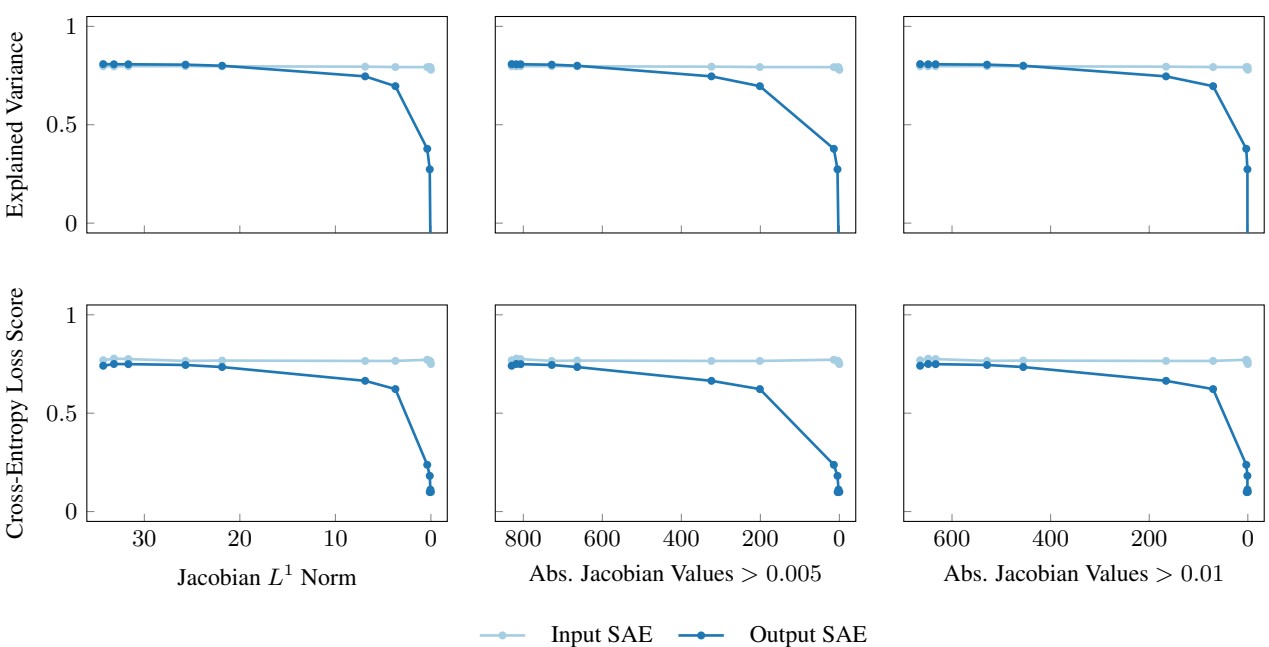

*Figure 28.* Pareto frontiers of the explained variance and cross-entropy loss score against different sparsity measures when varying the Jacobian loss coefficient. The coefficient has a relatively small impact on the reconstruction quality and sparsity of the input SAE, whereas it has a large effect on the sparsity of the output SAE and elements of the Jacobian matrix. Here, we consider Jacobian SAEs trained on the feed-forward network at layer 7 of Pythia-160m with expansion factor $R = 64$ and $k = 32$. Recall that the maximum number of (dead) latents is 49152 (64 times the model dimension 768), and the maximum number of non-zero Jacobian values is $k^2 = 1024$. See Appendix E for details of the evaluation metrics.

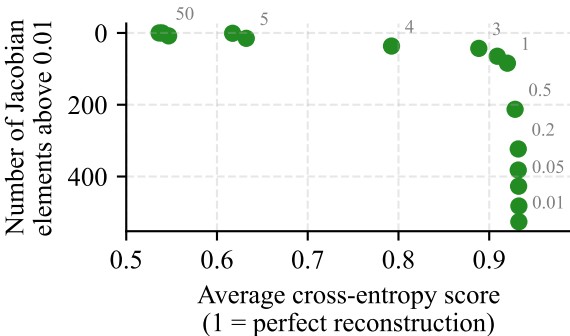

*Figure 29.* The trade-off between reconstruction quality and Jacobian sparsity as we vary the Jacobian loss coefficient. Each dot represents a pair of JSAEs trained with a specific Jacobian coefficient. Measured on layer 3 of Pythia-70m with $k = 32$.

We speculate that it is necessary to tune our hyperparameters for each layer individually to achieve improved performance; see, for example, Figures 26 and 4 for the variation of our evaluation metrics against the coefficient of the Jacobian loss term for individual layers of Pythia-70m and 160m.

## F. More data on Jacobian sparsity

In Figure 23 we showed that Jacobians are much more sparse with JSAEs than traditional SAEs. To this end, we provided a representative example of what the Jacobians look like with JSAEs vs traditional SAEs. Some readers may object that this is not an apples-to-apples comparison since JSAEs are optimizing for lower L1 on the Jacobian, so it may be the case that JSAEs merely induce Jacobians with smaller elements, but their distribution may still be the same. To address this criticism, the examples are L2 normalized; we provide un-normalized versions as well as L1 normalized versions of the example Jacobians in Figure 30. We also provide a histogram and a CDF of the distribution of absolute values of Jacobian elements in Figure 32, which is taken across 10 million tokens.

### F.1. Jacobian norms

In this section, we address an objection we expect some readers will have to our measures of sparsity. Our main metric for sparsity is the percentage of elements with absolute values above certain small thresholds (e.g. Figure 2). However, one can imagine two distributions with the same degree of sparsity, but vastly different results on this metric due to a different standard deviation. For instance, imagine two Gaussian distributions, both with $\mu = 0$ but with significantly different standard deviations, $\sigma_1 \gg \sigma_2$. They would score very differently on our metric, but their degrees of sparsity would not be meaningfully different (since sparsity requires there to be a small handful of relatively large elements). Since our $L_1$ penalty encourages the Jacobians to be smaller, it could be that they simply become more tightly clustered around 0. However, this is not the case. We can measure this by looking at the "norms" of the Jacobian, i.e. we flatten the Jacobian, treat it as a vector, and compute its $L_p$ norms. If the Jacobian is merely becoming smaller, we would expect all of its $L_p$ norms to decrease at roughly the same rate. On the other hand, if the Jacobian is becoming sparser, we would expect its $L_1, L_2$ norms to decrease while its $L_4, \ldots, L_\infty$ norms, which depend more strongly on the presence or absence of a few large elements, should stay roughly the same. We present these results in Figure 35, as we can see, the Jacobian does become slightly smaller, but most of the effect we see is indeed the Jacobian becoming significantly more sparse.

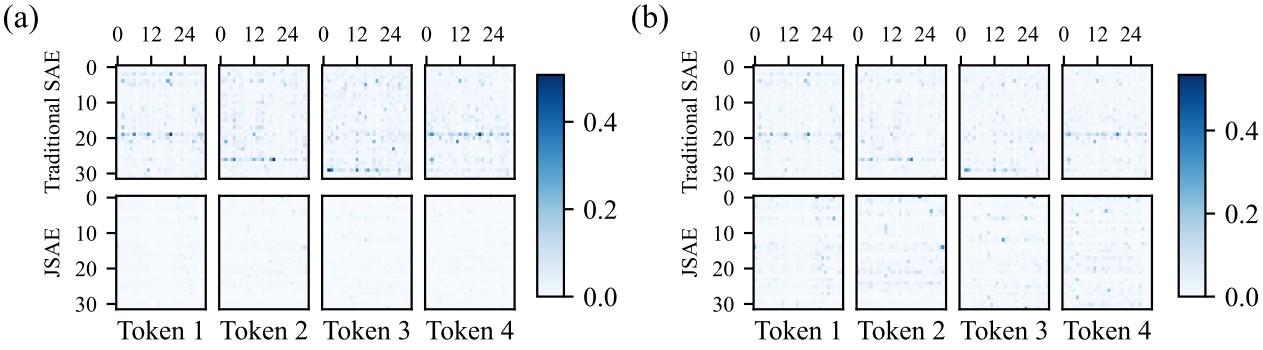

*Figure 30.* Comparison of Jacobians from traditional SAEs vs JSAEs, same as Figure 2 but with different normalization. (a) Not normalized. (b) L2 normalized. Measured on layer 15 of Pythia-410m.

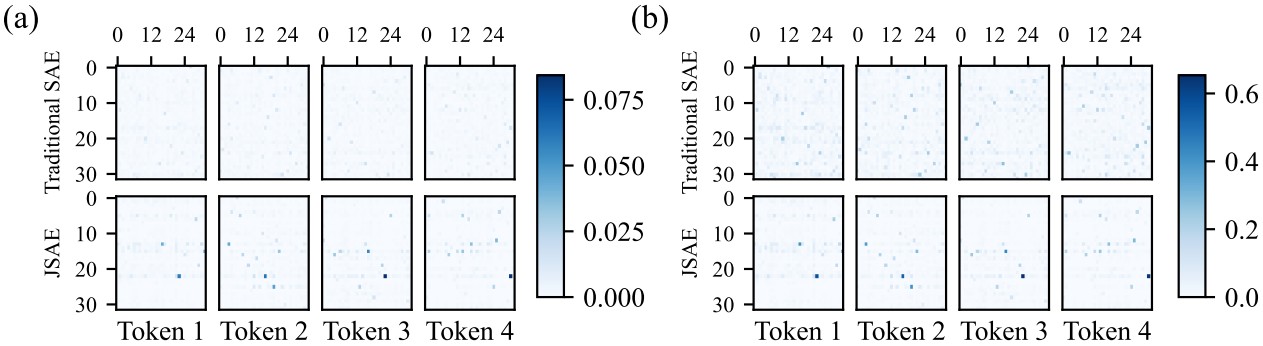

*Figure 31.* Comparison of Jacobians from traditional SAEs vs JSAEs, same as Figure 2 but with different normalization. (a) L1 normalized. (b) L2 normalized. Measured on layer 3 of Pythia-70m.

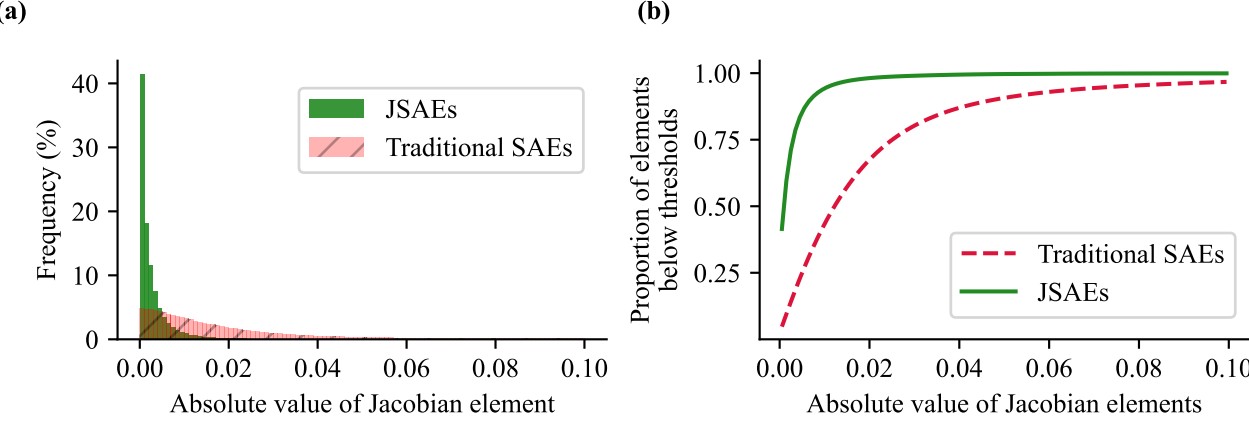

*Figure 32.* Further data showing that JSAEs induce much greater Jacobian sparsity than traditional SAEs. (a) A histogram of the absolute values of Jacobian elements in JSAEs versus traditional SAEs. JSAEs induce significantly more sparse Jacobians than standard SAEs. This means that there is a relatively small number of input-output feature pairs which explain a very large fraction of the computation being performed. Note that only the $k \times k$ elements corresponding to active latents are included in the histogram – the remaining $(n_y - k) \times (n_x - k)$ elements are zero by definition both for JSAEs and standard TopK SAEs. The histogram was collected over 10 million tokens from the validation subset of the C4 text dataset, which produced 10.24 billion feature pairs. (b) The cumulative distribution function of the absolute values of Jacobian elements, again demonstrating that JSAEs induce significantly more computational sparsity than traditional SAEs. Measured on layer 15 of Pythia-410m.

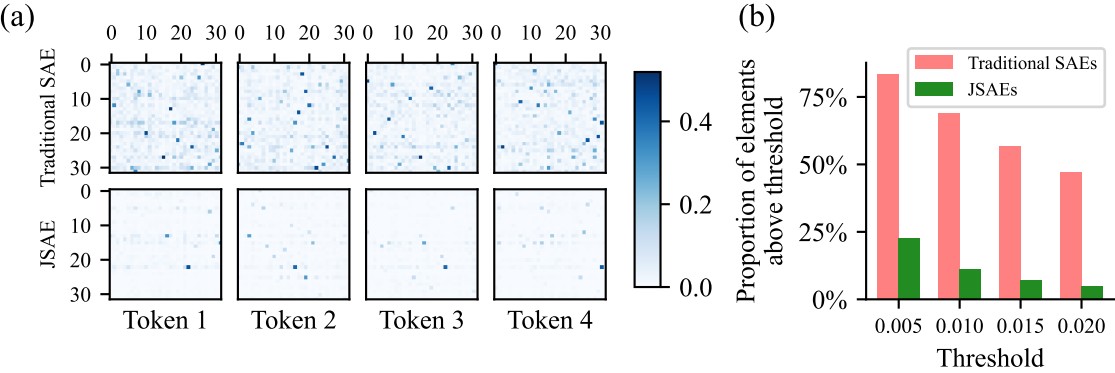

*Figure 33.* JSAEs induce a much greater degree of sparsity in the elements of the Jacobian than traditional SAEs. Identical to Figure 2 but measured on layer 3 of Pythia-70m.

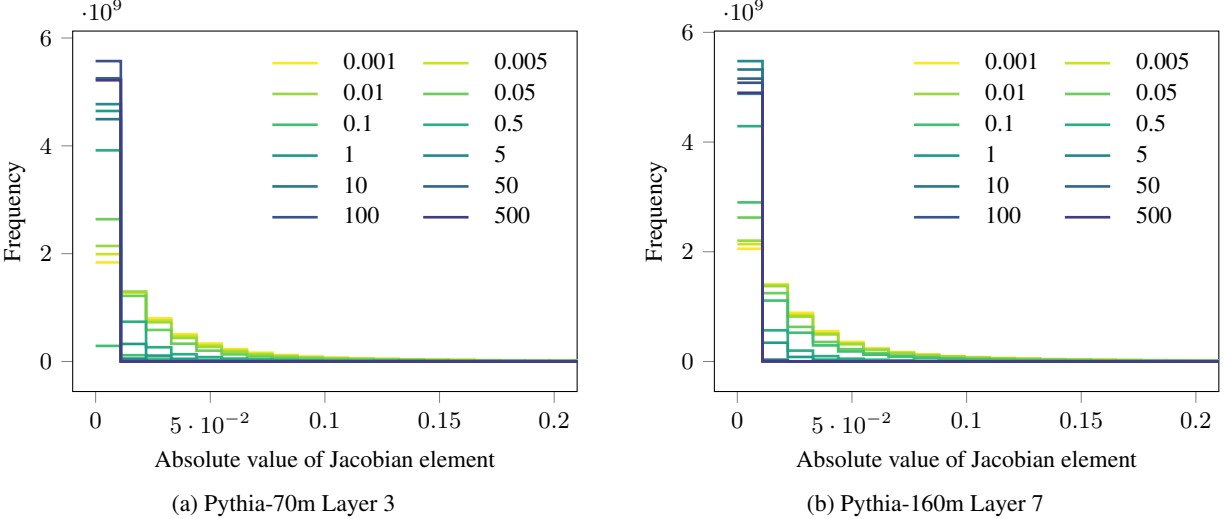

*Figure 34.* Histograms that show the frequency of absolute values of non-zero Jacobian elements for different values of the coefficient of the Jacobian loss term. As the coefficient increases, the frequency of larger values decreases, i.e., the Jacobian becomes sparser. We provide further details in Figure 32.

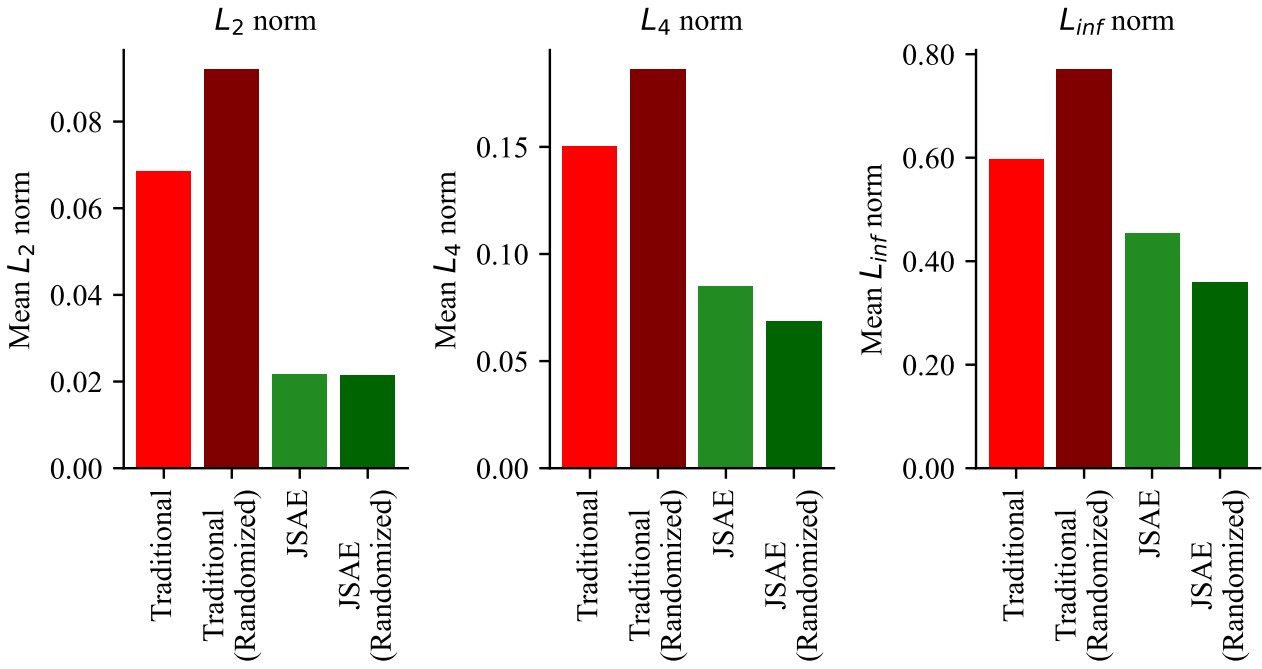

*Figure 35.* $L_p$ norms of the Jacobians. We measure these by flattening the Jacobians and treating them as a vector. These results imply that the Jacobians are in fact becoming more sparse, as opposed to merely becoming smaller (see Section F.1). Averaged across 1 million tokens, measured on layer 3 of Pythia-70m.

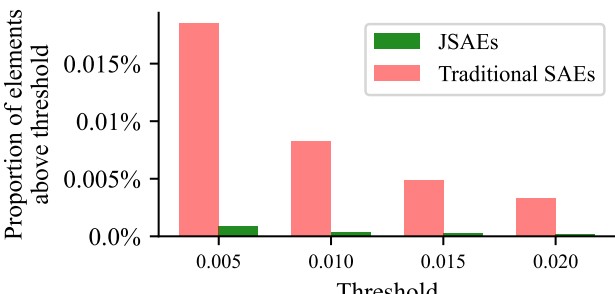

*Figure 36.* The Jacobians aren't only sparse locally (i.e. on each token in each prompt), but also globally (i.e. when averaged across many tokens), much more so than with traditional SAEs. In particular, here we consider the full $n_y \times n_x$ Jacobian (i.e. not slicing based on the TopK), which we average across 10 million tokens ($\frac{1}{N} \sum_{\text{prompt,token}} \mathbf{J}$) before considering its summary statistics. This is an important measure as it confirms that the connections found by JSAEs are indeed sparse in a global sense, not just when conditioning on a specific model input. Measured on layer 15 of Pythia-410m. Note that the small numbers on the y-axis are due to the fact that, unlike in e.g. Figure 2, here we set 100% to be $n_y \times n_x$ rather than $k \times k$. We also note that for each element in the Jacobian, we are only taking the average over the tokens on which the corresponding output SAE latent is selected by the TopK activation function (i.e. when at least one element in the row of the Jacobian is nonzero); this is important because otherwise this measure would significantly conflate the sparsity of the Jacobian itself with the sparsity of the activations of each individual latent.

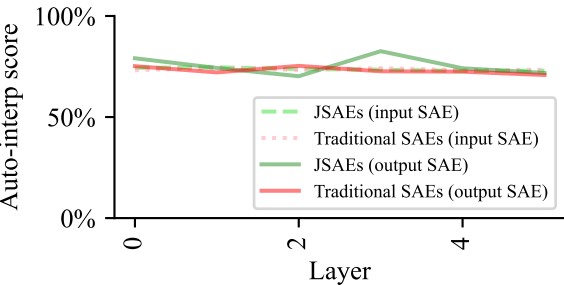

*Figure 37.* Automatic interpretability scores of JSAEs are very similar to traditional SAEs. Measured on all layers of Pythia-70m using the "fuzzing" scorer from Paulo et al. (2024).

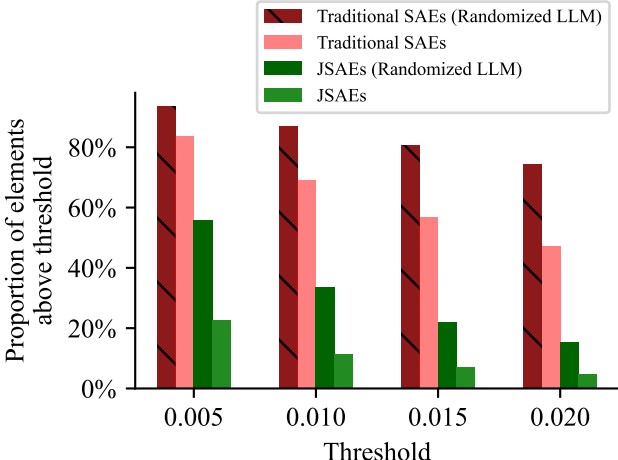

*Figure 38.* Jacobians are substantially more sparse in pre-trained LLMs than in randomly initialized transformers. This holds both when you actively optimize for Jacobian sparsity with JSAEs, and when you don't optimize for it and use traditional SAEs. The proportion of Jacobian elements with absolute values above certain thresholds. The figure shows the proportion of Jacobian elements with absolute values above certain thresholds. Identical to Figure 7 but measured on layer 3 of Pythia-70m.

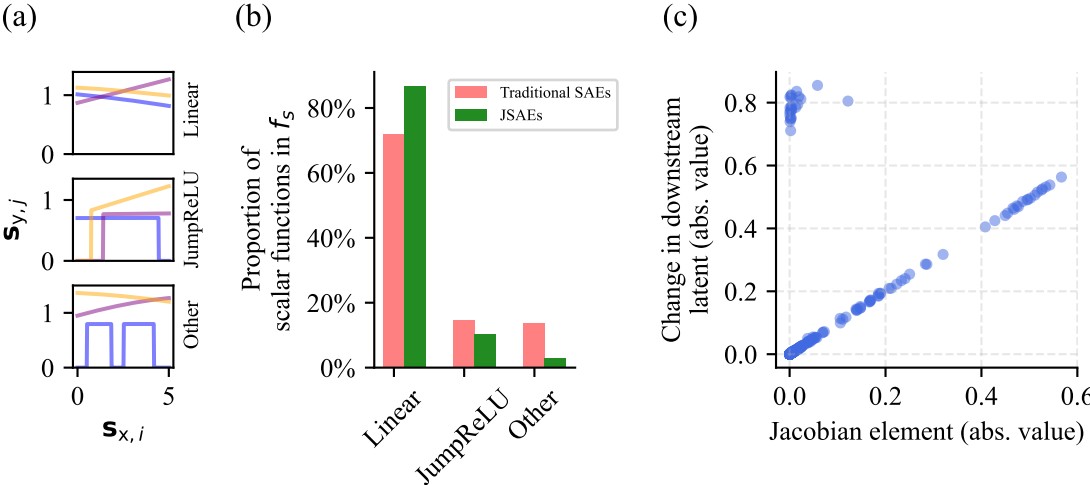

*Figure 39.* The function $f_s$, which combines the decoder of the first SAE, the MLP, and the encoder of the second SAE, is mostly linear. Identical to Figure 8 but measured on layer 3 of Pythia-70m.

