# OpenReview forum: "Jacobian Sparse Autoencoders: Sparsify Computations, Not Just Activations"
_ICML.cc/2025/Conference — ICML 2025 poster_

### Official Review · Reviewer_PVaJ · 2025-02-27

**Overall Recommendation:** 3

**Summary:**

**Summary**

Sparse Autoencoders (SAEs) help interpret latent activations in LLMs but do not explicitly reveal how computations are performed. This paper extends SAEs to study the sparsity of the computational transformations within MLP layers of transformer-based LMs. The authors train two SAEs—one before and one after an MLP transformation—and enforce sparsity in the Jacobian of the input-output mapping. This is done via an $\ell_1$ penalty on the Jacobian, encouraging learned features to be sparse linear functions of each other. To make this tractable, the paper introduces optimizations leveraging the inherent sparsity of SAE latents. Main results: (a) JSAEs induce Jacobian sparsity without significantly degrading reconstruction quality, (b) Trained LMs exhibit greater computational sparsity than randomly initialized ones, indicating that sparsity is a learned property, (c) Jacobian sparsity is a reasonable proxy for computational sparsity since the JSAE+MLP mapping is approximately linear.

**Strengths**
- JSAEs extend recent SAE-based interpretability techniques from analyzing activations to analyzing computations, providing a more structured view of how LLMs process information. The problem is formulated in a clean and well-motivated way. The paper also does a great job at differentiating JSAEs from standard SAEs and transcoders, clearly explaining the conceptual differences in what each method sparsifies.
- The proposed optimizations make Jacobian regularization computationally feasible for large models, where naive Jacobian computation would be intractable. JSAEs only increase compute by ~2x compared to standard SAEs.
- Evaluation metrics demonstrate minimal trade-offs between sparsity and reconstruction fidelity, suggesting that the Jacobian penalty does not significantly degrade representation quality.
- Strong empirical results show that JSAEs extract meaningful structure rather than fitting noise, as demonstrated in the random initialization ablation. Trained LMs exhibit far greater Jacobian sparsity than randomly initialized ones, suggesting that JSAEs capture properties learned during training.

**Weaknesses**
- The paper provides a tool to study MLP computaion but does not use it to provide new insights into MLP computation. For example, the work would be stronger with case studies showing what JSAEs reveal about computation: Can JSAEs recover known mechanistic circuits in toy models? Do they expose meaningful functional structures in large LMs? How would an interpretability researcher use JSAEs to answer practical questions about LLM internals?
- The study focuses on reconstruction and sparsity metrics, but does not evaluate whether JSAE features are useful for real-world interpretability tasks. The study should use recent benchmarks for SAEs to incorporate additional metrics. AxBench (https://arxiv.org/abs/2501.17148) evaluate SAE features for concept detection and model steering and SAEBench (https://www.neuronpedia.org/sae-bench/info) provides structured metrics for assessing interpretability and feature usefulness. Comparing JSAEs to these benchmarks would make the empirical results more convincing.
- The method is specialized for MLP modules and does not consider attention mechanisms, residual streams, or full transformer computations. It focuses on single MLP layers, making it unclear how JSAEs scale to studying entire transformer models with hundreds of MLPs. This limitation suggests that JSAEs may be more suited for local rather than global interpretability.
- Missing related work on interpretability techniques that focus on model computation. Shah et al. (https://arxiv.org/abs/2404.11534) uses linear surrogate models to analyze how model components (e.g., conv filters in vision models, MLPs) in a trained neural network contribute to final model outputs. Similarly, Balasubramanian et al. (https://arxiv.org/abs/2406.01583) uses attribution techniques to quantify the role of individual components in shaping vision model representations.

**Claims And Evidence:**

Yes. Please see my review above

**Essential References Not Discussed:**

Please see my review above

**Experimental Designs Or Analyses:**

Please see my review above

**Methods And Evaluation Criteria:**

Please see my review above

**Other Comments Or Suggestions:**

Please see my review above

**Other Strengths And Weaknesses:**

Please see my review above

**Questions For Authors:**

Please see my review above

**Relation To Broader Scientific Literature:**

Please see my review above

**Theoretical Claims:**

Yes, the appendix on efficiently computing MLP Jacobians.

---

> ### Author Rebuttal · Authors · 2025-04-01
>
> Thank you for your positive comments and useful recommendations.
>
>
> ### Qualitative results and insights
>
> We have updated the manuscript with qualitative [results](https://anonymous.4open.science/r/jacobian-saes-icml-D7BF/jacobian-saes-icml-examples.pdf). Specifically, we plot max-activating examples for an output feature, and the 16 input features that connect to it most strongly. The very first output latent (34455) in that document that activates fairly generally on German text. The input latents respond to:
> * Tokens that frequently appear in German text.  There is an input that mainly responds to "Pf", and another that responds to "sch", and another that responds to "Kle".
> * "German"
> * "von"
> * Place names such as "Austria" and "Berlin"
>
> There are also far more subtle examples, such as output latent 52843 (easiest to Ctrl-F). This latent is activated by "go" and "goes" in different figurative contexts, including "go on record" and "go behind his back"; it does not appear to respond to the literal meaning of movement through space. This distinction may be part of what this circuit is designed to clarify. One of the corresponding inputs include measure of extent and distance, like "a long way" and "right to the 6 th floor" and another to the word "back" in "go back". Other input latents include "go" in a figurative context, or words that have a similar role to "go" such as "proceed", "headed" and "come" and even one responding to "GO" or "GOTO" in computer code.
>
> Note that the output features are not hand-picked, but are ordered starting with the output with the largest element of the Jacobian averaged across all inputs.
>
> ### Evaluation
>
> These benchmarks were not available on the ICML submission date ([1] was published in March, and [2] was published in very late January).
> As such, we plan to run them for the camera ready.
>
> [1] https://arxiv.org/abs/2503.09532
>
> [2] https://arxiv.org/abs/2501.17148
>
> ### Other model components
>
> We agree that we used JSAEs only on MLPs, i.e. in the local interpretability setting, as it was a useful starting point to introduce the JSAE methodology.
> We note that other interpretability methods, e.g. SAEs and transcoders, were also initially developed only on a single MLP at a time.
>
> To understand out approach to moving towards using JSAEs to obtain a global understanding of LLMs, it is worth stepping back and thinking an overall approach in the interpretability literature, of which the JSAE forms a part. Perhaps the most exciting recent work following this approach is the Anthropic circuit tracing paper, which was published only a few days ago, and cites our work.
> The key idea in our work and [3] is:
> * Take multiple layers in network.
> * Decompose each layer into sparse latents (we use SAEs at different layers, Anthropic use transcoders at different layers).
> * Take the Jacobian for the mapping from sparse latents at one layer to sparse latents at later layers.
> * Identify strong connections using large elements in that Jacobian.
> * Interpret those connections (which is much _much_ easier if the connections are sparse).
>
> In any such approach, whether local (our current work) or global ([3]), the Jacobians are denser than they could be, simply because we don't have a term in the loss that explicitly encourages them to be sparse.
> Our key insight was that it is actually possible to develop an efficient loss term that sparsifies Jacobians.
> While we initially only applied this loss term to MLPs, the overall approach certainly could be applied globally, indeed, we are super-excited about further sparsifying the Jacobians between transcoders latents studied in [3] to get even sparser, more interpretable global circuits.
>
> [3] https://transformer-circuits.pub/2025/attribution-graphs/methods.html
>
> ### Additional Related work
>
> Thanks!  We have added this to the Related Work section in our working manuscript.
>
> ### Conclusion
>
> Thank you for generously outlining the paper's strengths in your original review.
> We hope this response (and especially the new qualitative results) have addressed your key concerns.
> If so, we would greatly appreciate it if you would reconsider your score.

---

> > ### Comment · Reviewer_PVaJ · 2025-04-01
> >
> > Thanks for addressing some of my concerns. I have increased my score.

---

> > > ### Author Response · Authors · 2025-04-06
> > >
> > > Thanks so much for carefully considering our rebuttal and increasing your score!
> > >
> > > We hope that our new qualitative results have addressed the main concern of all the reviewers. However, ZBhh has just ticked the box without commenting on the rebuttal, and has not changed their score from a 1.  Therefore, the paper will likely need an extra bump to get accepted.
> > >
> > > If you think the paper should be accepted, we were wondering whether you'd give us that bump by again considering your score?

---

### Official Review · Reviewer_M6HE · 2025-03-13

**Overall Recommendation:** 4

**Summary:**

The authors introduce Jacobian SAEs, as a form of dictionary learning that also facilitates circuit analysis by creating sparse computational graphs. They jointly train SAEs on both input and output to an MLP layer, and make the Jacobian sparse by adding its L1 norm to the loss function.

Though computing and differentiating the Jacobian is extremely complicated naïvely, the process is made tractable by analytically computing the Jacobian, leveraging the simple mathematical form of an MLP. Only standard MLPs studied, not the gated MLPs used in most common most modern models. Since only the k active latents matter, the operation only doubles the cost in the context, though the exact computational complexity as models scale isn't explored.

The most relevant prior work is transcoders which replace MLP layers with a wider sparser replacement. The authors argue that Jacobian SAEs are distinct as transcoders sparsify computation while their method sparsifies input and output.

To show JSAE effectiveness, they demonstrate:

-   While Jacobian sparsity regularization degrades other metrics there's a sweet spot with reasonable performance and better sparsity
-   Cross-entropy and explained variance are comparable to normal SAEs
-   Auto-interpretability scores are comparable to normal SAEs
-   They find sparser Jacobians with trained JSAEs on trained transformers than randomly initialised ones

To validate the approach of studying gradients to understand the computation of the MLP layer, on a given prompt they vary each SAE input latent to see effects on each output latents. They show that 88% of relationships are linear, many are ReLU like, and some are more complex

**Claims And Evidence:**

See below

**Essential References Not Discussed:**

No

**Experimental Designs Or Analyses:**

See below

**Methods And Evaluation Criteria:**

See below

**Other Comments Or Suggestions:**

## Major Comments
*The following are things that, if adequately addressed, would increase my score*

Summary; I overall consider this paper to be a weak accept. I think it provides some innovative new ideas, in particular using Jacobians as a way to measure the sparsity of a computational graph and tricks for how to compute and train this well. However, I don't feel that the authors provide a compelling enough argument for why one would prefer these over prior techniques like transcoders. If I were convinced there were compelling use cases where one would prefer JSAEs, I would be a lot more excited about this paper.

1. The paper doesn't clearly explain why I should prefer this method over transcoders. Transcoders already provide sparsity in their output, as the output is essentially a sparse linear combination of the decoder vectors from transcoder units. The main difference seems to be that transcoders don't explicitly sparsify the input, only the output. However, it’s unclear why sparsifying the input is especially valuable or significant. The input vector of a transcoder latent can be projected onto a normal residual SAE's decoder to see which concepts are used, though perhaps that would have too much interference?
One potential advantage of JSAEs could be clearer representation in cases where multiple transcoder latents share identical decoders. JSAEs would represent this scenario more cleanly with a single output latent.
If any of these differences matter, the paper should explicitly explain why and ideally provide a concrete example.
2. More broadly, the paper doesn't clearly motivate why Jacobian SAEs are interesting or practically useful. I'm sympathetic to the idea that high-quality circuit analysis with interpretable nodes is valuable, and am excited about improved techniques there. However, if you're trying to argue that JSAEs improve interpretability compared to methods like sparse feature circuits on normal SAEs or transcoders, then clear empirical evidence is needed. For example, analyzing tasks or contexts studied in [Dunefsky et al.](https://arxiv.org/abs/2406.11944) or [Marks et al.](https://arxiv.org/abs/2403.19647) and explicitly showing that JSAEs give better or clearer insights would significantly strengthen your argument. Or, pick a task where MLPs matter like the greater-than task from [Hanna et al.](https://arxiv.org/abs/2305.00586) and concretely show how JSAEs provide new insights or clearer interpretations.
3. Also, the paper would greatly benefit from more qualitative analysis. Right now, the argument primarily relies on summary statistics, which can obscure what's actually happening underneath. It would help if you showed the correlation or cosine similarity between transcoder latents and output JSAE latents, or between JSAE and normal SAE latents. Are these methods capturing similar concepts but with slightly different directions, or is the JSAE method genuinely capturing fundamentally different computations to achieve sparse Jacobians?
4. Another concern is about potential pathologies: [Lindsey et al.](https://transformer-circuits.pub/2024/crosscoders/index.html) demonstrated substantial redundancy and showed concept directions in the residual stream shifting pre- and post-MLP. My hypothesis is that this occurs because MLPs have a significant first order linear component that explains a fair amount of the variance in the output. Consequently, if this is true, achieving a sparse Jacobian might just reflect input and output SAEs capturing essentially the same concepts, with the MLP performing a simple linear transformation between them. This would result in a clean and sparse Jacobian, especially if the input features are nearly orthogonal. However, this scenario would not be particularly interesting or insightful.
5. A suggestion for improving this approach: inspired by [skip transcoders](https://arxiv.org/abs/2501.18823), if part of the issue is the MLP's output having a large linear component, why not explicitly train a linear approximation, subtract it off, and then apply your JSAE to the nonlinear remainder? This way, your Jacobian sparsity would more meaningfully represent non-linear computations rather than simple linear mappings. This might also simplify interpretation, as a purely linear output essentially extends the input SAE, which isn't particularly interesting and wastes capacity in the output SAEs.
6. Another concern: the paper implicitly mixes two distinct notions of sparsity—local sparsity (how each concept affects each output on a single prompt) and global sparsity (connections existing across all prompts). You might have an output latent activated by many different input latents but on any given prompt only fires due to a single input latent as they're all anti-correlated. Locally, that's sparse, but globally, it's dense. Currently, the paper only looks at sparsity on individual prompts. It would be helpful to see global statistics—for instance, looking at multiple prompts and measuring the probability that a given edge exists conditional on the output latent activating, then assessing sparsity at this global scale. My intuition is that it might still be somewhat sparse globally, but clarifying and verifying this explicitly is worthwhile.



## Minor Comments
*The following are unlikely to change my score, but are comments and suggestions that I hope will improve the paper, and I leave it up to the authors whether to implement them. No need to reply to all of them in the rebuttal*

1. I do like the trick used for efficiently computing Jacobians as a proxy for sparse computational graphs. It's clever and seems to be executed effectively.
2. Only studying GELU MLPs is somewhat limiting, as gated MLPs are now the standard, so this work is not directly applicable to modern LLMs. While your methods should transfer fine (albeit with more complicated derivative calculations), adding an appendix discussing this extension would help make the paper feel more complete.
3. Regarding your mention of feature absorption in the intro, it's not clear at all that JSAEs address this issue effectively. When concepts consistently co-occur, like "elephant" and "starts with E," they typically merge into something like "starts with E but isn't elephant." I don’t see why this would differ significantly in JSAEs. If this isn't a key claim, it might be better either to justify it clearly or just cut it.
4. The summary statistics about relationships between input/output latents have an unclear relationship to linearity. Since near-zero Jacobian values imply trivial linearity, it would help to filter first for active edges to better highlight meaningful relationships. The paper mentions that 88% of relationships are linear, but it isn't clear whether this percentage is notably high or low since I don't have good context for these numbers. Clarifying whether this is good or bad and providing context for this number by comparing to eg the fraction of trivial entries in the Jacobian, would significantly help interpretation.

**Other Strengths And Weaknesses:**

See below

**Questions For Authors:**

See above

**Relation To Broader Scientific Literature:**

See below esp re transcoders

**Theoretical Claims:**

N/A

---

> ### Author Rebuttal · Authors · 2025-04-01
>
> Thanks for your careful and considered review!
>
> ### 1. Transcoders vs JSAEs
>
> A few days ago, Anthropic published a huge new paper on circuit tracing, and which cites our work [1].
> The key idea in our work and [1] is:
> * Take multiple layers in network.
> * Decompose each layer into sparse latents (we use SAEs at different layers, [1] uses transcoders at different layers).
> * Take the Jacobian for the mapping from sparse latents at one layer to sparse latents at later layers.
> * Identify strong connections using large elements in that Jacobian.
> * Interpret those connections (which is much _much_ easier if the connections are sparse).
>
> Our insight is that in any such approach, the Jacobians are denser than they could be because we don't have an efficient loss term that explicitly encourages Jacobian sparsity.
> We introduce such a loss term.
> Of course, the most natural place to apply such methods is to SAEs at different points in the network, as we did in our paper.
> But you could equally extend our idea of sparsifying Jacobians to the transcoders at different layers in [1] to get even sparser, more interpretable circuits.
>
> > The input vector of a transcoder latent can be projected onto a normal residual SAE's decoder to see which concepts are used, though perhaps that would have too much interference?
>
> We agree you could do this, but based on our results, we would expect that the resulting projection would be dense, (or at least denser than it could be) and therefore difficult to interpret.  We will check for the camera-ready.
>
> [1] www.transformer-circuits.pub/2025/attribution-graphs/methods.html
>
> ### 2. Toy tasks
>
> We have not been able to run toy tasks during the short rebuttal period, but will explore this for the camera-ready.
>
> ### 3. Qualitative results
>
> We have updated the manuscript with qualitative [results](https://anonymous.4open.science/r/jacobian-saes-icml-D7BF/jacobian-saes-icml-examples.pdf). Specifically, we plot max-activating examples for an output feature, and the 16 input features that connect to it most strongly. The very first output latent (34455) in that document that activates fairly generally on German text. The input latents respond to:
> * Tokens that frequently appear in German text.  There is an input responding "Pf", another responding to "sch", and another responding to "Kle".
> * "German"
> * "von"
> * Place names such as "Austria" and "Berlin"
>
> There are also far more subtle examples, such as output latent 52843 (easiest to Ctrl-F). This latent is activated by "go" and "goes" in different figurative contexts, including "go on record" and "go behind his back"; it does not appear to respond to the literal meaning of movement through space. This distinction may be part of what this circuit is designed to clarify. One of the corresponding inputs include measure of extent and distance, like "a long way" and "right to the 6 th floor" and another to the word "back" in "go back". Other input latents include "go" in a figurative context, or words that have a similar role to "go" such as "proceed", "headed" and "come" and even one responding to "GO" or "GOTO" in computer code.
>
> Note that the output features are not hand-picked, but are ordered starting with the output with the largest element of the Jacobian averaged across all inputs.
>
> It's important to highlight that, as far as we can see, you could not get this kind of insight using transcoders of standard SAEs. Our qualitative results show that the MLP computes its output feature using _a specific set of input features_; transcoders and standard SAEs do not attempt to locate the input features used to compute a specific output feature.
>
> ### 4. Pathologies
>
> The qualitative examples above usually show one input feature that's similar, but a lot of other input features that are different. So the pathology you are suggesting does seem to occur, but it doesn't seem to obscure other interesting patterns.
>
> ### 5. Skip transcoders
>
> This is a great idea! This is definitely worth trying to see whether it makes things more interpretable. We will run analysis for the camera-ready deadline.
>
> ### 6. Sparsity
>
> We agree that our current manuscript could be clearer on this point and we will clarify it in the camera-ready! We did check the global sparsity levels and your intuition is correct, it is indeed quite sparse globally. We will include data on this in the camera-ready.
>
> ### Gated MLPs
>
> We've already worked out the math for this and will add an Appendix with experiments for the camera-ready
>
> ### Feature absorption
>
> We have cut the claim
>
> ### Linearity
>
> Agreed, we will add this nuance
>
> ### Conclusions
>
> Thank you for your extensive and excellent comments! Given the number of great suggestions you made, we hope you will forgive the need to explore several points in the camera-ready deadline. If this response (especially the new qualitative results) has nonetheless addressed your key concerns, we would greatly appreciate it if you would reconsider your score.

---

> > ### Comment · Reviewer_M6HE · 2025-04-03
> >
> > Thanks to the authors for their rebuttal. The qualitative data makes me much more reassured that the Jacobian SAEs are finding something genuinely interesting and interpretable, and I agree that this is meaningfully different from what standard transcoders can achieve. I would be excited to see the work applied to transcoders in the camera-ready version, but I understand this is unreasonable to expect within the rebuttal period. I also find the argument for how this could be applied to transcoders compelling and it makes me less concerned that there is insufficient value added on top of transcoders, since the techniques can be combined. Given both of these factors, I am increasing my score from 3 to 4.

---

### Official Review · Reviewer_WxFv · 2025-03-16

**Overall Recommendation:** 4

**Summary:**

This work introduces Jacobian Sparse Autoencoders (JSAEs), an extension of sparse autoencoders (SAEs) designed to sparsify not only latent activations but also the computational graph (approximated via the Jacobian) of an MLP layer in LLMs models.
Authors also show how to compute efficiently the Jacobian thanks to the top-k SAE approach (namely only the non-zero parts are computed), encouraging the Jacobian to be sparse. Their findings include:
- JSAEs significantly increase the sparsity of the Jacobian between input and output latent spaces compared to standard SAEs with minimal cost and small drop in reconstruction quality and model performance.
- Input and output latents learned  by Jacobian SAEs are approximately as interpretable as  standard SAEs, as quantified by auto-interpretability scores.
- When applied to randomly initialized transformers, JSAEs achieve much less Jacobian sparsity than when applied to pre-trained transformers. This suggests that Jacobian sparsity is a property of the trained model, indicating its potential as a tool to uncover learned computational structure, something latent sparsity alone may not capture.
- The function learned by the JSAE-MLP-JSAE pipeline is shown to be mostly linear, meaning the Jacobian is a reliable proxy for true computational sparsity in this context.

**Claims And Evidence:**

Yes, claims are well-supported by both theoretical derivations and experimental evidence.

**Essential References Not Discussed:**

Essential references are discussed. However, in Section 2 the authors should mention why SAEs are introduces in general, so the superposition hypothesis and polysemantic neurons.

**Experimental Designs Or Analyses:**

Yes, the setting is legit. However, the paper would benefit from more qualitatevely results (i.e. the standard mechanistic approach to identify what concepts or features the model has learned).

**Methods And Evaluation Criteria:**

Yes

**Other Comments Or Suggestions:**

Figure 1, write in the figure (not only in the caption) what $\tau_k$ is.

**Other Strengths And Weaknesses:**

### Strengths

- The idea of sparsifying the computation (not just the representation) is new and compelling. It nicely bridges concepts from automated circuit discovery and dictionary learning.

- Insightful findings for discovering sparse computation in LLMs.

- The papers is really well-written and organized (yes for me it's a strenght because it's not always the case, even for top conferences).

### Weaknesses

- Lack of qualitatevely results (i.e. the standard mechanistic approach to identify what concepts or features the (J)SAEs have learned).
- JSAEs are only trained on individual MLPs. This is fine as a starting point, but the paper lacks a discussion on how to extend this to full multi-layer transformer analysis.

**Questions For Authors:**

- Are the MLP $f$ parameters updated when training JSAEs?

- Figure 2, what is the Jacobian of traditional SAEs?

**Relation To Broader Scientific Literature:**

The paper builds directly on prior work on SAEs and contributes to sparsify not only the activations but also the computational graph.

**Theoretical Claims:**

Yes. The setups, loss, efficient computation of Jacobian and almost linearity of $f_s$ seems correct.

---

> ### Author Rebuttal · Authors · 2025-04-01
>
> Thanks for your positive review, which notes "The idea of sparsifying the computation (not just the representation) is new and compelling. It nicely bridges concepts from automated circuit discovery and dictionary learning.
> Insightful findings for discovering sparse computation in LLMs.
> The papers is really well-written and organized".
>
> ### Qualitative results
>
> We have updated the manuscript with qualitative [results](https://anonymous.4open.science/r/jacobian-saes-icml-D7BF/jacobian-saes-icml-examples.pdf). Specifically, we plot max-activating examples for an output feature, and the 16 input features that connect to it most strongly. The very first output latent (34455) in that document that activates fairly generally on German text. The input latents respond to:
> * Tokens that frequently appear in German text.  There is an input that mainly responds to "Pf", and another that responds to "sch", and another that responds to "Kle".
> * "German"
> * "von"
> * Place names such as "Austria" and "Berlin"
>
> There are also far more subtle examples, such as output latent 52843 (easiest to Ctrl-F). This latent is activated by "go" and "goes" in different figurative contexts, including "go on record" and "go behind his back"; it does not appear to respond to the literal meaning of movement through space. This distinction may be part of what this circuit is designed to clarify. One of the corresponding inputs include measure of extent and distance, like "a long way" and "right to the 6 th floor" and another to the word "back" in "go back". Other input latents include "go" in a figurative context, or words that have a similar role to "go" such as "proceed", "headed" and "come" and even one responding to "GO" or "GOTO" in computer code.
>
> Note that the output features are not hand-picked, but are ordered starting with the output with the largest element of the Jacobian across all inputs.
>
> ### Discussion of extending JSAEs to full multi-layer transformer analysis
>
> We agree that we applied JSAEs only to locally (to individual MLPs). This mirrors many other interpretability approaches, such as:
> * SAEs, which allow you to interpret the activations at a single location in the network
> * transcoders, which allow you to interpret the activations again for an MLP.
>
> The difference between JSAEs and these approaches is that there is a natural extension of JSAEs to "global interpretability".  In particular, you can train SAEs at many points in the network, and minimize the Jacobian for the mappings between latent activations in these SAEs.
> Indeed, we believe that this methodology could naturally be to further sparify the Jacobians between _transcoder latents_ in the fantastic work released by Anthropic only a few days ago [1].
>
> [1] https://transformer-circuits.pub/2025/attribution-graphs/methods.html
>
> We have added this discussion to our working draft.
>
> > Figure 1, write in the figure (not only in the caption) what  is.
>
> Thanks!  Fixed.
>
> > Are the MLP parameters updated when training JSAEs?
>
> No.  The underlying MLP parameters are fixed to those in pre-training.
> The only thing that is updated is the encoder/decoder of the SAEs.
> The new thing is including a term in the loss for the encoder/decoder of the SAE which encourages sparse Jacobians when we consider the mapping from sparse inputs to sparse outputs.
>
> > Figure 2, what is the Jacobian of traditional SAEs?
>
> Traditional SAEs still have a Jacobian. You can still train a traditional SAE at the input and output of an MLP, then look at the Jacobian from sparse inputs to sparse outputs. The difference between an SAE and a JSAE is simply in the training objective. Specifically, the JSAE objective includes a term that encourages sparse Jacobians, while the SAE does not. As such, the SAE is also equivalent in e.g. Figure 3 to a Jacobian Loss Coefficient of zero.
>
> ### Conclusions
>
> Thank you for generously outlining the paper's strengths in your original review. We hope this response (especially the new qualitative results) has addressed your key concerns. If so, we would greatly appreciate it if you would reconsider your score.

---

### Official Review · Reviewer_ZBhh · 2025-03-23

**Overall Recommendation:** 4

**Summary:**

This paper addresses the problem of better understanding computations in deep models, particularly LLMs. Recently sparse autoencoders (SAEs) have become popular as a tool to mechanistically understand a model, by decomposing features learnt at any layer into a sparse set of disentangled concepts. This work proposes Jacobian SAE (JSAE), which aims to additionally sparsify the computations by MLP layers in LLMs, by enforcing sparsity on connections between TopK latents at the input and output of the MLP, approximated using the Jacobian. Experimental evaluation is performed to show that JSAEs can learn to reconstruct features similar to traditional SAEs while having much sparser connections for the computations between the constituent input and output SAE. Sanity test experiments are also performed to show that what is learnt is meaningful, e.g. that the sparsity is better for JSAEs trained on a learnt MLP as opposed to a randomly initialized MLP.

## Update after rebuttal
Thank you for your response, particularly for providing examples. They look quite interesting, and it would be very helpful to have more examples (or some latent exploration tool) in the final paper. Regarding the question about zeroing out connections: I actually meant something even simpler---one could pick a small threshold, e.g. 1e-4, and set any weight in the JSAE below that threshold to zero _post-hoc_, and then compute the performance, to see if the weights below the threshold are actually meaningfully important. This was intended to be more of a sanity check to verify results from Figure 3.

Overall however I believe this is an interesting and useful paper, so I would like to change my score to accept.

**Claims And Evidence:**

Generally yes, except for concerns raised in the Weaknesses section below.

**Essential References Not Discussed:**

None that I am aware of.

**Experimental Designs Or Analyses:**

The experimental design and analyses appear sound to the extent they have been done. However, additional evaluation is needed, as discussed in the Weaknesses section below.

**Methods And Evaluation Criteria:**

The method makes sense. The evaluations are reasonable but inadequate, as discussed in the Weaknesses section below.

**Other Comments Or Suggestions:**

- L048: $L^1 \to L_1$

**Other Strengths And Weaknesses:**

## Strengths
1. The paper looks into an interesting and potentially useful problem, of understanding computations made by large models, as opposed to just examining what features were learnt as is typically done with SAEs.
2. The proposed idea of learning sparse connections using the Jacobian is interesting and appears to be novel.
3. Experimental evaluation shows that the learnt JSAEs perform comparably to traditional SAEs while also being significantly sparser.
4. The paper is generally well written and easy to follow.


## Weaknesses
1. The biggest weakness is that there does not seem to be any example of a use case of such JSAEs. In typical SAEs, one can see examples of latents learning human interpretable features, along with evaluations, both qualitative and quantitative, to show that they are (for the most part) disentangled and meaningful (e.g. Bricken et al. 2023, Bills et al. 2023). Evaluations also often show downstream uses such as model steering. However, this work does not have any indication on what JSAEs could be used for at all. Since the stated motivation is to learn a sparse mapping between SAE concepts before and after an MLP, it would be critical to see what this learnt mapping actually encodes, particularly since the point of training such SAEs is to aid human interpretability. At the very least, one should have qualitative examples of concepts at the input, output, and weights connecting them, and ideally have a more thorough quantitative analysis.
2. The sparsity of JSAEs is shown by thresholding weights and counting how many are above the threshold. However, it is possible that the weights below this threshold still contribute meaningfully—it would be make more sense to evaluate reconstruction quality by explicitly zeroing out everything below the threshold and then computing the metrics in Figure 3. This would after all be the likely practical use of such trained JSAEs.

**Questions For Authors:**

Please refer to the Weaknesses section. While the idea and approach appear interesting, in particular Weakness 1 is a critical omission and needs to be addressed. Any discussion on this in the rebuttal would be helpful.

**Relation To Broader Scientific Literature:**

This work builds upon existing literature that aims to disentangle concepts learnt by deep models such as LLMs using SAEs. However, different from typical SAEs, it also focuses on sparsifying and understanding the computations in the LLM and not just features at a given layer. This is similar to the idea of transcoders, but not identical, as discussed in Section 2.2.

**Theoretical Claims:**

I looked over the claims in Appendix A and B but did not go through them in full detail. To that extent, they appear reasonable.

---

> ### Author Rebuttal · Authors · 2025-04-01
>
> Thank you for your thoughtful review, noting "(1) The paper looks into an interesting and potentially useful problem, of understanding computations made by large models, as opposed to just examining what features were learnt as is typically done with SAEs. (2) The proposed idea of learning sparse connections using the Jacobian is interesting and appears to be novel. (3) Experimental evaluation shows that the learnt JSAEs perform comparably to traditional SAEs while also being significantly sparser. (4) The paper is generally well written and easy to follow."
>
> ### Qualitative results
>
> We have updated the manuscript with qualitative [results](https://anonymous.4open.science/r/jacobian-saes-icml-D7BF/jacobian-saes-icml-examples.pdf). Specifically, we plot max-activating examples for an output feature, and the 16 input features that connect to it most strongly. The very first output latent (34455) in that document that activates fairly generally on German text. The input latents respond to:
> * Tokens that frequently appear in German text.  There is an input that mainly responds to "Pf", and another that responds to "sch", and another that responds to "Kle".
> * "German"
> * "von"
> * Place names such as "Austria" and "Berlin"
>
> There are also far more subtle examples, such as output latent 52843 (easiest to Ctrl-F). This latent is activated by "go" and "goes" in different figurative contexts, including "go on record" and "go behind his back"; it does not appear to respond to the literal meaning of movement through space. This distinction may be part of what this circuit is designed to clarify. One of the corresponding inputs include measure of extent and distance, like "a long way" and "right to the 6 th floor" and another to the word "back" in "go back". Other input latents include "go" in a figurative context, or words that have a similar role to "go" such as "proceed", "headed" and "come" and even one responding to "GO" or "GOTO" in computer code.
>
> Note that the output features are not hand-picked, but are ordered starting with the output with the largest element of the Jacobian averaged across all inputs.
>
> ### Zeroing-out connections with small Jacobians
>
> There are two different kinds of zeros in our Jacobians.
> * First, there are zeros in the Jacobian that arise due to most of the inputs and outputs being ``off'', as they are zeroed out by the k-sparse SAEs.  This sparsity is already accounted for in the reconstruction errors in the main text.
> * Second, there are small elements of the Jacobian even where the input and output features are on.
>
> Given our overall approach, this second kind of sparsity in the Jacobian is very difficult to zero-out.
> We chose a particular approach which has many advantages, but sadly does not allow for an analysis such as this.
> Specifically, we needed to choose between two potential approaches:
> * You could compute the Jacobian of the \textit{actual} MLP in the underlying network. This has the critical advantage of having no approximation error, as it is working with the Jacobian of the actual MLP.
> * You could train a function with a simple/sparse Jacobian to approximate the input/output behavior of the MLP. This has the disadvantage that there is now approximation error.  But on the other hand, this function can designed to be manipulated more easily.
>
> We chose the first approach, as it removes a source of approximation error (from the function you've trained to approximate the MLP).  But it does mean that you can't zero-out an element of the Jacobian, because this is the Jacobian of the underlying MLP, and we don't know how to zero-out elements of a Jacobian from a complex MLP. The second approach would allow for this, and we intend to investigate it in future work.
>
> > $L^1 \rightarrow L_1$.
>
> Thanks!  Fixed.
>
> ### Conclusions
>
> Thank you for generously outlining the paper's strengths in your original review.
> We hope this response (and especially the new qualitative results) has addressed your key concerns.
> If so, we would greatly appreciate it if you would reconsider your score.

---

### Decision · Program_Chairs · 2025-05-01

**Decision:**

Accept (poster)

**Comment:**

This paper studies sparse autoencoders and proposes a new methodology to sparsify also the computations and not just the activations, while preserving reasonable performance. The discussion between the reviewers and the authors was insightful, and the paper's scores significantly improved after the rebuttal, arriving to unanimous acceptance. Therefore, in consultation with the reviewers, the AC suggests acceptance and recommends the authors to include all rebuttal discussions and new qualitative results in the final version of the paper.